



# Long-term firn and mass balance modelling for Abramov glacier, Pamir Alay

Marlene Kronenberg[1], Ward van Pelt[2], Horst Machguth[1], Joel Fiddes[3], Martin Hoelzle[1], and Felix Pertziger[4]

[1]Department of Geosciences, University of Fribourg, Fribourg, Switzerland
[2]Department of Earth Sciences, Uppsala University, Uppsala, Sweden
[3]WSL Institute for Snow and Avalanche Research SLF, Davos, Switzerland
[4]GIS consultant, Auckland, New Zealand

**Correspondence:** Marlene Kronenberg (marlene.kronenberg@unifr.ch)

**Abstract.** Several regional studies identified heterogeneous mass changes in western High Mountain Asia over the last decade. Causes for these mass change patterns are still not fully understood. Modelling the physical interactions between glacier surface and atmosphere over several decades can provide insight into relevant processes. Such model applications, however, have data needs which are usually not met in these data scarce regions. Unique glaciological and meteorological data exist for the

Abramov glacier in the Pamir Alay range. In this study, we use weather station measurements in combination with downscaled reanalysis data to force a coupled surface energy balance–multilayer subsurface model for Abramov glacier for 52 years. Available *in situ* data are used for model calibration and validation. We find an overall negative mass balance of -0.27 m w.e. a$^{-1}$ for 1968/1969-2019/2020 and a loss of firn pore space causing a reduction of internal accumulation. Despite increasing air temperatures, we do not find an acceleration of glacier-wide mass loss over time. Such an acceleration is compensated by

increasing precipitation rates (+0.0022 m w.e. a$^{-1}$, significant at a 90% confidence level). Our results indicate a significant correlation between annual mass balance and precipitation (R$^2$=0.72).

## 1   Introduction

Spatially heterogeneous mass changes of glaciers in High Mountain Asia (HMA) during the last decade have been detected by several regional studies (e.g. Kääb et al., 2012; Brun et al., 2017; Shean et al., 2020; Jakob et al., 2021). Topographical effects in

combination with precipitation increases are suggested as reasons for balanced or positive mass changes for numerous glaciers in the Karakoram, Kunlun Shan, Pamir, Pamir Alay, and Tibetan Plateau subregions (Miles et al., 2021). Whereas reliable precipitation data from *in situ* measurements are very scarce for the region (Pohl et al., 2015), the analysis of the gridded Global Precipitation Climatology Project over thirty years has indicated a precipitation increase in the Western part of HMA due to large-scale atmospheric circulation patterns (Yao et al., 2012). Based on regional climate model data, glacier modelling

and moisture tracking, De Kok et al. (2020) conclude that changes in irrigation patterns and climate are responsible for the identified mass balance patterns in HMA.




Including *in situ* data and investigating processes at a local scale over several decades can be helpful in better understanding the influence of atmospheric conditions on glacier mass changes (Mölg et al., 2012; Zhu et al., 2020). Mass balance models of varying complexity have been applied to investigate the mass balance response of mountain glaciers to climate (e.g. Klok and Oerlemans, 2002; Pellicciotti et al., 2009; Sicart et al., 2011). Models solving the energy balance at the glacier surface are more physically based and therefore considered more suitable for longer time periods than temperature-index parametrisations (Hock and Holmgren, 2005). Energy balance models are, however, only applicable if sufficient data are available to generate a complete climate forcing and to calibrate uncertain model parameters. Important processes in the accumulation zone, which acts as a buffer against mass loss due to refreezing and water storage, are not included into surface energy balance models. Several studies have applied energy balance models coupled to multi-layer snow models to simulate refreezing processes within the snow and firn as well as heat conduction which are relevant for the glacier mass and energy balance (e.g. Reijmer and Hock, 2008; Huintjes et al., 2015b). Simulating the physical connection between the atmosphere and the glacier provides insights into the climatic control of glacier mass gain or loss (Mölg and Hardy, 2004).

In the overall data scarce HMA, only a few glaciers have been modelled with (distributed) energy-balance models which have high data requirements and even fewer studies have applied coupled energy-balance subsurface models. The availability of historical data for the Abramov glacier located in the Pamir Alay provides a unique opportunity for detailed modelling over longer periods than previous studies did for HMA glaciers: Kayastha et al. (1999) applied a point-scale energy balance model to Glacier AXOIO in the Nepalese Himalaya. Azam et al. (2014) modelled the point energy balance of Chhota Shigri Glacier glacier in the Western Himalaya. Several studies focus on glaciers and ice caps located on the Tibetan Plateau (Mölg et al., 2012; Zhang et al., 2013; Huintjes et al., 2015b, a, 2016). Zhu et al. (2020) applied a surface energy balance model to Muji Glacier, located in the north-eastern Pamir. Except for Huintjes et al. (2016), who used a coupled snowpack and ice surface energy and mass balance model (COSIMA) to reconstruct the climate on the Tibetan Plateau during the little ice age, only relative short periods up to one decade have been investigated.

Here, we apply a coupled surface energy balance – multilayer subsurface model (van Pelt et al., 2012, 2019) for a glacier located in the western part of HMA. We simulate 52 years of firn and mass balance evolution of the Abramov glacier which is located in the Pamir Alay region, Kyrgyzstan (Fig. 1). For this temperate valley-type glacier, detailed glaciological as well as meteorological measurements exist (Kislov, 1982; Pertziger, 1996; Schöne et al., 2013; Hoelzle et al., 2017; Kronenberg et al., 2021a). These *in situ* data, which are unique for the region, allow us to apply a rather complex model with relatively high data requirements. The overall aim of this study is to model the energy and mass fluxes for a period of five decades for Abramov glacier in order to better understand its firn and mass balance evolution as a response to climatic conditions. According to our knowledge, this study represents the first long-term application of a model with this degree of complexity for this data scarce region.



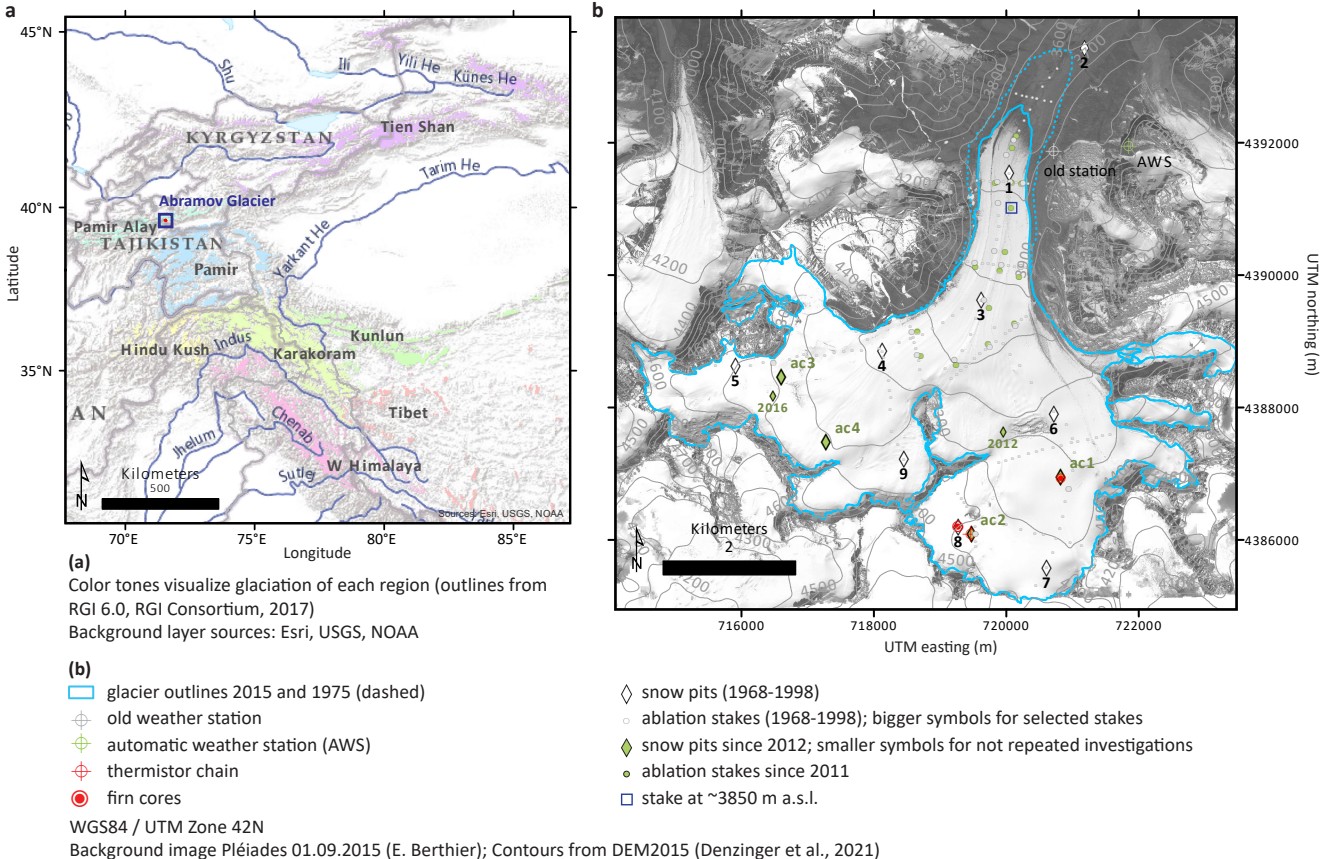

**Figure 1.** (a) Overview map showing the location of Abramov Glacier in the Pamir Alay (indicated in blue). (b) Map of Abramov Glacier and surroundings showing the location of weather stations and the mass balance observation network. (Own maps; layer sources are indicated in the legend)

## 2 Study site and data

### 2.1 Abramov glacier

Abramov glacier (39.50°N, 71.55°E) is located in the Pamir Alay (north-western Pamir) in Kyrgyzstan (Fig. 1). The north facing valley-type glacier spans an elevation range from 3650-5000 m a.s.l. and covers an area of 24 km$^2$ (in 2015). Since 1975, the glacier retreated by about 1 km (Barandun et al., 2015). An advance of the glacier tongue was reported for 1972 and 1973 (e.g Glazyrin et al., 1993). Mean annual air temperature (1968-1998) measured at 3837 m a.s.l. next to the glacier tongue was −4.1°C and annual precipitation sums for the same period are 750 mm a$^{-1}$. Abramov glacier has temperate firn conditions

(Kislov, 1977) and cold subsurface conditions were measured in the ablation area throughout the year (Kislov et al., 1977). Liquid water was observed at depths of about 8.5 to 13 m around 4400 m a.s.l. (Glazyrin et al., 1977; Kislov, 1982). The glacier





had one of the most detailed and comprehensive mass-balance time series that ended abruptly in 1999. Thanks to international efforts, mass balance measurements were re-initiated in 2011 (Hoelzle et al., 2017).

Barandun et al. (2015) have previously calculated the long term mass balance of Abramov glacier by applying a mass balance
model and calibrating it to *in situ* measurements. Their model is based on a simplified energy-balance approach by Oerlemans (2001). Denzinger et al. (2021) computed the geodetic mass balance based on aerial imagery from 1975 and satellite stereo pairs from 2015. Both studies found an overall negative mass balance. According to Barandun et al. (2021), Abramov glacier has a similar mass balance than other glaciers in the Pamir Alay. Based on *in situ* data, Kronenberg et al. (2021a) identified a trend towards a net accumulation increase for a point site on Abramov glacier, suggesting the accumulation regime changed
since the 1970s.

## 2.2   Topographical data

In this study we use digital elevation models (DEM) with an initial resolution of 4 m. The DEMs are derived from historical aerial photographs from 1975 (DEM1975) and from three Pléiades stereo pairs from 2015 (DEM2015) provided by Denzinger et al. (2021). Glacier outlines are available for 1975 and 2015 from Denzinger et al. (2021) and for 1986, 1998 and 2005 from
Barandun et al. (2015). For the DEM2015, Stainbank (2018) manually removed artefacts and created a gap-filled DEM with 25 m resolution which we use as a reference DEM. Furthermore, we use the 1975 glacier area as the reference glacier surface. We use later glacier extents (one per decade) to estimate the glacier area change (reduction with respect to 1975) over time. Based on these data we calculate an annual height change grid, an elevation grid for 1968 (DEM1968) assuming the same linear elevation trend as between 1975 and 2015 , and a reference glacier mask (mask1975) with a resolution of 100 m and an
extent of 10.7 km × 10.7 km (107×107 grid points).

## 2.3   Weather station data

### 2.3.1   Original weather station data

The original Abramov glacier weather station was located on a moraine next to the glacier tongue at 3837 m a.s.l. and was operational from October 1967 until summer 1999. The meteorological data are published in Pertziger (1996) who also compiled
the data in digital format (daily resolution). Here, we use data from January 1968 until December 1998 for the following parameters: daily average, minimum and maximum air temperature, air pressure (daily averages), relative humidity (daily averages), wind speed (daily averages), precipitation (daily sums) and cloud cover observations (daily minimum and daily average). Most meteorological parameters were recorded eight times a day, however, to our knowledge, these raw data have not been archived and only daily values were available for this study. Precipitation measurements were not corrected for undercatch. More details
about the station data can be found in the supplementary material (Table S1).

Based on these data, we created three-hourly value for each variable. We use a scaled sine function to calculate three-hourly air temperatures. The scaling factor is determined for each day based on measured minimum and maximum temperatures and daily averages of the three-hourly time series correspond to reported average air temperature measurements. For air pressure





and relative humidity the mean value is applied throughout the daily cycle. Daily precipitation sums are divided by eight to

obtain three-hourly data. During the melt season, convection is a main driver of cloudiness and cloud formation mainly takes place along the mountain ridges (Suslov and Krenke, 1980). Consequently, we assume cloud cover to be lower in the morning hours and lower for large areas of lower parts of the glacier than observed averages. We assign observed daily minimum cloud cover to the first four timesteps and daily average cloud cover for the rest of the day.

### 2.3.2   Automatic weather station data

Since August 2011, an automatic weather station (AWS) is operating at 4100 m a.s.l. about 1 km from the glacier tongue (Fig. 1) and the site of the historical meteorological observations. Please refer to Schöne et al. (2013) for technical information including sensor specifications. Long data gaps and outliers are abundant during the first operational year and we therefore only use data recorded from January 2014 onwards. In contrast to the original Abramov glacier weather station, only liquid precipitation is measured at the station, which is very rare even in summer. Hence, no precipitation data are available. Here,

we use air temperature, air pressure, relative humidity, and incoming short- and longwave radiation.

We fix a clock error and apply filtering (as described below) to the raw measurements which are registered 12 times an hour before calculating three-hourly and daily averages of several parameters as described in Stainbank (2018). At least 18 measurements are necessary (50 % of the total measurements) for three-hourly averages, otherwise we set the value to Not a Number (NaN). Daily averages are only calculated if data is available for each three-hourly time step of a day.

For air temperature and pressure we first remove extreme outliers by the application of an inter quartile range filter. Second, we calculate three-hourly mean and standard deviation for the entire time series and remove values that are more than three standard deviations from the three-hourly mean. A visual inspection of the time series indicated that for a few short periods, some outliers were not detected by the automatic filters and therefore removed manually before rerunning the second filter. For relative humidity we applied an upper (100 %) and a lower threshold (0 %), setting values below or above to NaN. The

four component net radiation sensor delivers incoming short- and longwave radiation which are used here. Furthermore, the measurements may be affected by sensor freezing. We do not specifically correct for these effects, but apply some general filters. Incoming shortwave radiation measurements below $0 \, \mathrm{W \, m^{-2}}$ are set to $0 \, \mathrm{W \, m^{-2}}$ and measurements above $1500 \, \mathrm{W \, m^{-2}}$ are excluded. Incoming longwave radiation is processed using the same approach as air temperature and pressure.

### 2.4   Gridded climate data

### 2.4.1   TopoSCALE ERA5 data

We use hourly output from ERA5 reanalysis by ECMWF Hersbach et al. (2020) for the grid point located nearest to Abramov glacier for 1980-2020. Data from the two Abramov weather stations is completely independent from the ERA5 data set, as the stations are not used during the assimilation procedure (Personal communication from H. Hersbach, ECMWF 2021). Air temperature, air pressure, relative humidity, precipitation, global and clear sky radiation are downscaled using TopoSCALE





(Fiddes and Gruber, 2014) which performs a 3D interpolation of upper air fields stored on pressure levels and corrects radiation fields according to topographical setting (e.g. slope, aspect, shading, sky view factor).

### 2.4.2 Cloud fraction calculated from TopoSCALE ERA5 data

Cloud cover fraction is not directly available from ERA5. We therefore, calculate the TopoSCALE cloud fraction from the TopoSCALE global $I$ and the clear-sky radiation $I_{cs}$. In a first step, the cloud transmissivity $\tau_{cl}$ is calculated following Klok

and Oerlemans (2002):

$$\tau_{cl} = I/I_{cs} \tag{1}$$

We use the three-hourly average values of $I$ and $I_{cs}$ to calculate $\tau_{cl}$ for each time step for which $I$ and $I_{cl}$ are both above 0. We use the the expression by Greuell et al. (1997):

$$\tau_{cl} = 1.00 - c1 \times n - c2 \times n^2 \tag{2}$$

to calculate the cloud cover $n$ from the cloud transmissivity $\tau_{cl}$. The parameters $c1$=0.128 and $c2$=0.346 are adopted from van Pelt et al. (2012). Night time cloud cover $n$ values are linearly interpolated from neighboring, non-missing values.

### 2.4.3 Bias correction of TopoSCALE ERA5 data and creation of a continuous data set

ERA5 is a global reanalysis product and its quality is dependent on density of assimilated observations, which varies globally. Central Asia in general and mountainous regions of the Pamir specifically, are relatively data poor and therefore the reanalysis

is less well constrained as compared to data rich regions such as Europe, for example. Therefore, even after downscaling (accounting for resolution differences between the model grid and point of interest) we can expect there to be residual biases, which we address with the following procedure. We aggregate TopoSCALE data to monthly averages to compare them to data from the original weather station. Biases are calculated for monthly air temperature, pressure and wind speed for the period 1980-1998 and then used to correct TopoSCALE air temperature, pressure and wind speed for 1980-2020. Monthly

average ratios between monthly aggregated station measurements and TopoSCALE data are calculated for precipitation (sums) and cloud fraction (averages) and relative humidity (averages) for the period 1980-1998 and used to correct the TopoSCALE time series for 1980-2020. The resulting cloud fraction time series for summer months (July to September) shows a reduced amplitude compared to the station time series for 1968-98. We use the precipitation time series to correct for this by setting the cloud cover to 0 for days without precipitation and to 1 if precipitation is above a threshold value. Monthly averages of the

final cloud fraction time series correspond to observed values for the years for which measurements are available. We combine the three-hourly observed data from 1968 to 1998 with the bias corrected TopoSCALE ERA5 data for the years 1999-2020 to obtain a final data set for 1968-2020. An alternative data set is created using historical measurements for the period 1968-1979 and TopoSCALE ERA5 data for the years 1980-2020. The both data sets thus differ for the period 1980-1998.





## 2.5 Glaciological data

### 2.5.1 Monthly point mass balance data 1967-1998

Monthly mass balance was measured on Abramov glacier from the beginning of the hydrological year 1967/1968 until summer 1999. Winter and annual mass balance data are published in Pertziger (1996) who also compiled the data in digital format (monthly resolution, totalling 42961 stake and 2179 snow pit measurements). Observations are reported with local coordinates, which we transferred to WGS84 / UTM Zone 42N as described in Kronenberg et al. (2021a). Monthly snow height and density measurements respective to the previous summer horizon are available from up to nine snow pit locations. Eight snow pits are located on the glacier surface and one snow pit is located just below the glacier tongue in the Koksu valley (Fig. 1b). A total of 165 stakes installed along 13 profiles were used to measure monthly surface height changes since the beginning of the hydrological year (1 October). For stakes in the ablation area, negative height changes are converted to water equivalents (w.e.) using a density of ice $900\,\mathrm{kg\,m^{-3}}$. For stakes above the firn line, a firn density of $610\,\mathrm{kg\,m^{-3}}$ is used (Kislov, 1977; Pertziger, 1996). Here, we use a constant elevation threshold of 4200 m a.s.l. to select either the firn or ice density for conversion into ablation in w.e. Pertziger (1996) converted positive stake readings to w.e. either using snow densities measured at nearby snow pits or a density of $550\,\mathrm{kg\,m^{-3}}$. If measured densities were high, they assumed a maximum density of $610\,\mathrm{kg\,m^{-3}}$. Here, we consistently use a density of $550\,\mathrm{kg\,m^{-3}}$ to convert positive stake readings to w.e. According to Pertziger (1996), the main source of error in mass balance point measurements are errors in density estimates. Another source of uncertainty is related to the time of observation. Exact dates are reported for stake readings. For data from snow pits, we do not know the exact measurement dates as only months are reported. Furthermore, the reference date or exact age of the reference surface in the snow pit are unknown due to the stratigraphic nature of snow pit measurements. We assume these dates to correspond to the beginning of the hydrological year until the end of the investigated month.

We mainly use snow pit measurements from March and annual mass balance measurements from September including data from up to eight snow pits and 165 stakes per year for 1967/1968-1997/1998. A set of 19 stakes is selected based on the location of current point observations (see following paragraph). Data from the remaining 146 stakes are used separately for validation.

### 2.5.2 Annual point mass balance data 2011-2020

Annual mass balance measurements from up to 20 points on the glacier are available since 2011/2012 (Barandun et al., 2015; Hoelzle et al., 2017). The new monitoring network consists of 16 stakes in the ablation area and up to four snow pits in the accumulation area (Fig. 1). Whereas the stake locations were roughly kept constant over time, the locations of the snow pits varied during the first years. Field visits took place once a year during July/August and exact observation dates are available. Only ablation stake readings from snow free locations are considered and the density of ice ($900\,\mathrm{kg\,m^{-3}}$) is used to calculate floating-date mass balances between the two observation dates. Due to the stratigraphic nature of snow pit observations, the beginning of the accumulation season is not known and we assume the last summer surface to have formed at the beginning of the hydrological year. Point accumulation values are thus calculated for the period from the 1 October until the observation date using the snow density and depth measurement of each snow pit.





## 2.6 Firn profile data

Density profiles dating back to the 1970s are available from deep firn pits that were located in the eastern branch of the accumulation area at ∼4400 m a.s.l. and ∼4250 m a.s.l. (Kislov, 1982). In 2018, firn cores were drilled at similar and nearby

locations. We refer to Kronenberg et al. (2021a) for a detailed description of legacy and recent firn profiles. In addition to density measurements, continuous subsurface temperature measurements from four thermistors located in a borehole at ∼4380 m a.s.l. are available from February 2018 until April 2020.

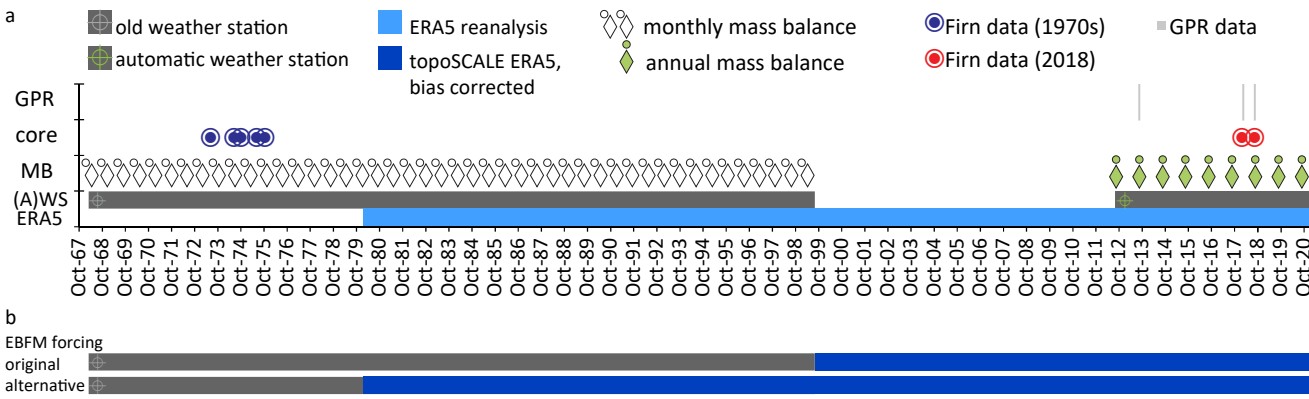

**Figure 2.** Temporal overview of data available for this study. The availability of different *in situ* and gridded datasets (ERA5 reanalysis) is shown in a. Panel b summarizes which data are used for forcing the model. We mainly show results using the original forcing. A sensitivity model run was performed using the alternative forcing. Results are shown in the supplementary material

## 3 Energy balance and firn model (EBFM)

We simulate the mass balance evolution of Abramov glacier using a coupled surface energy balance–multilayer subsurface

model by van Pelt et al. (2012). The surface energy balance model was developed following Klok and Oerlemans (2002) and the subsurface model based on SOMARS by Greuell and Konzelmann (1994). The original model was first employed to simulate the mass balance of Nordenskiöldbreen located on Svalbard. A parameterized water percolation routine was introduced by Marchenko et al. (2017) and the albedo decay scheme was updated based on the work by Bougamont et al. (2005). The model has participated in the firn meltwater Retention Model Intercomparison Project (RetMIP) as "UppsalaUniDeepPerc"

(Vandecrux et al., 2020) and it has been applied to several other glaciers located in the Arctic and the Alps (e.g. van Pelt and Kohler, 2015; van Pelt et al., 2021; Mattea et al., 2021). Here, we use the model version described by van Pelt et al. (2019) and we include an updated parametrisation for seasonal snow densification after van Kampenhout et al. (2017). As the model has been previously described in detail, only a short model overview is given in the following. For full details the reader is referred to van Pelt et al. (2019) and preceding studies.





## 3.1 Surface energy balance model

The surface energy balance model is forced by meteorological input data and calculates the energy fluxes which contribute to the surface energy budget:

$$Q_{melt} = SW_{net} + LW_{net} + Q_{sens} + Q_{lat} + Q_{sub}, \tag{3}$$

with the total energy available for melting $Q_{melt}$, the net shortwave radiation $SW_{net}$, the net longwave radiation $LW_{net}$, the turbulent sensible and latent heat fluxes $Q_{sens}$ and $Q_{lat}$ and the heat flux into the subsurface $Q_{sub}$. The model iteratively solves Eq. 3 to find the surface temperature, which is the only unknown and cannot be larger than 0°C.

Surface ablation either occurs in form of melt or as sublimation when the latent heat flux is negative. Surface accumulation occurs either in form of solid precipitation, which is calculated based on air temperature and precipitation forcing using a transition from rain to snow around a threshold temperature ($T_{sr} \pm 1$ K), or in form of riming. In case of a snow or firn surface, liquid water originating from surface melt, rain or condensation is added to the subsurface (see below) and in case of an ice surface leaving the system as runoff. For the entire modelling period, the incoming shortwave radiation $SW_{in}$ is simulated as described in Klok and Oerlemans (2002) and van Pelt et al. (2012) and accounts for grid aspect and shading by surrounding terrain. The attenuation by clouds ($\tau_{cl}$) is calculated using Eq. 2 with parameters determined for Nordenskiöldbreen, Svalbard, by van Pelt et al. (2012). We also tested the parameter values determined by Greuell et al. (1997) for an Alpine glacier. Using the values determined for Svalbard, however, yielded a better agreement between modelled $SW_{in}$ and AWS measurements. The outgoing shortwave radiation $SW_{out}$ depends on the surface albedo. In case snow is present, the albedo $\alpha_{snow}$ is a function of the snow temperature, wetness and age, as described in van Pelt et al. (2019). In absence of snow the ice albedo $\alpha_{ice}$ applies. The computation of the longwave radiation follows the Stefan-Boltzmann law. The sky emissivity depends on cloud cover, air temperature and humidity following relations in Konzelmann et al. (1994) and Greuell and Konzelmann (1994).

The turbulent sensible and latent heat fluxes $Q_{sens}$ and $Q_{lat}$ are calculated for large-scale atmospheric conditions following Oerlemans and Grisogono (2002). The equations are available in Klok and Oerlemans (2002). The subsurface heat flux $Q_{sub}$ is calculated based on the modelled temperature and conductivity of the subsurface. To obtain $Q_{sub}$, a linear gradient through the two uppermost layers is applied (see below). The heat supplied by liquid precipitation is neglected.

## 3.2 Subsurface model

The subsurface model computes the temporal evolution of subsurface temperature, density and liquid water content for discrete layers. The temperature depends on heat conduction (vertical diffusive heat flux) and refreezing of percolating melt, rain water and condensed moisture (van Pelt et al., 2012). The expressions for heat capacity and effective conductivity are adopted from Sturm et al. (1997) and Yen (1981). Heat (and mass) advection is accounted for by describing vertical layer movement on a Lagrangian grid. The penetration of shortwave radiation into the subsurface and therefore also subsurface melting are neglected. We use an updated subsurface densification routine compared to previous EBFM applications. For layer densities above $\rho_{firn}$





= 500 kg m$^{-3}$, the in van Pelt et al. (2012) described parametrisation based on the gravitational densification by Arthern et al. (2010) and modified after Ligtenberg et al. (2011) is applied. For layers with a density below $\rho_{firn}$ = 500 kg m$^{-3}$ a newly introduced fresh snow densification parametrisation following van Kampenhout et al. (2017) is used. Seasonal snow densifies due to destructive metamorphism and compaction by overburden pressure. Van Kampenhout et al. (2017) furthermore include a

snow densification due to drifting snow for densities below 350 kg m$^{-3}$. Snow drift is not included here, as we are not focusing on dynamics of fresh snow. In order to increase numerical efficiency, the fresh snow $\rho_{fresh}$ density is set to 350 kg m$^{-3}$. The densification due to destructive snow metamorphism is depth independent but varies according to the layer temperature $T$:

$$\frac{\partial \rho}{\partial t} = c_{dm3} c_{dm2} c_{dm1} exp(-c_{dm4}(T_0 - T)), \tag{4}$$

with the constants $c_{dm3} = 2.777 \times 10^{-6}$ s$^{-1}$, $c_{dm4} = 0.04$ K$^{-1}$, $c_{dm2} = 1$ ($c_{dm2} = 2$) in case of absence (presence) of liquid water

and a tapering constant $c_{dm1} = 1$ in the range of $\rho = \in [0, \rho_{max}]$ and exponentially decreasing above $\rho_{max}$. The densification due to overburden pressure $P$ (kg m$^{-3}$) depends on a viscosity coefficient $\eta$ (kg s$^{-1}$ m$^{-3}$)

$$\frac{\partial \rho}{\partial t} = \frac{P}{\eta}, \tag{5}$$

The viscosity coefficient $\eta$ is calculated following van Kampenhout et al. (2017) who slightly simplified the expression of Vionnet et al. (2012):

$$\eta = f_1 \times 4 \times \eta_0 \frac{\rho}{c_\eta} \exp(a_\eta(T_0 - T) + b_\eta \rho), \tag{6}$$

with a correction factor accounting for the liquid water content $f_1$, $\eta_0 = 7.62237 \times 10^6$ kg s$^{-1}$ m$^{-3}$, $a_\eta = 0.1$ K$^{-1}$, $b_\eta = 0.023$ m$^3$ kg$^{-1}$ and $c_\eta = 358$ kg m$^{-3}$.

The subsurface water within the firn column originates from percolating melt water, rain or condensed moisture. Preferential percolation is parametrised as described by Marchenko et al. (2017). Liquid water is instantly distributed along the depth axis

following a normal distribution until a maximum depth $z_{lim}$ unless it reaches an impermeable ice layer before. Refreezing of percolating water raises subsurface temperatures and densities until the melting point or the density of ice is reached. A small amount of liquid water is stored as irreducible water held by capillary forces and the remaining water percolates further downwards until an impermeable ice layer is reached where it forms a slush layer. The maximum irreducible water content is calculated from the porosity (ratio between pore space and the total volume of the snow/firn layer) following Schneider and

Jansson (2004). Below $z_{lim}$, percolation occurs non-preferentially following a bucket-scheme. Water moves to the next underlying layer if the refreezing capacity is eliminated (layer density or temperature reach density of ice or melting temperature) and the maximum irreducible water content is reached. Surface runoff happens instantaneous and occurs when bare ice is at the surface.

### 3.3 Model set up and initialisation

The surface energy balance and subsurface profiles are updated with a temporal resolution of three hours. We use a grid of 107×107 grid points with a horizontal resolution of 100 m, only 2654 of these grid points are assigned to the glacier using the





glacier outlines from 1975. For the initial elevation information we use the DEM1968 and update elevation information for each 3 h time step based on the linearly downscaled annual height change grid. For each 3-h time step, we derive topographical parameters used for the computation of incoming solar radiation as described in van Pelt et al. (2012). While modelling, the glacier surface is assumed to be constant and glacier grid points are classified as glacier throughout the modelling period. Distributed characteristics are thus computed for a fixed reference glacier surface and spatial varying elevation in time. We later account for a reduction of glacier surface when analysing the model output as described below.

The subsurface modelling domain consists of 100 vertical layers extending down to a depth of about 35 m below the surface in the accumulation area. The initial layer thickness at the surface is 0.1 m and at layer numbers 15, 25 and 35 (corresponding to initial depths of 1.5, 3.5 and 7.5 m). Layer thickness reduces with time due to gravitational densification of snow and firn. The subsurface layers move along the depth axes to respond to mass gain or loss at the surface. Due to accumulation, the thickness of the uppermost layer can increase until a certain threshold thickness is reached, additional accumulation leads to the creation of a new layer and the removal of the lowermost layer of the modelling domain. In the EBFM, horizontal mass and energy fluxes are neglected and mass and energy exchange between grid cells is only possible along the vertical depth axes.

To initialize subsurface conditions the model is run twice over the period 1968-98 using the three-hourly weather station data to force the model. The first iteration is started with identical subsurface conditions throughout the glacier with a vertical grid consisting of temperate ice (273.15 K). The second initialisation run is started from the final stage of a first run.

We perform several model runs for selected grid points corresponding to the locations of selected ablation stakes and snow pits indicated in Fig. 1 to adjust model parameters (as described below) using the final combined forcing consisting of station (1968-1998) and bias corrected TopoSCALE ERA5 data (1999-2020). Thereafter, we perform a final distributed model run for the period from 1 January 1968 until 31 December 2020 using the same forcing. An additional distributed run is performed using the alternative data set with a shorter period of station measurements (1968-1979) to assess the model forcing sensitivity.

### 3.4 Analysis of model output and mass balance calculation

We calculate the climatic mass balance as the sum of the surface and the internal mass balance for hydrological years (1 October-30 September) (Cogley et al., 2011). The surface mass balance is the result of accumulation (+) and ablation (-) at the surface including precipitation (+), moisture exchange (+/-), mass loss through runoff (-) and refreezing above the previous summer surface (+). The internal mass balance accounts for re-freezing and storage of liquid water below the previous summer surface.

While the model grid elevation is updated for each time step, the modelling extent is kept constant using the glacier mask from 1975 for the entire simulation period. After modelling, the glacier wide mass balance and other results are calculated for decade-wise updated glacier extents. Until the end of the hydrological year 1978, the entire model output is analysed corresponding to the glacier area of 1975. For the next ten hydrological years, a glacier mask corresponding to the glacier area from 1986 is used and output outside this domain is not considered. Masks based on outlines from 1998, 2005 and 2015 are used for 1988/1989-1997/1998, 1998/1999-2007/2008 and 2008/2009-2019/2020.





The equilibrium line altitude ELA is calculated as the mean elevation of grid points with a mass balance equal to 0 m w.e. at the end of a the hydrological year. Grid points with a negative mass balance value at the end of the hydrological year are used to calculate ablation gradients, which is the linear relation between elevation and modelled ablation.

For comparison to in situ point measurements, the modelled surface mass balance of the grid point nearest to the stake/snow pit location is used. The daily model output is aggregated at the end of the month of interest with respect to the beginning
of the hydrological year (1 October) to compare to accumulation observations. For comparison to ablation observations, data is accumulated until the reported observation date (section 2.5.1). For *in situ* observations since 2011, the stake installation dates are earlier in the melt season and model output is extracted for exact periods between stake installation and stake reading (section 2.5.2).

### 3.5 Parameter selection and calibration

In order to reflect the local conditions of Abramov glacier we adjusted several model parameters based on data available from *in situ* measurements and other studies from HMA. We selected the calibration parameters based on their relevance for our site and the existence of data. The order of calibration is chosen considering the dependence of parameters on calibration of other parameters. First the incoming radiation parameters are estimated. Thereafter, we optimise accumulation parameters and finally parameters affecting summer melt. Model parameters different from values used by van Pelt et al. (2019) are summarized in
Table 2, the last column indicates whether a parameter was optimised (y/n).

We extrapolate the meteorological forcing over the glacier surface by means of linear and temporally constant elevation gradients. For Abramov glacier, Kislov (1982) reported a temperature lapse rate of $dT/dz = -0.005 \,\mathrm{K\,m^{-1}}$ which we adopt here. The precipitation lapse rate $dPrec/dz$ is calculated from March snow pit measurements for 1969-98 (243 points). The monthly mass balance measurements available for 1969-98 indicate that no melt occurred before April and therefore, the
320 March snow pit measurements are unaffected by melt. The pressure decay parameter $dPres$ is calculated from average air pressure measurements at 3837 m a.s.l. (January 1968 - December 1998) and 4100 m a.s.l. (October 2011 - September 2020). The gradient of relative humidity $dRH/dz$= 0.01 % m$^{-1}$ is adopted from Huintjes et al. (2016), who calculated it from High Asia Refined Analysis data (Maussion et al., 2014) for two Tibetan glaciers. The chosen value is corroborated by measured relative humidity on Abramov glacier for June and July (Suslov and Krenke, 1980). The calculation of the vertical lapse rate
of potential temperature $\gamma$ is adopted from Stainbank (2018).

*In situ* measurements from the AWS for 2014-2020 as well as literature values are used to estimate incoming radiation parameters. The aerosol transmission constant $k_{aer}$ in the $SW_{in}$ formulation is determined using the linear relation between $k_{aer}$ and the elevation proposed by Klok and Oerlemans (2002). We estimate $k_{aer} = 0.98$ for the median glacier elevation (4300 m a.s.l.). The $LW_{in}$ parameter $b = 0.43$ is determined based on daily AWS data (T, RH and LW$_{in}$) for manually identified
clear sky conditions. The value is similar to the value found by Klok and Oerlemans (2002) for Morteratsch glacier. The second longwave radiation parameter $\epsilon_{cl}$ is set to the default value of 0.96.

We further use *in situ* mass balance measurements from eight snow pits located on the glacier and data from a selection of 19 ablation stakes for the period 1969-1998 (Table 1) for calibration. With these data we manually optimise a set of accumulation



and ablation parameters. Final parameters are determined by minimizing bias between modelled surface mass balances for grid
cells corresponding to point locations with *in situ* measurements. We use precipitation correction factors in order of debiasing
the precipitation forcing. The correction mainly compensates for undercatch, but also accounts for all other biases present in the
precipitation forcing. We consider March accumulation measurements to be least affected by melt and use them to calibrate the
precipitation bias correction factor $Prec_{c-w}$ applied for October-June. During summer months, the precipitation undercatch is
assumed to be reduced compared to autumn, winter and spring and we adopt a value of $Prec_{c-s}$=1.15 determined for Alpine
locations in Switzerland from Sevruk (1985). The bias between modelled and measured June and July snow pit measurements
is reduced by a two-parameter exploration for $\alpha_{fresh}$ and $t^*_{wet}$. And finally, $\alpha_{ice}$ is optimised using September surface mass
balance measurements (snow pits and ablation stakes).

Parameters in the subsurface model are default values except for the critical density of destructive metamorphism $\rho_{max}$
which is optimised to obtain a better fit between modelled and measured subsurface densities at ∼4410 m a.s.l.

**Table 1.** Number of point measurements used for calibration of surface energy balance parameters (cf. Table 2). Accumulation measurements
were performed at eight sites on the glacier surface and all available measurements are used for the selected months. We use 19 out of 165
ablation stakes for calibration. Stakes are selected to correspond to current observation sites

| type | period | use | number of data points |
|---|---|---|---|
| March snow pit measurements | 1969-1998 | calibration $Prec_{c-w}$ | 225 |
| June snow pit measurements | 1969-1998 | calibration $\alpha_{fresh}, t^*_{wet}$ | 203 |
| July snow pit measurements | 1969-1998 | calibration $\alpha_{fresh}, t^*_{wet}$ | 141 |
| annual mass balance measurements | 1969-1998 | calibration $\alpha_{ice}$ | 532 |

## 4 Results

### 4.1 Long term mass balance

The distributed mean annual mass balance for 1968/1969-2019/2020 is shown in Fig. 3. Table 3 lists modelled glacier-wide
climatic mass balances and internal accumulation for different periods; annual climatic mass balances are shown in Fig. 4.
Overall, the mass balance of Abramov glacier is negative for the years from 1968/1969 to 2019/2020. We do not find a
significant trend in annual mass balances (+0.0002 m w.e. a$^{-1}$, p-value=0.979). The most negative mass balances are modelled
for the beginning of the modelling period. The two decades between 1978 and 1998 are characterised by almost balanced mass
budget. Mass balances are more negative after the end of historical investigations in 1998.

The elevation distribution of mass balance for different decades is shown in Fig. 5b. The accumulation is lowest during the
first decade (1968/1969-1977/1978) and highest during the last modelled decade (2008/2009-2017/2018). Ablation is highest
during the first decade, followed by the second last decade (1998/1999-2007/2008).





**Table 2.** List of EBFM parameter choices with references. For optimised parameters (opt), the initial value is given in brackets.

| | parameter | unit | value (initial) | reference | opt |
|---|---|---|---|---|---|
| $dT/dz$ | temperature lapse rate | $\mathrm{K\,m^{-1}}$ | -0.005 | Kislov (1982) | n |
| $dPrec/dz$ | precipitation lapse rate (factor) | $\mathrm{m^{-1}}$ | 0.0013 | calculated from in situ data | n |
| $dPres$ | Pressure decay parameter | - | $-1.45*10^{-4}$ | calculated from in situ data | n |
| $dRH/dz$ | relative humidity gradient | $\mathrm{\%\,m^{-1}}$ | 0.01 | Huintjes et al. (2016) | n |
| $\gamma$ | potential temperature lapse rate | $\mathrm{K\,m^{-1}}$ | 0.0055 | from in situ and pseudostation data | n |
| $Prec_{c-w}$ | precipitation bias correction winter (factor) | - | 1.85 (2) | - | y |
| $Prec_{c-s}$ | precipitation bias correction summer (factor) | - | 1.15 | Sevruk (1985) | n |
| $\alpha_{ice}$ | ice albedo | - | 0.23 (0.3) | Mölg et al. (2012) | y |
| $\alpha_{fresh}$ | fresh snow albedo | - | 0.81 (0.85) | Mölg et al. (2012) | y |
| $t_{wet}^*$ | albedo decay time scale wet snow | days | 7 (15) | van Pelt et al. (2019) | y |
| $P_{th}$ | snowfall threshold to reset to $\alpha_{fresh}$ | $\mathrm{m\,w.e.\,s^{-1}}$ | 3.5d-8 | Zhu et al. (2020) | n |
| $C_b$ | background turbulent exchange coefficient | - | 0.0037 | Klok and Oerlemans (2002) | n |
| $c1$ | cloud transmissivity parameter | - | 0.128 | van Pelt et al. (2012) | n |
| $c2$ | cloud transmissivity parameter | - | 0.346 | van Pelt et al. (2012) | n |
| $\mathrm{k}_{aer}$ | aerosol transmission constant | - | 0.982 | Klok and Oerlemans (2002) | n |
| $b$ | clear sky emissivity parameter | - | 0.433 | calculated from in situ data | n |
| $\epsilon_{cl}$ | overcast emissivity | - | 0.960 | default value | n |
| $\lambda$ | optical thickness empirical constant | - | 3.00 | Smith (1966) | n |
| $z_{lim}$ | preferential percolation depth | m | 6.0 | default value | n |
| $\rho_{max}$ | critical density destructive metamorphism | $\mathrm{kg\,m^{-3}}$ | 300 (200) | - | y |
| $\rho_{fresh}$ | fresh snow density | $\mathrm{kg\,m^{-3}}$ | 350 | Klok and Oerlemans (2002) | n |

## 4.2 Point surface mass balance versus measurements

Figs. 6 and 7 compare modelled surface mass balance to *in situ* measurements. Fig. 6 shows the final calibration results for eight snow pit locations at the end of March (panel a) and annual mass balance measurements from 19 stakes and up to eight snow pit locations (panel b). Both Fig. 6a and b contain data for the period 1968/1969-1997/1998. A different set of point measurements is used for model validation shown in Fig. 7. In Fig. 7a, an additional set of annual mass balance observations from 146 stake locations for 1968/1969-1997/1998 (equal to the calibration period, stakes not used for calibration) is compared to modelled surface mass balances. In Fig. 7b, the model output is compared to mass balance measurements for 2011/2012-2019/2020.






**Table 3.** Mean glacier-wide climatic mass balance, internal accumulation Equilibrium Line Altitude ELA and ablation gradients for each decade and different periods used for comparison with other studies. The periods are hydrological years (e.g. 1968/1969-1977/1978 refers to 1 October 1968 - 30 September 1978) unless precise dates are specified. The mean glacier surface used for the mass balance calculation is also indicated

| period | climatic mass balance | internal accumulation | ELA | ablation gradient | mean glacier surface |
|---|---|---|---|---|---|
| | m w.e. $a^{-1}$ | m w.e. $a^{-1}$ | m a.s.l. | m w.e. $(m\,a.s.l.)^{-1}$ | $km^2$ |
| 1968/1969-1977/1978 | -0.63 | 0.11 | 4250 | 0.0077 | 26.54 |
| 1978/1979-1987/1988 | -0.04 | 0.1 | 4170 | 0.0078 | 25.78 |
| 1988/1989-1997/1998 | -0.08 | 0.13 | 4180 | 0.0083 | 25.24 |
| 1998/1999-2007/2008 | -0.33 | 0.12 | 4220 | 0.0076 | 25.16 |
| 2008/2009-2017/2018 | -0.23 | 0.1 | 4222 | 0.0066 | 25.06 |
| 1968/1969-1997/1998 | -0.25 | 0.12 | 4200 | 0.008 | 25.85 |
| 1968/1969-2013/2014 | -0.26 | 0.12 | 4207 | 0.0077 | 25.6 |
| 1971/1972-1993/1994 | -0.26 | 0.11 | 4200 | 0.0079 | 25.87 |
| 1975/1976-2014/2015 | -0.26 | 0.11 | 4208 | 0.0076 | 25.42 |
| 15 Jul 1975-1 Sep 2015 | -0.30 | 0.11 | 4208 | 0.0076 | 25.42 |
| 1998/1999-2019/2020 | -0.31 | 0.11 | 4225 | 0.007 | 25.11 |
| 2011/2012-2019/2020 | -0.38 | 0.08 | 4248 | 0.0065 | 25.06 |
| 1968/1969-2019/2020 | -0.27 | 0.11 | 4211 | 0.0076 | 25.54 |

### 4.3 Variation of energy fluxes and climate variables

Figs. 8a and b show mean modelled energy fluxes per decade for a grid point in the ablation area and site 2 in the accumulation
area. The only negative heat flux at both sites is the latent heat flux, which becomes less negative over time. It increases for both sites by $+0.05\,W\,m^{-2}\,a^{-1}$ (p-value = 0.000). In the ablation area, the total melt energy is lowest during the second modelled decade (1978/1979-1987/1988, Fig. 8a) and is more homogeneous in the accumulation area (Fig. 8b). In the ablation area, the net radiation flux ($R_{net}$=27.6 W m$^{-2}$) clearly dominates over the sensible heat flux ($Q_{sens}$=18.4 W m$^{-2}$) at the beginning of the modelling period. The sensible heat flux increases in the course of time ($+0.06\,W\,m^{-2}\,a^{-1}$, p-value = 0.001) and almost equals
the net radiation during the last modelled decade ($Q_{sens}$=20 W m$^{-2}$ and $R_{net}$=23.8 W m$^{-2}$, Fig. 8a). In the accumulation area, the sensible heat flux always dominates over the net radiation and also increases with time by $+0.04\,W\,m^{-2}a^{-1}$, p-value = 0.003 (Fig. 8b). The net radiation mainly varies due to a reduction in incoming shortwave radiation, which is highest at the beginning of the modelling period and decreases over time (ablation area: $-0.4\,W\,m^{-2}\,a^{-1}$ p-value = 0.000; accumulation area: $-0.3\,W\,m^{-2}\,a^{-1}$ p-value = 0.000). At the same time a lower, but significant increase of the incoming longwave radiation is found for the ablation area (Table 4). Mean decadal energy fluxes are given in more detail in Table S2 in the supplementary
material.

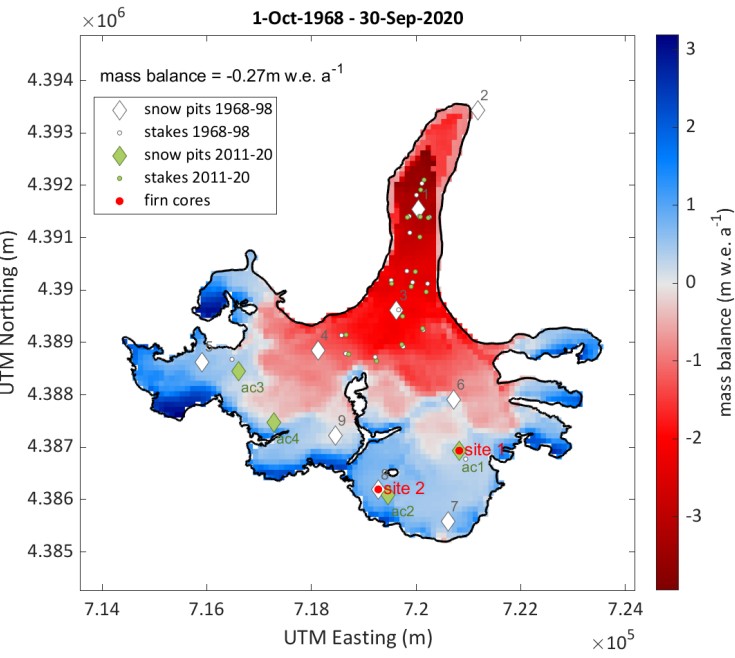

**Figure 3.** Modelled mean annual distributed mass balance for updated glacier extents for the period from 1 October 1968 to 30 September 2020. Note, that the mean annual mass balance for the entire period and updated glacier surfaces is shown. Values are thus reduced on the glacier tongue, where the glacier area reduced over time. Furthermore, the location of point observations used for calibration and validation are indicated with symbols, further details are shown in the legend.

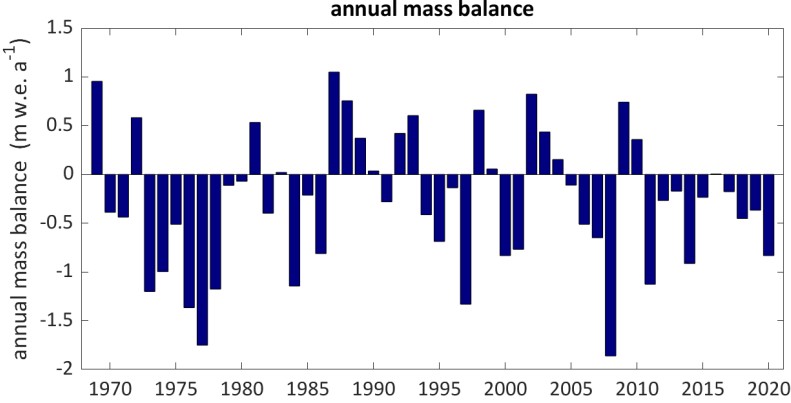

**Figure 4.** Modelled mean annual mass balance for updated glacier extents.

Significant increasing trends for model forcing air temperature (mean over hydrological years) and mean summer air temperatures are found for 1968/1969-2019/2020 (Table 4). Trends are also significant for air pressure (increase), relative humidity (decrease), cloud cover fraction (decrease) and precipitation sums (increase) (Table 4).





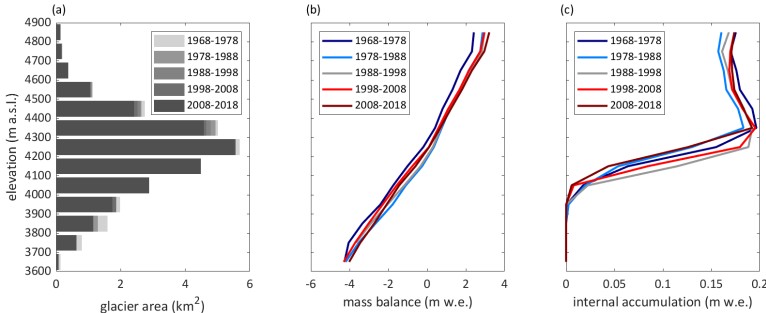

**Figure 5.** Modelled mass balance and internal accumulation versus elevation. The elevation distribution of the glacier area for different glacier extents are shown in panel panel (a). In panel (b) the climatic mass balance is plotted versus the elevation for the modelled decades. In panel (c) the internal accumulation is plotted versus elevation. The periods are hydrological years (e.g. 1968-1978 refers to 1 October 1968 - 30 September 1978).

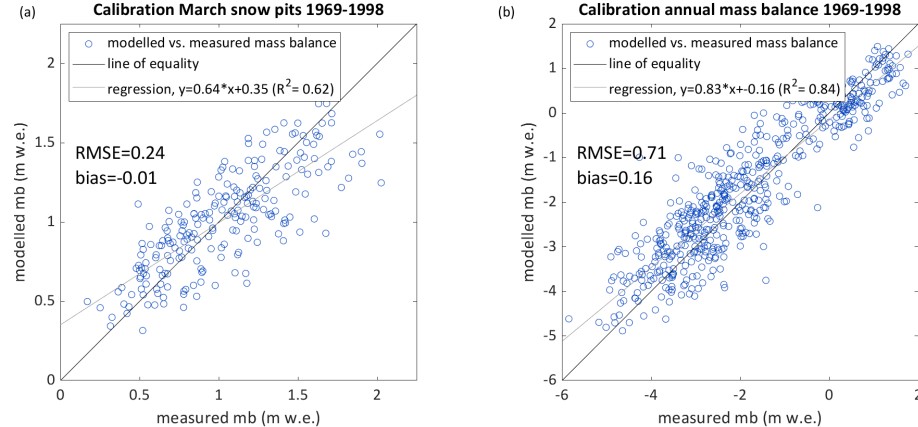

**Figure 6.** Model parameter calibration: Measured versus modelled surface mass balance for end of March (a) and September (b) for point measurements which are used for model calibration (1968/1969-1997/1998). The final calibration is shown. The bias between modelled and measured w.e. for snow pit measurements at the end of March (a) is minimised by optimising the precipitation correction factor. The bias between modelled and measured September mass balance measurements is optimised by tuning the albedo of ice.

In figure 9, annual mass balances are plotted against summer temperatures and precipitation sums over the hydrological year.

## 4.4 Internal accumulation

The average glacier-wide modelled internal accumulation is $0.11 \, \mathrm{m\,w.e.\,a^{-1}}$ (Fig. 10). Internal accumulation appears in large parts of the accumulation area and is more pronounced in the orographic right accumulation area. In Fig. 5c the internal accumulation is plotted against elevation for different decades. Highest internal accumulation rates are modelled for the years

1968/1969-1977/1978 between 4400 and 4500 m a.s.l. The decades 1988/1989-1997/1998 and 1998/1999-2007/2008 are char-



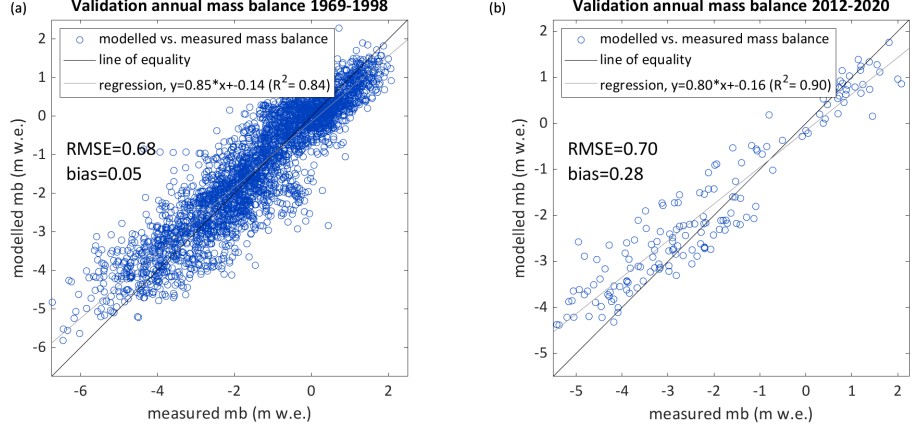

**Figure 7.** Model validation: Measured versus modelled annual surface mass balance for an independent set of point measurements from 1968/1969-1997/1998 (a) and from 2011/2012-2019/2020 (b).

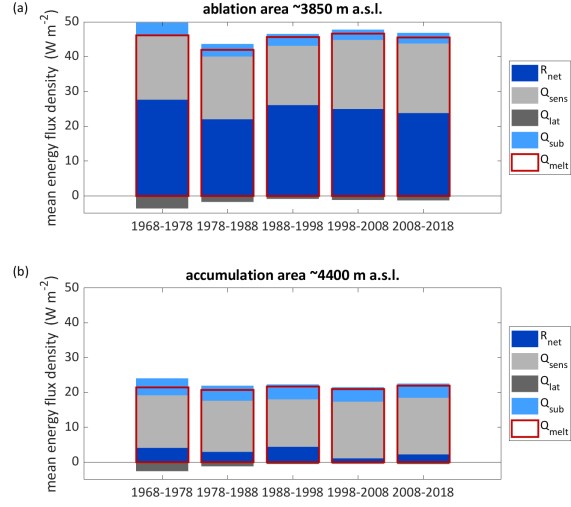

**Figure 8.** Mean modelled energy fluxes per decade for a grid point located in the ablation area at ∼3850 m a.s.l. (a) and in the accumulation area at at ∼4400 m a.s.l. (b). The point locations are indicated in Fig.1. $R_{net}$ is the net radiation, $Q_{sens}$ the sensible and $Q_{lat}$ the latent heat flux, $Q_{lat}$ the heat flux from/into the subsurface and $Q_{melt}$ the total energy available for melt. The periods are hydrological years (e.g. 1968-1978 refers to 1 October 1968 - 30 September 1978).

acterised by higher internal accumulation rates at lower elevations around 4250 m a.s.l. Lowest internal accumulation rates are modelled for the second and the last decade.





**Table 4.** Trends and p-values for climate variables and energy balance components for the period from the 1 October 1968 until the 30 September 2020 (hydrological years) are listed for a grid point in the ablation area at ~3850 m a.s.l. Values in brackets refer to site 2 located in the accumulation area at ~4400 m a.s.l. y/n stand for significant or not significant at a 90% confidence level. The point location of both grid points are indicated in Fig.1. The trends for glacier-wide mass balance, glacier wide internal accumulation and original climate forcing (for the elevation of the weather station at 3837 m a.s.l.; 1 October 1968 - 30 September 2020) are also given

| variable | trend | unit | p-value | significant |
|---|---|---|---|---|
| mean annual air temperature | +0.0295 (+0.0236) | $K\,a^{-1}$ | 0.000 (0.000) | y (y) |
| mean summer (JJAS) air temperature | +0.0208 (+0.0149) | $K\,a^{-1}$ | 0.003 (0.003) | y (y) |
| annual precipitation sum | +0.0018 (+0.0068) | m w.e. $a^{-1}$ | 0.421 (0.076) | n (y) |
| mean annual incoming shortwave radiation | -0.4297 (-0.2815) | $W\,m^{-2}\,a^{-1}$ | 0.000 (0.000) | y (y) |
| mean annual outgoing shortwave radiation | -0.3989 (-0.2362) | $W\,m^{-2}\,a^{-1}$ | 0.000 (0.000) | y (y) |
| mean annual incoming longwave radiation | +0.0702 (+0.0432) | $W\,m^{-2}\,a^{-1}$ | 0.074 (0.251) | y (n) |
| mean annual outgoing longwave radiation | +0.0507 (+0.0388) | $W\,m^{-2}\,a^{-1}$ | 0.005 (0.029) | y (y) |
| mean annual sensible heat flux | +0.0550 (+0.0417) | $W\,m^{-2}\,a^{-1}$ | 0.001 (0.003) | y (y) |
| mean annual latent heat flux | +0.0515 (+0.0473) | $W\,m^{-2}\,a^{-1}$ | 0.000 (0.000) | y (y) |
| mean annual heat flux from the subsurface | -0.0202 (-0.0153) | $W\,m^{-2}\,a^{-1}$ | 0.000 (0.001) | y (y) |
| mean annual glacier wide mass balance | +0.0002 | m w.e. $a^{-1}$ | 0.979 | n |
| mean annual internal accumulation | -0.0003 | m w.e. $a^{-1}$ | 0.159 | n |
| mean annual air temperature 3837 m a.s.l. | +0.0222 | $K\,a^{-1}$ | 0.000 | y |
| mean summer (JJAS) air temperature 3837 m a.s.l. | +0.0136 | $K\,a^{-1}$ | 0.047 | y |
| annual precipitation sum 3837 m a.s.l. | +0.0022 | m w.e. $a^{-1}$ | 0.074 | y |
| mean annual cloud cover fraction 3837 m a.s.l. | -0.0012 | - | 0.002 | y |
| mean annual relative humidity 3837 m a.s.l. | -0.1240 | $\%\,a^{-1}$ | 0.002 | y |
| mean annual air pressure 3837 m a.s.l. | +1.6274 | $Pa\,a^{-1}$ | 0.008 | y |

## 4.5 Firn evolution

Modelled subsurface densities and temperatures are shown for two sites in Figs. 11 and 12. In Figs. 13 and 14, modelled
subsurface densities and temperatures are shown together with *in situ* measurements.

Modelled subsurface densities are higher at ~4250 m a.s.l. (Fig. 11a-c) than at ~4400 m a.s.l. (Fig. 12a-c) and increase over time at both sites. At site 1, a significant increasing trend of subsurface densities for the depth range of 0-10 m is found for 1968/1969-2019/2020 (+1.14 kg m$^{-3}$ a$^{-1}$, p-value = 0.014), whereas the trend is significant at site 2 when the first two decades are excluded (+1.53 kg m$^{-3}$ a$^{-1}$, p-value = 0.005 for 1988/1989-2019/2020). At the lower site (~4250 m a.s.l., Fig. 11a-c)
densities at depth reach the density of ice. Modelled firn densities are compared to measurements for four different dates in Fig. 13. The mean biases between modelled and measured densities for the depth covered by measurements are -18.9 kg m$^{-3}$ for June 1973, +52.6 kg m$^{-3}$ for June 1974, +20.4 kg m$^{-3}$ for June 1975 and -23.8 kg m$^{-3}$ for August 2018.



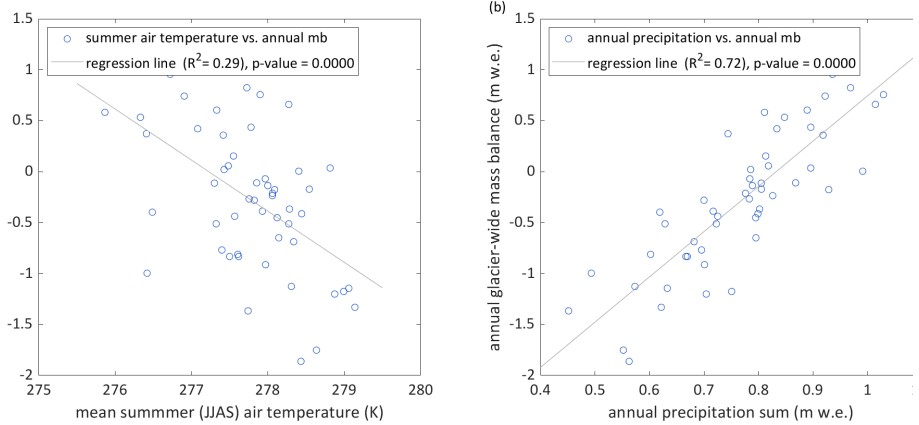

**Figure 9.** Climate variables at the station location (3837 m a.s.l.) versus modelled mass balance: Mean summer air temperature versus annual mass balance (a) and annual precipitation sums versus mean annual mass balance.

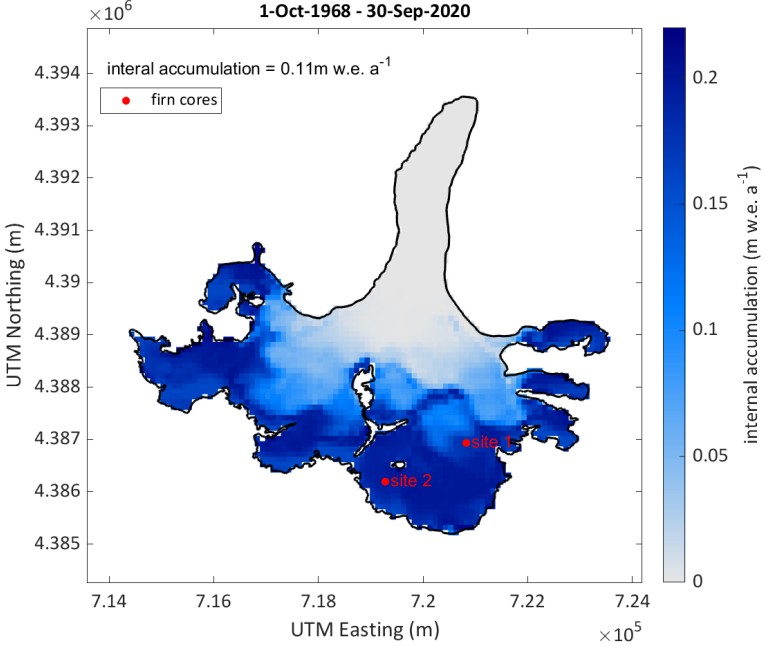

**Figure 10.** Map of mean annual internal accumulation for the mass balance years 1968/1969-2019/2020. Location of firn core drill sites used for validation are indicated with red dots.

For early years, the modelled subsurface temperatures indicate temperate firn conditions and propagation of winter cooling down to depths of about 10 m in the accumulation area at ~4250 m a.s.l. (Fig. 11d). In later years, the cold content propagates




to greater depths and overall cold subsurface conditions are modelled for most recent years (Fig. 11e-f). For the depth range of
0-10 m, the subsurface cooling trend is -0.036 K a$^{-1}$ (p-value=0.000 for 1968/1969-2019/2020).

The modelled firn temperatures indicate temperate firn conditions and propagation of winter cooling down to depths of about
10 m in the accumulation area at ~4400 m a.s.l. (Fig. 12d-f). Also for site 2, an overall cooling trend is identified (-0.0041 K a$^{-1}$
p-value=0.043 for 1968/1969-2019/2020). In Fig. 14 modelled firn temperatures are compared to thermistor measurements
from spring 2018. Measured as well as modelled temperatures remain temperate at depths greater than about 10 m. In March
(Fig. 14 a,b), modelled temperatures near the surface correspond well with observations. At a depth of about 7 m, the modelled
temperatures are a bit higher. In April (shown in Fig. 14 c), modelled firn temperatures are warmer than measurements also for
the uppermost thermistor location. At a depth of about 7  minimum subsurface temperatures are recorded for 6 June 2018. On
this date, modelled firn temperatures are temperate except for a small zone around 7 m depth (Fig. 14 d).

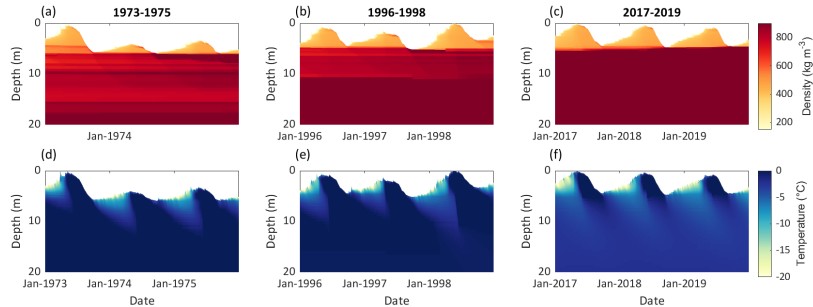

**Figure 11.** Modelled subsurface conditions at site 1 (~4250 m a.sl.) for three selected periods: 1973-1975 (panels a and d), 1996-1998 (b,e)
and 2017-2019 (c,f). Panels a, b and c show the subsurface density and panels e, d and f the subsurface temperature. The location of the site
is indicated in Fig.1 b

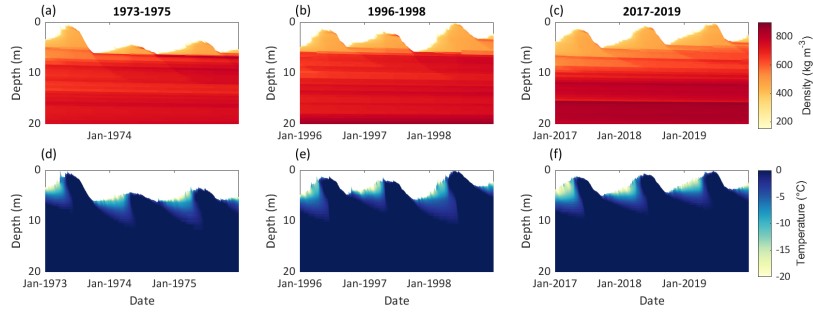

**Figure 12.** Modelled subsurface conditions at site 2 (~4400 m a.sl.) for three selected periods: 1973-1975 (panels a and d), 1996-1998 (b,e)
and 2017-2019 (c,f). Panels a, b and c show the subsurface density and panels e, d and f the subsurface temperature. The location of the site
is indicated in Fig.1 b





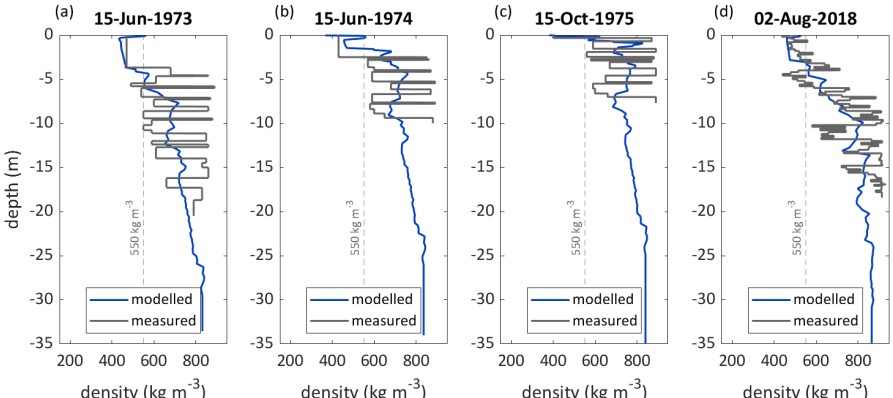

**Figure 13.** Modelled and measured subsurface densities for site 2 (~4400 m a.sl.). Measured subsurface densities from the 1970s are digitised from figures 2.1 and 2.2 in Kislov (1982) (a-c) and own measurements are shown for 2018 (d). The location of the site is indicated in Fig.1b.

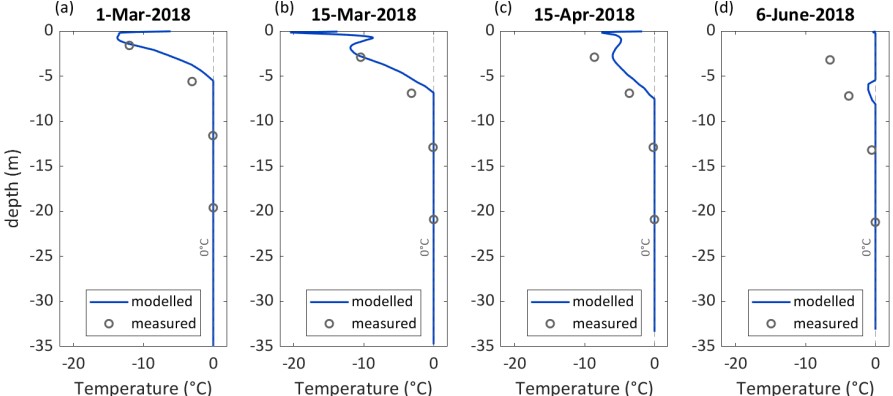

**Figure 14.** Modelled and measured subsurface temperature for a station located nearby site 2 (~4400 m a.sl.). The location of the site is indicated in Fig.1b.

## 4.6 Results of additional model run with alternative forcing

An additional distributed run (hereafter named alternative run) is performed using the alternative data set with a shorter period of station measurements to assess the model forcing sensitivity. The original model forcing consists of station measurements for 1968-1998 and bias corrected TopoSCALE ERA5 for 1999-2020, whereas the additional model run is forced by station measurements for 1968-1979 and corrected TopoSCALE ERA5 for 1980-2020. Results from this additional run yield a more negative mass balance than the original data set of $-0.34\,\mathrm{m\,w.e.\,a^{-1}}$ for 1968-2020. For the years 1980-1998, the alternative forcing yields a balance of $-0.23\,\mathrm{m\,w.e.\,a^{-1}}$, whereas the balance was almost balanced using the original forcing ($-0.05\,\mathrm{m\,w.e.\,a^{-1}}$). The original forcing produces a higher internal accumulation and more positive mass balances for most of the glacier surface. Figures of the alternative model run are shown in the supplementary material (Figs. S2-S6).





# 5 Discussion

## 5.1 Long-term mass balance and firn evolution


The modelled long-term mass balance indicates an overall mass loss of Abramov glacier for the period 1968/1969-2019/2020 which is generally in agreement with the findings of other recent studies. Barandun et al. (2015) and Denzinger et al. (2021) found somewhat more negative mass balances. Barandun et al. (2015) estimate the mass balance for 1967/1968-2013/2014 as $-0.44 \pm 0.10$ m w.e. a$^{-1}$. Their estimate is based on the application of a calibrated surface mass balance model and a sim-

ple approximation of internal accumulation as well as basal ablation which together contribute $+0.07$ m w.e. a$^{-1}$. In Fig. 15 annual mass balances of different studies are compared to our results: Previous analysis of annual mass balance data for 1971-1994 yielded more negative mass changes (Barandun et al. (2015): $-0.39$ m w.e. a$^{-1}$, Dyurgerov (2002): -0.50 m w.e. a$^{-1}$, and Pertziger (1996): $-0.61$ m w.e. a$^{-1}$) than the EBFM ($-0.26$ m w.e. a$^{-1}$). Denzinger et al. (2021) calculate the geodetic mass balance as $-0.38 \pm 0.12$ m w.e. a$^{-1}$ for 15 July 1975 until 1 September 2015. Our result for the same time period

($-0.30$ m w.e. a$^{-1}$) is within their calculated uncertainties.

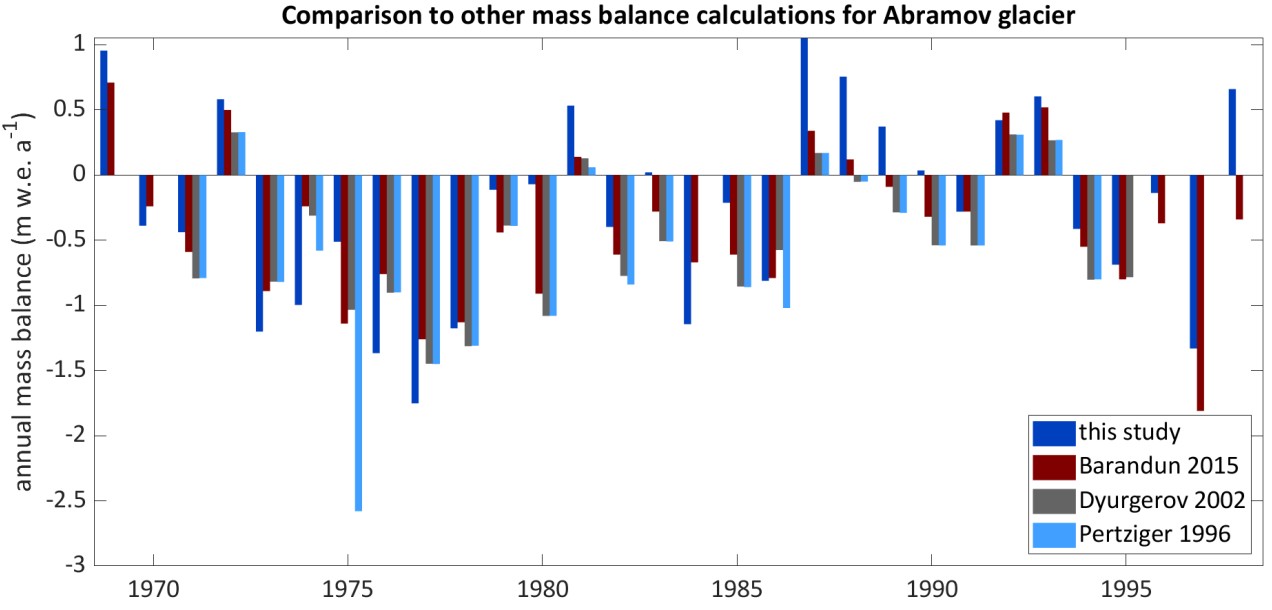

**Figure 15.** Comparison of annual mass balance results to previously published values for Abramov glacier (Pertziger, 1996; Dyurgerov, 2002; Barandun et al., 2021)

.

We do neither find a significant trend in the evolution of annual mass balances ($+0.0002$ m w.e. a$^{-1}$, p-value = 0.979), nor in the reduction of mean annual internal accumulation (-0.0003 m w.e. a$^{-1}$, p-value = 0.159) for the period 1968/1969-2019/2020.



But significant trends are found for all variables used as model forcing (Table 4). Different changes may thus compensate for each other. Comparing annual mass balances with annual precipitation sums and summer air temperatures shows that annual

mass balances are stronger related to precipitation sums than to summer air temperatures (Fig. 9). The effect of the substantial warming (+0.0222 K a$^{-1}$, p-value = 0.000) seems thus to be attenuated by increasing precipitation (+0.0022 m w.e. a$^{-1}$, p-value = 0.074). The increase of net accumulation in the accumulation area (Fig. 5b) may thus be related to an increase of solid precipitation (Tables S2 and 4). Barandun et al. (2015) also find an increase of accumulation based on their modelling results. An accumulation increase is as well in agreement with Kronenberg et al. (2021a), who identified a net accumulation increase for

a point location at ∼4400 m a.s.l. Abramov glacier might be affected by the same changes in precipitation patterns that could be partly responsible for the mass balance anomaly in western HMA (e.g. Miles et al., 2021). Barandun et al. (2015) furthermore describe a tendency towards more ablation during recent decades. We find high ablation rates for the first decade and for 1998/1999-2007/2008 (Fig. 5b). In contrast to previous studies (e.g. Dyurgerov and Dwyer, 2001), we do not find a steepening of ablation gradients. The ablation gradient is higher for 1968/1969-1997/1998 (0.0080 m w.e. (m a.s.l.)$^{-1}$) than for 1998/1999-

2019/2020 (0.0070 m w.e. (m a.s.l.)$^{-1}$). Miles et al. (2021) calculate an ablation gradient of 0.0084 m w.e. (m a.s.l.)$^{-1}$ for 2012-2016. Whereas their ablation gradient is higher, their ELA of 4163 m a.s.l. is located lower than the here modelled ELA (Table 3).

We find a relevant contribution of refreezing below the last summer horizon to the overall mass balance. As visible in Fig. 5c and from data in Table 3, internal accumulation also evolves in the course of time. Important changes are visible for the

lower accumulation area up to ∼4300 m a.s.l., where the internal accumulation strongly increases with elevation and lowest values are modelled for the last decade (Fig. 5c). Site 1, for which the subsurface evolution of density is shown in Fig. 11a-c, is located within this zone. For this point location, the subsurface density reaches the density of ice what hinders internal accumulation. An increase of subsurface densities also happens around ∼4400 m a.s.l. as shown in Fig. 12a-c. The lowest internal accumulation rates are found for the second (1978/1979-1987/1988) and the last (2008/2009-2017/2018) modelled

decade. Both of them follow decades with clearly negative mass balances and high refreezing rates (Table 3). This is confirmed by *in situ* measurements from several years during the first decade which indicate negative annual mass balances at several or all point observation sites in the accumulation area (Kislov, 1982; Pertziger, 1996). The model results suggest, that thanks to high accumulation and limited ablation, the firn could recover during the following years allowing for higher internal accumulation rates thereafter. Results from the most recent decade and years, however, indicate that the necessary pore space for refreezing

is again reducing. Whereas the subsurface density at ∼4250 m a.s.l. decreases since the beginning of the modelling period, a significant decrease is also found for ∼4400 m a.s.l. after 1987/1988. Abramov glacier is thus loosing its mass loss buffering capacity through refreezing. The loss of pore space occurs despite the high amounts of solid precipitation (Table S2) responsible for the high accumulation compared to the first decade (Fig. 5b) and affects large areas of the accumulation zone (Fig. 5a). For the upper elevation bands, modelled internal accumulation remains high also for recent decades. With further warming, these

zones are, however, expected to also lose pore space in future.

Mass balance observations often neglect internal accumulation or account for the process only in a simplified manner. This applies for mass balance calculation based on glaciological data but also for geodetic mass change estimates. The latter usually





use constant density conversion factors to convert elevation to mass changes. Our results indicate that internal accumulation plays an important role also for glaciers with temperate accumulation areas. Whereas mass change calculations neglecting the

internal accumulation may be overestimating the mass loss for periods with available pore space, a smaller overestimation of mass loss may become more likely as glaciers loose pore space.

Our results suggest that the firn of Abramov glacier is currently losing pore space and that the loss of pore space is more advanced in the lower accumulation area. The warm and also cold infiltration zones of glaciers located at higher locations may not yet be affected. Evidence from *in situ* measurements in HMA is hardly available. Lambrecht et al. (2020) expect a strong

elevation gradient of melt water refreezing on Fedschenko Glacier, western Pamir. Only at elevations above 5200 m a.s.l. the annual accumulation layers are currently preserved, but evidence of summer melt was found at high elevations of more than 5300 m a.s.l. Based on a sensitivity experiments with a region-wide application of a relatively simple model, Wang et al. (2019) concluded that under warmer conditions, refreezing will increase for continental glaciers and decrease for glaciers located in more humid and warmer environments. In their study, Abramov glacier is located in a region for which they find an increase of

refreezing. Our results indicate a decrease of internal accumulation related to the retreat of the firn line. A retreat of the firn line is likely also occurring elsewhere, however, the underlying processes are often not sufficiently included into simple, regional approaches.

## 5.2   Uncertainties and validation

The comparison of modelled and measured surface mass balances (Fig. 7) and especially the comparison of subsurface proper-

ties (Figs. 13, 14) show, that the EBFM overall satisfyingly reproduces the observations. Our study furthermore demonstrates, that results with higher confidence can be reproduced for the period for which *in situ* measurements serve as a model forcing.

The presented modelling results are influenced by several sources of uncertainties which contribute to deviations between observed and measured data. Especially towards the end of the modelling period, the model underestimates melt rates in the lower part of the ablation area (7b). The modelled ablation gradient may thus be somewhat underestimated for recent decades.

In the following we discuss how these uncertainties may affect the presented results.

Several uncertainties are related to the model setup. The spatial vertical and horizontal as well as temporal resolutions are too coarse to resolve all the observed variations and to reproduce all relevant processes. Consequences of the horizontal resolutions are for example that several point observations shown in Figs. 6 and 7 are located within one modelled grid cell. The model is unable to fully reproduce the spatial heterogeneity of *in situ* observations. This also applies for subsurface conditions. On

Abramov glacier highly variable firn conditions and accumulation rates are found based on firn cores and ground penetrating radar data (Kronenberg et al., 2021a). This study showed that annual accumulation, measured in firn cores, can vary by a factor of 1.5 over a horizontal distance of 250 m. The model satisfyingly reproduces the bulk densities measured at site 2 (Fig. 13), however the lower measured densities at nearby drill sites (c4381 and c4382 in Kronenberg et al. (2021a)) are not reproduced satisfactorily as the model does not simulate the horizontal small scale variability of accumulation and does furthermore not

consider fine scale processes such as wind redistribution of snow. Another uncertainty related to the model set up is the use of a linearly updated glacier surface elevations and decade-wise changing glacier outlines. These approximations of topographical





changes may be responsible for underestimations (overestimations) of melt rates on the glacier tongue if surface elevations are to high (to low). The initial conditions derived from initialisation runs using the first thirty year of climate input are a further source of uncertainties. It is very likely, that the atmospheric conditions were different from the used data set prior to
the modelling period. Nevertheless, modelled subsurface conditions agree well with measurements during early years of the modelling period (Fig. 13a-c).

An important source of uncertainties is related to the model forcing. While station data is available for the first thirty years, data from a gridded climate product (ERA5) has to be used for the remaining modelling period. To homogenize the input, the downscaled ERA5 data is bias corrected to match with monthly averages of observations for the overlapping period. Despite
these corrections, the differences in both data sets affect the model output as evident from the results of the alternative model run which yielded a more negative mass balance for the period for which both data sets are available (alternative forcing: -0.23 m w.e. a$^{-1}$ original forcing -0.05 m w.e. a$^{-1}$). Uncertainties in the model forcing may thus be a further reason for the misfits between measured and modelled surface mass balances.

The parametrisations used in modelling energy balance and subsurface conditions are based on simplifications causing
further uncertainties. The snow albedo parametrisation serves as an example. Snow albedo is re-set to fresh snow albedo after each snowfall event greater than 3 mm w.e. d$^{-1}$. Thereby, also the albedo decay scheme starts again with the value for fresh snow. During field visits in summer, we observed the deposition of fresh snow on a longer exposed snow surface with a visibly reduced albedo. The fresh snow then melted away within hours to days again exposing the darker snow. The used parametrisation is not able to reproduce this evolution. This simplification may be compensated by the calibration of $t^*_{wet} =$
7 days, which is more suitable to represent conditions in Central Asia than the higher default value of 15 days used in the Arctic (Bougamont et al., 2005; van Pelt et al., 2019). As previously discussed in van Pelt and Kohler (2015), the model is not able to reproduce the small scale density variability of the subsurface (Fig. 13) as several processes such as wind crust formation, modelling of firn/snow grains or local vertical pooling of melt water on existing high density layers are not included.

Further model uncertainties are related to the choice of parameters. Thanks to the exceptional data availability for Abramov
glacier, several model parameters can be constrained. These data, however, are only available for point locations. We use spatially and temporally constant parameter values for periods with and without *in situ* measurements. This necessary simplification has implications. We use temporally constant albedo parameters, whereas a darkening of the glacier surface is likely (Sarangi et al., 2019). An underestimation of recent ablation rates (Fig. 7b) may be related to albedo decreases (Schmale et al., 2017) which are not considered due to a lack of respective calibration data.

Our study shows, that long-term *in situ* measurements are of great value for simulating the long-term evolution of Abramov glacier with a coupled surface energy balance–multilayer subsurface model. The comprehensive model output complements the exceptional observational data set for this glacier. The combination of both data sets provides an opportunity to discuss processes on a high spatial and temporal resolution for a period of more than five decades which is unprecedented for the data sparse HMA.



# 6 Conclusions

In this study, we apply a distributed coupled surface energy balance and firn model to simulate 52 years of mass balance and firn evolution of Abramov glacier, Pamir Alay. The model is forced with weather station and downscaled reanalysis data. The modelled surface mass balance as well as subsurface conditions agree well with *in situ* measurements for the beginning of the modelling period and recent years. We find an overall negative mass balance of -0.27 m w.e. a$^{-1}$ for 1968/1969-2019/2020, which is somewhat less negative than the mass balance determined by previous studies. The first modelled decade 1968/1969-1977/1978 is characterized by the most negative mass balance and is followed by two decades of almost balanced conditions. More recent years are again characterised by clearly negative conditions. Our results indicate a loss of pore space and reduction of internal accumulation on Abramov glacier. The correlation of annual mass balance with annual precipitation sums (R$^2$=0.72, p-value = 0.000) is stronger than the correlation with summer air temperature (R$^2$=0.29, p-value = 0.000). Increasing precipitation rates have thus compensated for increasing air temperatures, preventing an acceleration of mass loss for Abramov glacier during the last five decades. To our knowledge, this is the first application of a model of similar complexity for such a long time period for a glacier in HMA and may thus provide valuable insights into processes of this data scarce region.

*Code and data availability.* The EBFM code, the model forcing and topographical grid used in this study are available at https://doi.org/10.5281/zenodo.5773796 (Kronenberg et al., 2021b). Due to their large volume, modelled grids are available on request. The majority of in situ mass balance data used for calibration and validation are available from the World Glacier Monitoring Service (doi.org/10.5904/wgms-fog-2021-05) (WGMS, 2021). Additional mass balance data at monthly resolution and for recent years are also available on request. Measurements of the automatic weather station can be downloaded from http://178.217.169.232/sdss/index.php?&page=measure_page (Schöne et al, 2013).

*Author contributions.* MK performed the analysis and wrote the paper. WvP supplied the EBFM model code and support to use it. MK, HM and MH took place in several field campaigns acquiring *in situ* data. JF applied TopoSCALE to downscale the ERA5 data. FP provided the historical meteorological and mass balance data as well as extensive clarification on the data set. All authors participated in the discussion of the results.

*Competing interests.* The authors declare that they have no conflict of interest.

*Acknowledgements.* We acknowledge all the people who have been involved in the collection of field data on Abramov glacier. Especially we would like to thank to Erlan Azsivov, Ruslan Kenzhebaev Bolot Moldobekov and Ryskul Usubaliev from the Central Asian Institute for Applied Geosciences (CAIAG) and Sultanbek Belekov and Iurii Novomlintsev from KyrgyzHydromet for organizing the mass balance measurements on Abramov glacier during the last years and to Martina Barandun and Tomas Saks for their continuous efforts in coordinating



the monitoring activities. We greatly acknowledge the help of Stanislav Kutuzov, Ivan Lavrentiev, Alex Merkushkin, Yuri Tarasov, François Valla and Andrey Yakovlev for providing access to literature and data published in Russian and complementary information. We thank
Enrico Mattea for the valuable exchange about the model application and also for sharing his code used for modelling experiments. This study was financed by the Swiss National Science Foundation (SNSF), grant 200021_169453 and the Project CICADA (Cryospheric Climate Services for improved Adaptation, contract no. 81049674 between the Swiss Agency for Development and Cooperation and the University of Fribourg).



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
