# Peer review of "Long-term firn and mass balance modelling for Abramov glacier, Pamir Alay"

_The Cryosphere, 2021_

## Referee Comment (RC1)

Review to Long-term firn and mass balance modelling for Abramov glacier, Pamir Alay

Summary:

The authors apply an energy and mass balance model for firn and ice to a glacier in High-Mountain Asia (HMA) using the almost 50-year record of meteorological weather station data (AWS) together with down-scaled reanalysis data from ERA5. There is no significant trend in the annual mass balance found, though differences in space and time exist.

The manuscript is quite extensive and technical in the methods, for the main results as well as the abstract focusing on the modeling aspect. In particular, it describes the weather station data treatment quite extensively. These parts of the manuscript would benefit greatly from a shortening. The written part of the results is concise, but there are many figures which are not well included in the results or discussion.

The manuscript describes the measurements, measurement data handling, and treatment in an extensive fashion. Furthermore, throughout the manuscript, the value of this data for the model performance is emphasized. This should be either reduced or also stressed in the abstract and title.

General suggestions and comments :

Shorten and homogenize sections 2.3.1 and 2.3.2 with 2.4.1, which are quite lengthy in comparison to the TopoSCALE and bias correction section which is quite brief. It is also not clear to me, what is done with the precipitation forcing for the AWS time period. For the monthly averages used for the bias correction of the TopoSCALE data mentioned in 2.4.3, is this a monthly average for every January or is it bias corrected for each individual month of the time series? In case of the first, this statement is wrong "Monthly averages of the final cloud fraction time series correspond to observed values for the years for which measurements are available", if the latter, it is a strange way to bias correct, please clarify this.

Why is the cloud cover used and not incoming long-wave radiation?

Try to shorten 2.5.1 or maybe move the more detailed part to an appendix or supplement.

Some statements are very vague or numbers are missing, try to avoid rather/certain thresholds/relatively/etc.

Reconsidering your comment on the surface mass balance: The surface mass balance is the result of accumulation (+) and ablation (-) at the surface including precipitation (+), moisture exchange (+/-), mass loss through runoff (-) and refreezing above the previous summer surface (+).

What is refreezing? If it is surface melt water, it is not accumulation for the SMB, as the melted snow was above the previous summer surface too before it melts. If it is refreezing of rain then it falls to the category of precipitation. Please clarify this. For example with "including solid precipitation" and "refreezing of rainwater above the previous summer surface".

Section 3.5 is also quite extensive, try to shorten it, you already have most information in the table anyway

For p-values where your statistical analysis tool gave you less than the significant digits change it to "<0.001" instead of = 0.000.

In the results section there are a lot of figures: Check for each figure if it is referred to in the main text. Do you really need it is as part of the main manuscript, can even more of them be shown in the supplement or removed altogether? The correlation plots and one of 11 and 12 could be potential starting points. Furthermore, check your figure axes, in the case of shared y-axes readability may be harder than expected.

The uncertainty discussion and estimation did not quantify or investigate the influence of any assumptions like basing fresh snow albedo tuning only on the summer month, the bias correction approach on the input data, parameter choices, etc. What is the influence of the precipitation under catch correction? What is the influence of splitting the cloud cover differently over the day, not conforming o the daily average?

How does the correlation between measured and modeled SMB depend on the point and time? Figure 7 a/b does not show us if the model fails for certain time periods or certain point measurements. This could be further investigated. Additionally, basing the quantification of agreement on $R^2$ is tricky, as this is just about the correlation and not absolute errors, so systematic over- and underestimation are not accounted for. There are multiple ways how to compare measurements with models (Zolles et al. 2019). The choice of comparison method has a direct influence on the evaluation.

There were two different simulations conducted, as mentioned in the summary, this could be used more to emphasize the improvement that additional measurement data could provide.

During the entire discussion, the uncertainties are all given as the relation to the model and model forcing, though not quantified apart from one alternative run based on a shorter tuning period. The precipitation is here most likely the dominant factor due to uncertainty in climate model and measurements. In addition surface mass balance measurements are also uncertain, Zemp et al. 2013 mention that the related uncertainty of the field measurements at point locations is estimated to be 0.14 m w.e. a−1. What is the impact of this on the uncertainties? How does this change the confidence intervals?

Specific comments:

P2 L27: Wrong Hock reference: I guess: Hock 2005: Progress in Physical Geography 29, 3 (2005) pp. 362–391

P2 L29: acts → remove s

P2 L34-43: Mention the other studies first, then relate to Pamir Alay

P2 L48: Remove relatively

P2 L49: Change to "...mass fluxes over the period from YYYY – YYYY "

P2:L50: Delete "to our knowledge"

P3 L58: The mean annual …. add "The"

P4 L64-70: Could this be moved to the introduction, feels a bit out of place

P4 L87: Remove sentence starting with "Most recorded "

P4 L89: Could be misleading as you did correct for undercatch later?

P5 L98: We assign observed daily minimum cloud cover to the first four time steps and daily average cloud cover for the rest of the day. What is the impact of this assumption, did you test it, could you verify it? It does not conform to the daily average if 4 steps are lower and the rest is the average?

P5 L105: What is done for precipitation in this period?

P5 L106-109: Remove the entire paragraph

P5 L110: What does this interquartile range filter do? Is this physically reasonable to remove your so to speak outliers, even more so for the outliers that were not detected? The SMB is non-linear with regard

to the forcing, is the curve not smoothed this way and the SMB higher? Please clarify, investigate and add to the discussion.

P5 L117: Why 1500W/m²?

P6: Section 2.4.2 If you are using ERA5, why is the incoming long-wave radiation not used directly rather than using a cloud cover, which is then adjusted and strangely distributed over the day, and an empiric parametrization using $c_1$ and $c_2$ which are likely different for HMA as your reference used the model on Svalbard

P7 L155: section 2.5.1: Shorten, or put to appendix

P8 L228: Is the temperature of the snowfall considered when fresh mass is added to the snowpack? If rain's is not.

P8 L230: Remove $2^{nd}$ mention of "subsurface" in this line.

P8 L234: What would be the impact of the penetrating short wave radiation with quite thin layers? In addition, as it is mentioned a fresh snow layer in summer often melts extremely fast this might be even more relevant?

P11 L277: Quantify what the certain threshold is.

P11 L289ff: Check general comments on refreezing.

P12 L 315-331: Is this necessary as full text or is table 2 not enough?

P13 L344: See my general comment for how to compare measurements and model results, which objective did you use? Bias, MAD, RMSD, etc (Zolles et al. 2019)

P13 Table 1: What is the impact of calibrating fresh snow albedo only in summer?

P14 Table 2: The fresh snow density is huge if compared to what is measured. I have used the same value before, but did you try different values?

Page 17:

Figure 5: The left subplot is not readable, with your choice of colors for the different time periods, you cannot see the values if there is a larger area at a later time for previous times, this is clear for 4000-4300m and maybe at the top (could also be non-changing area there). This has to be changed.

The shared Y-axis may be a bit too far off from the other subplots, reduce the white space in between the panels or add the Y elevation axis to each.

Figure 6: As mentioned above, the different stakes/stake locations or time could be highlighted here, this has the possibility to show more information, else remove.

L 360: If that is the only sentence about figure 9, remove figure 9.

P 18: Figure 7 same as for figure 6

Figure 8: Maybe go for multiple colors. The red frame overlaps with baseline at 0, you hardly can see Qlat. Maybe do not make it a full rectangle but just up from zero but no overlap with x-axis/baseline.

P19 Table 4: p-values $0.000 \rightarrow <0.001$

P20 Figure 9: Shared Y-axis on the left panel not right, remove or supplement.

P23: L421: Mention your value at "an overall mass loss" so the comparison to the other studies works, this might also give the word "somewhat" in L423 a meaning, else remove it.

P24 L435: related $\rightarrow$ correlated

P24 L44ff: Is the unit for this not simple m w.e. m$^{-1}$

References:

Hock 2005 other paper: Progress in Physical Geography 29, 3 (2005) pp. 362–391

Zemp, M., Thibert, E., Huss, M., Stumm, D., Rolstad Denby, C., Nuth, C., Nussbaumer, S. U., Moholdt, G., Mercer, A., Mayer, C., Joerg, P. C., Jansson, P., Hynek, B., Fischer, A., Escher-Vetter, H., Elvehøy, H., and Andreassen, L. M.: Reanalysing glacier mass balance measurement series, The Cryosphere, 7, 1227–1245, https://doi.org/10.5194/tc-7-1227-2013, 2013.  a, b

Zolles, T.; Maussion, F.; Galos, S. P.; Gurgiser, W.; Nicholson, L. (2019): Robust uncertainty assessment of the spatio-temporal transferability of glacier mass and energy balance models. In: The Cryosphere 13, pp. 469 - 489

---

## Referee Comment (RC2)

**TC-2021-380-review**

Lindsey Nicholson
Niklas Richter

March 2022

**1 General comments**

Originality: The study makes use of key datasets at Abramov glacier to perform a novel modelling study to reconstruct 52 years of mass balance with concurrent process understanding. While this in itself is novel and the glacier is unusually rich in data for the region,a more explicit statement of the scientific motivation and purpose of the paper as well as critical discussion of how it might contribute to wider glacier process understanding in this region would be welcome. The contribution of modelled subsurface firn conditions is certainly valuable for improving density assumptions for geodetic mass balance estimates and for understanding the evolving potential internal accumulation capacity of the glacier over time.The paper concludes that the effects of warming on glacier mass balance are partially mitigated by increased accumulation, which may be relevant at a regional scale, and highlights that the buffering effect of internal refreezing is diminishing over time.

Scientific quality: The purpose of the paper could be more clearly stated with reference to what the key outputs are. For example saying that a process model allows you to build on the existing field firn study, and that the climate drivers of glacier mass change remain contested in this region, and can be investigated with a process-based model. We acknowledge that the parameter calibration is performed with rigorous manual calibration, yet by overlooking potential measurement uncertainties, as well as not using a multi-objective simulation/parameter evaluation this is not state of the art. Could you thus expand on the performance of the calibration? It would be good to justify your choice of sample number and location for the calibration datasets, and the treatment of uncertainties could benefit from recognition of a parameter equifinality issue (Rounce et al., 2020) in the optimisation. Rather than reporting only a single best-fit solution, incorporating a sensitivity analysis and its related uncertainties would be more robust. Indeed your comparison of the two sets of forcing data highlights the sensitivity of the modelled mass balance to the forcing data which demonstrates the value of a more comprehensive sensitivity study. This deserves even more emphasis given that the only difference between the two forcing data sets resides

in the timespan 1980-1998, the period for which you calculated the biases and should hence be closest to the observations.

Given the number of pre-processing steps in the forcing data and selection of model parameters, the impact of uncertainty in these choices is poorly quantified, making the model output and performance difficult to interpret, as we cannot be clear about the quality of the forcing data compared to reality. It may be helpful to show plots of the forcing data in the supplementary material as a start. This would also help addressing the difference between the two forcing data sets. Relating to the question of the overarching goal of this study: if the goal is to draw out the connection between the climate forcing and glacier response, investigations on the annual time series of mass balance alongside climatological properties - for example by adding climatological information into Table 3 or attempting to understand the causes of positive mass balance years in the timeseries - could be a starting point.

The timeframe of this study allows for an in-depth investigation on the drivers of glacier response also on the decadal scale, as continuously highlighted interdecadal differences in glacier response exist. Therefore, it could be worthwhile to investigate whether there exist any breakpoints in the climatological data that could explain the simulated decadal variations in the mass balance (Table 3).

Relevance: Key findings are that negative mass balance of -0.27m w.e./year persists for the period 1968/1969-2019/2020 alongside a loss of firn pore space causing a reduction of internal accumulation. Despite increasing air temperatures, no acceleration of glacier-wide mass loss was found over time as a result of increasing precipitation rates which appears to exert a strong control over annual mass balance at this glacier. This is of interest given that knowledge of precipitation in this region is quite poor and is thought to contribute to anomalous glacier behaviour in this region..

Presentation: (1) We suggest restructuring so that the model description comes before the description of the datasets - this will help guide the reader to know the constraints on the construction of the forcing data. If this move is made, we suggest headings as follows under a 'methods' section: study site, model description, forcing data, calibration data, validation data. (2) Please consider also the best way to order Sections 3.5 and 3.4 as they describe parameterisations applied before the application/analysis of the modelled results as well as bias corrections based on whole model simulations. (3) Please revisit the naming of your sites: you have sites 1 and 2 in the accumulation zone, but it would be helpful to call your ablation zone site, site 3 to be consistent in the naming convention and also to show its location on the maps. Relatedly, please can you add a statement as to how representative these chosen sites are for mean ablation zone/accumulation zone conditions. (4) Please show which stake locations/data are used for calibration cf which for validation in one of the map figures as its currently unclear if the locations for calibration are identical to the locations of validation but just a different subset of the data. (5) As

currently presented the figures and tables are often far from the associated text which could be improved for the final layout, some suggested changes to figures are included in the specific comments, and some English edits suggested in an annotated pdf.

**2  Specific comments**

L13: Is this heterogenous behaviour really only in the last decade?

L18: Is this increase progressive, or stepwise, and what is its magnitude (to help readers not so familiar with the region)?

L25: It may be valuable to specify that the advantage of a physically based model is that it is expected to be more suitable for projecting processes into unknown climates, compared to a temperature index model calibrated for one period in time applied within a non-stationary climate projection. The additional caution of course is that many physically based models include stationary parameterizations.

L28: Please specify which processes and/or which models include/exclude them, or maybe better still just delete the sentence starting "Important ..." as it's really not needed here

L34: Suggest "In the data poor HMA, ..." or maybe better still just delete the sentence starting "In the ..." as it's really not needed given the following sentence

L44: Suggest restructuring of this closing paragraph to more explicitly state some goals of the study. First state the problem/motivation (e.g. why do we care about this glacier/timeseries/region/... need for a continuous record in this poorly understood and anomalously behaving part of the mountain cryopshere?) and how doing this modelling study delivers a novel and useful solution (e.g. to evaluate decadal trends in surface energy balance fluxes, firn evolution ... and their connection to the forcing climate conditions, attempt more process understanding of previously suggested claims?)

L48: Delete "These in situ data, which are unique for the region, allow us to apply a rather complex model with relatively high data requirements."

L50: Delete the last sentence.

L64-69: Give time period of Barundun long term study; and state the long term mass balance of both studies.

L69: Where is this point site on the glacier?

L70: You mention a regime shift in the 1970s - please return to this point in the discussion and in the evidence from your modelled study, does your model show this increase, what are its effects on the glacier scale mass balance and firn development?

L76: Careful how you use this term reference glacier surface - it has a specific meaning as explained in Huss 2012, but I am not sure you mean this.

L78: The creation of the 1968 DEM needs more clarification on how you did it and how reasonable it is. Why do you use only 2 DEMs and not intervening satellite DEMs (e.g. SRMT, ASTER), to examine the trend from timestep to timestep? Would it be possible to use coarser, but more frequent dDEMs to check the reasonableness of the linear backwards extrapolation of surface height change over time, e.g. the ones mentioned from Barandun et al. (2015)?

L83: Was this weather station manually monitored? Is the non-digitized data a good candidate for https://www.zooniverse.org/projects/edh/weather-rescue/

L85: Suggest rephrasing to "Here, we use data from January 1968 until December 1998 for the following parameters: daily average air pressure, windspeed, relative humidity, temperature and cloud cover, as well as daily precipitation sum, daily minimum temperature and cloud cover and daily maximum temperature."

L89: Can you say something about expected undercatch based on any other measurements in the region or on the likely mountain undercatch for the device used? To my knowledge snow undercatch can be ¿50 percent in mountain regions depending on the sensor so it would be good to mention here.

L91: Why 3 hour timesteps, when the ERA5 data is available hourly? If this is because the model needs this time resolution add this to the model description which should preceed this section. Did you consider better ways of reconstructing the daily cycle e.g. by using the daily cycle of the ERA variables? For example it seems inconsistent that you introduce a daily cycle for cloud but then not for precipitation ... these choices need to be justified and their implications considered in the discussion section.

Section 2.3.2: It seems that you don't actually make much use of the recent weather station - is that correct? Just used in the pressure parameterisation and the radiation parameterization? a. Consider mentioning explicitly the use of each dataset in the analysis.

L107: Need some more explanation of the occurrence of missing data, what percentage is typically missing and what happens to the NaN data periods?

L110: What do you mean by an IQ range filter? Can you specify the scale of this here please? With this and the subsequent double application of the std filter it sounds like you substantially reduce your dataset variability range in a more or less arbitrary fashion before you begin with any physically based QC? Can you explain why?

L116: Other publications show the way for some more rigorous quality control e.g. could correct for snow over SW sensor, using the albedo readings, and for freezing of station using non-variant wind direction, and for RH¿100 it is commonly reset to 100 according to many standard instrument manuals (e.g. Campbell scientific). RH values above 100% are usually set to 100%. Also you should report how many/what percentage of the total readings are filtered out (by each of the filtering steps)? Potentially include the whole quality controlled data series in the supplement.

L122: Could you please just mention why you did not use the extended ERA5 1950-1979 dataset. It might be still preliminary, but did you explore using it at all?

L124: Possibly expand on what TopoSCALE does - you input upper air field and a DEM, and get out radiation modifications based on the DEM, which all other fields remain the same for the given pressure level? For readers unfamiliar with this product it would be worth explaining also why you want to do this to your data first. As a side note, how does TopoSCALE compare to TopoCLIM now available? https://gmd.copernicus.org/articles/15/1753/2022/ Can you also show the ERA5 gridcell location on your map please?

L128: Its not clear why you need cloud cover fraction but presumably for the SEB model, so perhaps state this at the start of the section, or better still put the model description before the data description so that its clear to the reader what is needed to force the model. Does the clear sky value come from ERA5 + TopoSCALE? Why usage of cloud cover fraction instead of readily available longwave radiation from ERA5?

L129: You may want to add units for completeness here.

L135: Are the parameter values developed for a particular site, or globally valid?

L140: "Even after downscaling (accounting for resolution diff.)..." here are you referring to what TopoSCALE does, without an explanation of what TopoSCALE is it is not clear to the uninitiated reader at this stage that it is a downscaling scheme, by which I understand it changes the resolution of the reanalysis data?.

L142: Explain why you aggregate to monthly for the bias correction when your data are available at different timescales, these steps seem confusing. Furthermore, you assume that biases 1980-1998 are the same as in recent decades, is

that likely? Why didn't you for example check the other station data to correct biases. Is the height difference between the two weather station locations handled by TopoSCALE?

L147: After correction using monthly data, cloud cover fraction in summer showed bad fit, therefore set cloud cover to 0 for days without precipitation and to 1 if precipitation is above threshold, this seems flawed as cloudy days could occur without precipitation, would it not be better to correct this with longwave from ERA5 data? Or at least explain and justify these seemingly arbitrary choices - perhaps it is appropriate for this site, but please make a case for this here, using data or citations.

L150: So what is the timescale of these two forcing datasets? Original ERA5 data is hourly, ERA5 TopoSCALE bias corrected data is monthly, and is this dataset 3 hourly. Here please state the purpose of this duplicate dataset ... what do you gain from running the model with two datasets. I think it tells you something about the sensitivity of your results to the forcing data, but please explain this clearly here, and what it tests and does not test..

L150: Can you also show a comparison of the overlapping period of these two datasets, alongside the available weather station data so we can see how well they duplicate each other over periods of overlap - this is important to convince the reader of the homogeneity and robustness of the forcing data derived in this way. Probably fine in the supplement.

L165: Density assumptions including a constant elevation threshold, could you please justify this, especially as your abstract implies the firn is changing density, and presumably changing in spatial extent as well? You highlight in the text that this is a major source of error so it might be nice to consider simple uncertainty analyses based around output sensitivity to varying these assumptions. Also why is the firn line 4200m a.s.l., is this based on satellite data or the stake data, please justify this choice in the text.

L174: Please add in the text that March is selected as the end of the winter seasons before the snowpack is modified by melt and that September is the end of the hydrolological year at this site, or otherwise justify these with your measured data, or citations.

L175: What is this selection of 19 stakes for - later it becomes clear this is for calibration, but check this whole paragraph for what you really want to explain.

Section 3: Moving this section to before the data description would make it much clearer what forcing data is actually needed to drive the model and therefore make it easier to understand the data preparations steps.

Figure 2: It would be useful to refer to this figure more in the data description - e.g. when discussing the two forcing datasets, and also maybe include the borehole temperature data on it?

L197: Does the Svalbard development location mean its particularly suitable for this glacier site? Better than alternative models for example? Certainly refreezing is important in Svalbard glaciers, right?

L201: Could you include a comment on the model performance from these studies - did it do well in comparison to others?

L206: State here which meteorological forcing data are required.

L212: Indication on connotation of fluxes (positive, negative) with respect to surface.

L214: How does the transition occur? Binary or gradual?

L216-208: This modelling of radiation seems odd as I thought this was the point of the TopoSCALE downscaling to give relevant radiative properties for a surface? Please start this section with the required forcing data for this model, and why you chose to not force it with the ERA5 incident radiative fluxes? Are cloud parameters for Svalbard (sea level and maritime) appropriate for Abramov? Can you quantify the better agreement and also state which weather station data was used here?

L223: Longwave radiation is computed in ERA5, why not use it to begin with?

L254: Liquid water instantly distributed following normal distribution, is this a widely used approach and do you know how it behaves in reality? Clearly this is a tricky parameter to know, but are there implications of this assumption?

L280: Does this initial subsurface condition differ greatly from just running the first year multiple times - would this not be a more precise way to evolve the starting conditions rather than a 20 year run? Either way its not known if this has a substantial impact on the results. Were there early borehole temperatures as well?

L283: So 'adjust' here means optimisation of parameters in some way described later on .. are these point scale optimizations and how does that relate to the fact that the glacier outline is updated over time - are all optimization points within the final glacier boundary?

L303: In the comparisons does the SMB include refreezing (so the modelled climatic mass balance cf Cogley et al., 2011) or not? Looks like it from L347, but maybe clarify here?

Table 2: Those parameters identified as optimized in this table are only those optimized through a whole model simulation. This seems a bit misleading as others are also optimized but to other data externally to a full model simulation, perhaps instead just separate literature/optimized values.

L317: Clarify pressure decay from historic average and modern average (October 2011 - 2020) as you mention the modern timeseries was only used from 2014 onwards due to data gaps (L102).

L336: This precipitation correction factor is performed within a SEB model simulation like the albedo optimization or is it simply applied to the precipitation timeseries compared to the glaciological data? If the latter doesn't it belong in the forcing data preparation section? Here do you compare the March accumulation in the monthly snowpits to the March accumulation int he forcing data - if so this could be stated more clearly. Maybe also justify the choice of summer bias correction (1.15) from a single different site, is this representative of a range of sites for this instrument?

L345: "Overall, the mass balance of Abramov glacier is negative for the years from 1968/1969 to 2019/2020." - what is meant here? The mean over the whole period? Give a value or consider giving the cumulative mass balance numerically (which could also be included in Figure for as a line plot on a secondary axis).

L347: Consider how to refer to the results - e.g. rather than listing what is shown in each figure you could focus on the description of what is revealed and cite the figure: e.g. The distributed mean annual mass balance for 1968/1969-2019/2020 is shown in Fig. 3. The mass balance of Abramov glacier is predominantly negative for the years from 1968/1969 to 2019/2020, despite part of this being taken up by glacier retreat, and shows no significant trend in annual mass balances (+0.0002mw.e. a1, p-value=0.979). The most negative modelled mass balances occur at the start of the period, while the two decades between 1978 and 1998 are characterised by almost balanced mass budget, thereafter becoming more negative again (Figure 4). The modelled mass balance gradient shows that accumulation is lowest during the first decade (1968/1969-1977/1978) and highest during the last modelled decade (2008/2009-2017/2018), and ablation is largest during the first decade, followed by the second last decade (1998/1999-2007/2008).

Figure 3: Please highlight the different glacier extents with dates or emphasize the fact that the average over the whole timespan with the adjustment of the glacier area leads to the distinct pattern. It might also be nice to report a long term mean ELA in this or the gradient figure (5b).

Figure 4: It woulds be nice to show the cumulative modelled mass balance

on a secondary axis.

L357: Again here, avoid just listing what is in the figures, but rather tell us what the figures *show*. The title makes it seems like this is part of the model validation but in fact here you also show the model calibration results, right? Suggest moving the calibration part to the end of section 3.5. (e.g. The model calibration was performed on eight snow pits locations at the end of March and annual mass balance measurements from 19 stakes and up to eight snow pit locations for the period 1968/1969-1997/1998; the comparison of optimized model simulations to these in-situ observations used for model calibration shows a stronger RMSE for the annual mass balance, and despite some bias in the linear fit, the datasets generally approach the 1:1 linear fit line (Figure 6).) Then here in this section on validation say: e.g. For model validation we compare the simulated mass balance to a set of point measurements not used in the calibration, which can therefore provide an independent measure of model performance. This comparison is based on 146 stake readings not used in model calibration for 1968/1969-1997/1998 (Figure 7a) and all available point mass balance measurements for 2011/2012-2019/2020 (Figure 7b). In both cases the comparison scatter plots cluster around the unity fit line, but inherit the bias tendencies seen in the calibration (Figure 6)) Consider colouring the points in Figure 6 by decade or year as then you might also reveal periods of better/worse model calibration, which could be interesting.

Figure 6: The different axis labelling in (b) might create a false impression, please homogenize them

Figure 7: So the performance isn't any better in the measurement period (a) compared to downscaled ERA5 period (b)? This could also be rephrased to the fact that both periods show good agreement, but given the impact of the two different forcing data sets, I'm a bit hesitant to jump to that conclusion. Further: Could you maybe indicate the number of observations here just to be clear. The scatterplot also doesn't give us an indication of the point or location of the measurements. Generally, I would have preferred to clearly see the points on a map maybe with a highlighting of the calibration sites and also evaluate how you chose the calibration measurement. For the ablation stakes you use less than 10% to calibrate - where does this number come from?

Figure 8: Depending on the direction of the analysis, the decadal analysis could be further complemented with monthly or seasonal plots.

Table 4: Here annual trends, but earlier figure showed decadal MB, would be interesting to see if there are certain breakpoints in the timeseries corresponding to external forcing (disturbances in atmospheric forcing etc.)?

L371: Interesting that you have decreasing RH but increasing precipitation? Does need explanation?

L396: the correspondence in Fig 13 seems really good to me, perhaps comment in line here that these biases are accompanying a generally good agreement, as evidenced in the figure? Is this match in line with expectations from modelled densification?

L403: Do you return to explain this cooling trend in the discussion?

Figure 11 and 12 could be a bit larger.

L421: Can you add the value you find in the text here to facilitate comparison with the other studies for which you do quote the values.

Figure 15: Its cool to see these data which have not been widely included in previous Abramov studies. What is the added value of this study compared to the others, how do the different studies compare to each other? (Please check the data of the Barandun paper - its labelled 2015 in legend and 2021 in caption). This should be mentioned in the discussion. For example, due to the updating of the glacier extent and glacier elevation, the results of this study are more positive compared to the assumption of static glacier extents in the other studies? You could also include your model simulation results from the recent period and the more recent mass balances studies to complete the picture (e.g. given in Table 6 Barandun et al., 2018 and Barandun et al., 2021 Zenodo data set). This would also open the possibility to compare your modelled results to geodetic mass balances for the glacier available e.g. Hugonnet et al, 2021 and/or Miles et al., 2021 which includes an ASTER based elevation change rate based on results of Brun et al., 2017 (https://doi.org/10.5281/zenodo.3843292).

L472: Move these first 2 sentences to the conclusions, as they seem a nice summary of what you have already said in this discussion section.

L485: This statement of performance is related to the validation against point measurements being better for the main run than the alternative run? Where do we see this in the results shown? Or is it just in the supplement. Suggest moving this statement to the part on uncertainty due to forcing data as it deserves more full discussion.

L490: Ideally this section would provide a formal uncertainty assessment. In your situation, as there are many unquantified errors it would be good to include a sensitivity analysis. If this is at all feasible, we strongly encourage you to add this, otherwise please justify why not. You can then describe your approach at end of this paragraph: " As the sources of uncertainty are diverse and sometimes not readily quantifiable, we here discuss the general sources and likely implications, rather than producing a formal uncertainty assessment for the modelled output."

L512: What is the implication of these large differences? Does that mean that the more recent timeframe is wrongly estimated? How can this happen if the bias correction is also based on this period? To get an overview over this, plots of the different forcing datasets (at least in Supplement) could help.

L513: Where have you shown that the standard run is 'better' than the alternative run? Can you here add some quantification on the goodness of fit of the main and alternative run?For example in Figure S5 it would be good to see a comparisons with the actual measured MB for this period and report on which forcing data version matches most closely This is of major importance as, if the alternative forcing produces a very poor match, then it calls into question the validity of the simulated mass balance from 1998 onwards, right? Also consider if this is not better in the main paper rather than the supplement?

**Correspondence:** Marlene Kronenberg (marlene.kronenberg@unifr.ch)

**Abstract.** Several regional studies identified heterogeneous mass changes in western High Mountain Asia over the last decade. Causes for these mass change patterns are still not fully understood. Modelling the physical interactions between glacier surface and atmosphere over several decades can provide insight into relevant processes. Such model applications, however, have data needs which are usually not met in these data scarce regions. Unique glaciological and meteorological data exist for the
5  Abramov glacier in the Pamir Alay range. In this study, we use weather station measurements in combination with downscaled reanalysis data to force a coupled surface energy balance–multilayer subsurface model for Abramov glacier for 52 years. Available *in situ* data are used for model calibration and validation. We find an overall negative mass balance of -0.27 m w.e. a$^{-1}$ for 1968/1969-2019/2020 and a loss of firn pore space causing a reduction of internal accumulation. Despite increasing air temperatures, we do not find an acceleration of glacier-wide mass loss over time. Such an acceleration is compensated by
10  increasing precipitation rates (+0.0022 m w.e. a$^{-1}$, significant at a 90% confidence level). Our results indicate a significant correlation between annual mass balance and precipitation (R$^2$=0.72).

**1 Introduction**

Spatially heterogeneous mass changes of glaciers in High Mountain Asia (HMA) during the last decade have been detected by several regional studies (e.g. Kääb et al., 2012; Brun et al., 2017; Shean et al., 2020; Jakob et al., 2021). Topographical effects in
15  combination with precipitation increases are suggested as reasons for balanced or positive mass changes for numerous glaciers in the Karakoram, Kunlun Shan, Pamir, Pamir Alay, and Tibetan Plateau subregions (Miles et al., 2021). Whereas reliable precipitation data from *in situ* measurements are very scarce for the region (Pohl et al., 2015), the analysis of the gridded Global Precipitation Climatology Project over thirty years has indicated a precipitation increase in the Western part of HMA due to large-scale atmospheric circulation patterns (Yao et al., 2012). Based on regional climate model data, glacier modelling
20  and moisture tracking, De Kok et al. (2020) conclude that changes in irrigation patterns and climate are responsible for the identified mass balance patterns in HMA.

[Figure]

Including *in situ* data and investigating processes at a local scale over several decades can be helpful in better understanding the influence of atmospheric conditions on glacier mass changes (Mölg et al., 2012; Zhu et al., 2020). Mass balance models of varying complexity have been applied to investigate the mass balance response of mountain glaciers to climate (e.g. Klok

25    and Oerlemans, 2002; Pellicciotti et al., 2009; Sicart et al., 2011). Models solving the energy balance at the glacier surface are more physically based and therefore considered more suitable for longer time periods than temperature-index parametrisations (Hock and Holmgren, 2005). Energy balance models are, however, only applicable if sufficient data are available to generate a complete climate forcing and to calibrate uncertain model parameters. Important processes in the accumulation zone, which acts as a buffer against mass loss due to refreezing and water storage, are not included into surface energy balance models.

30    Several studies have applied energy balance models coupled to multi-layer snow models to simulate refreezing processes within the snow and firn as well as heat conduction which are relevant for the glacier mass and energy balance (e.g. Reijmer and Hock, 2008; Huintjes et al., 2015b). Simulating the physical connection between the atmosphere and the glacier provides insights into the climatic control of glacier mass gain or loss (Mölg and Hardy, 2004).

35     The availability of historical data for the Abramov glacier located in the Pamir Alay provides a unique opportunity for detailed modelling over longer periods than previous studies did for HMA glaciers: Kayastha et al. (1999) applied a point-scale energy balance model to Glacier AXOIO in the Nepalese Himalaya. Azam et al. (2014) modelled the point energy balance of Chhota Shigri Glacier glacier in the Western Himalaya. Several studies focus on glaciers and ice caps located on the Tibetan Plateau (Mölg et al.,

40    2012; Zhang et al., 2013; Huintjes et al., 2015b, a, 2016). Zhu et al. (2020) applied a surface energy balance model to Muji Glacier, located in the north-eastern Pamir. Except for Huintjes et al. (2016), who used a coupled snowpack and ice surface energy and mass balance model (COSIMA) to reconstruct the climate on the Tibetan Plateau during the little ice age, only relative short periods up to one decade have been investigated.

Here, we apply a coupled surface energy balance – multilayer subsurface model (van Pelt et al., 2012, 2019) for a glacier

45    located in the western part of HMA. We simulate 52 years of firn and mass balance evolution of the Abramov glacier which is located in the Pamir Alay region, Kyrgyzstan (Fig. 1). For this temperate valley-type glacier, detailed glaciological as well as meteorological measurements exist (Kislov, 1982; Pertziger, 1996; Schöne et al., 2013; Hoelzle et al., 2017; Kronenberg et al., 2021a).  The overall aim of this study is to model the energy and mass fluxes for a period of five decades for Abramov

50    glacier in order to better understand its firn and mass balance evolution as a response to climatic conditions.

[Figure]

[Figure]

**(a)**
Color tones visualize glaciation of each region (outlines from
RGI 6.0, RGI Consortium, 2017)
Background layer sources: Esri, USGS, NOAA

**(b)**

◻ glacier outlines 2015 and 1975 (dashed)
⊕ old weather station
⊕ automatic weather station (AWS)
⊕ thermistor chain
◉ firn cores

◇ snow pits (1968-1998)
○ ablation stakes (1968-1998); bigger symbols for selected stakes
◆ snow pits since 2012; smaller symbols for not repeated investigations
● ablation stakes since 2011
◻ stake at ~3850 m a.s.l.

WGS84 / UTM Zone 42N
Background image Pléiades 01.09.2015 (E. Berthier); Contours from DEM2015 (Denzinger et al., 2021)

[revised manuscript text omitted]

---

## Author Comment (AC1)

**Response to reviewer 1 (Tobias Zolles)**

**Long-term firn and mass balance modelling for Abramov glacier, Pamir Alay**

Marlene Kronenberg, Ward van Pelt, Horst Machguth, Joel Fiddes, Martin Hoelzle, Felix Pertziger

Dear Reviewer,

We would like to thank you for your attention to our manuscript. We appreciate the interest in our study and thank for your constructive review and suggestions how to improve the quality of the paper. Below, we respond point by point to all comments, and state how we plan to account for them in a revised version of the paper. The responses (normal font style) to the reviewer's comments are written directly into the reviews (displayed in italic font style). Revised/additional figures can be found at the end of this response letter and are labeled with Roman numerals to them from figures in the manuscript.

Marlene Kronenberg,

*Fribourg, May 24, 2022*

**1    Summary**

*The authors apply an energy and mass balance model for firn and ice to a glacier in High-Mountain Asia (HMA) using the almost 50-year record of meteorological weather station data (AWS) together with down-scaled reanalysis data from ERA5. There is no significant trend in the annual mass balance found, though differences in space and time exist.*

*The manuscript is quite extensive and technical in the methods, for the main results as well as the abstract focusing on the modeling aspect. In particular, it describes the weather station data treatment quite extensively. These parts of the manuscript would benefit greatly from a shortening. The written part of the results is concise, but there are many figures which are not well included in the results or discussion.*

*The manuscript describes the measurements, measurement data handling, and treatment in an extensive fashion. Furthermore, throughout the manuscript, the value of this data for the model performance is emphasized. This should be either reduced or also stressed in the abstract and title.*

We agree that the manuscript is rather extensive and technical and thank the reviewer for the suggestions of shortening. The data set available for the site is exceptional for this region and allowed for the application of a model of such complexity. This will be further stressed in the abstract. We consider it difficult to capture everything within the title. A slight modification to 'Long-term firn and mass balance modelling for Abramov glacier in the data-scarce Pamir Alay' is suggested.

————

**2    General suggestions and comments**

*Shorten and homogenize sections 2.3.1 and 2.3.2 with 2.4.1, which are quite lengthy in comparison to the TopoSCALE and bias correction section which is quite brief. It is also not clear to me, what is done with the precipitation forcing for the AWS time period. For the monthly averages used for the bias correction of the TopoSCALE data mentioned in 2.4.3, is this a monthly average for every January or is it bias corrected for each individual month of the time series? In case of the first, this statement is wrong "Monthly averages of the final cloud fraction time series correspond to observed values for the years for which measurements are available", if the latter, it is a strange way to bias correct, please clarify this.*

We will shorten the sections about station data in the main manuscript and provide additional information in the supplementary material. There is no precipitation data available from the AWS (2014-2020) and TopoSCALE ERA5 data is used as a model forcing. We will more clearly state which data set is used for what (as suggested by Reviewer II). The correction was done for a monthly average of every January etc. Indeed, the statement about the cloud time series is wrong (should be "Monthly averages of the final cloud fraction time series correspond to the monthly averages prior to correction"). Will be corrected.

————

*Why is the cloud cover used and not incoming long-wave radiation?*

We use cloud cover for several reasons: (i) The EBFM parametrisations use cloud cover to compute incoming shortwave and longwave radiation. Cloud cover is thus a necessary model

input. (ii) There are no long-wave incoming radiation measurements available from the original station and thus no such data available for the first years of the study period. (iii) The AWS longwave radiation is affected by a nearby located rock face and there are substantial data gaps. (iv) The TopoSCALE ERA5 long-wave data can thus not be validated/corrected (which would definitely be necessary given the bias corrections needed for the other TopoSCALE ERA5 variables).

———

*Try to shorten 2.5.1 or maybe move the more detailed part to an appendix or supplement.*

We will shorten and move details to the supplement.

———

*Some statements are very vague or numbers are missing, try to avoid rather/certain thresholds/relatively/etc.*

We will use clearer language and provide numbers.

———

*Reconsidering your comment on the surface mass balance: The surface mass balance is the result of accumulation (+) and ablation (-) at the surface including precipitation (+), moisture exchange (+/-), mass loss through runoff (-) and refreezing above the previous summer surface (+). What is refreezing? If it is surface melt water, it is not accumulation for the SMB, as the melted snow was above the previous summer surface too before it melts. If it is refreezing of rain then it falls to the category of precipitation. Please clarify this. For example with "including solid precipitation" and "refreezing of rainwater above the previous summer surface".*

It is mainly snow melt water (precipitation is predominantly solid) which is refreezing within the seasonal snow pack. Indeed, the statement was misleading, as melt does not automatically mean mass loss in the EBFM. Will be corrected.

———

*Section 3.5 is also quite extensive, try to shorten it, you already have most information in the table anyway*

Agreed. We will shorten this description.

———

*For p-values where your statistical analysis tool gave you less than the significant digits change it to "<0.001" instead of = 0.000.*

ok.

———

*In the results section there are a lot of figures: Check for each figure if it is referred to in the main text. Do you really need it is as part of the main manuscript, can even more of them be shown in the supplement or removed altogether? The correlation plots and one of 11 and 12 could be potential starting points. Furthermore, check your figure axes, in the case of shared y-axes readability may be harder than expected.*

We will move figures to the appendix and more often refer to the remaining ones in the text.

———

*The uncertainty discussion and estimation did not quantify or investigate the influence of any assumptions like basing fresh snow albedo tuning only on the summer month, the bias correction*

*approach on the input data, parameter choices, etc. What is the influence of the precipitation under catch correction? What is the influence of splitting the cloud cover differently over the day, not conforming o the daily average?*

We will provide a comprehensive sensitivity discussion including the quantification of parameter and forcing (also based on the two different runs) sensitivities. We perform model runs for selected grid points using perturbed parameters and show corresponding results in the appendix (cf. Figs. I and II). Additionally, we will present the sensitivities regarding different cloud cover forcings (please note that there is no average cloud cover data available - average in line 87 was wrong, should be max.). See also answer to specific comment regarding cloud cover below.

————

*How does the correlation between measured and modeled SMB depend on the point and time? Figure 7 a/b does not show us if the model fails for certain time periods or certain point measurements. This could be further investigated. Additionally, basing the quantification of agreement on $R^2$ is tricky, as this is just about the correlation and not absolute errors, so systematic over- and underestimation are not accounted for. There are multiple ways how to compare measurements with models (Zolles et al. 2019). The choice of comparison method has a direct influence on the evaluation.*

We will add a visualization for different periods in the scatter plots and show the annual average bias together with the annual mass balance (current figure 4). $R^2$ was not used for optimization. We aimed on reducing the bias while keeping an eye on the RMS and regression line to omit compensating effect (e.g. a compensation by an underestimation of melt by an underestimated accumulation).

————

*There were two different simulations conducted, as mentioned in the summary, this could be used more to emphasize the improvement that additional measurement data could provide.*

The differences between both simulations will be more comprehensively discussed and pointing out that measurements are necessary to evaluate the model performance and that additional data would be an asset.

————

*During the entire discussion, the uncertainties are all given as the relation to the model and model forcing, though not quantified apart from one alternative run based on a shorter tuning period. The precipitation is here most likely the dominant factor due to uncertainty in climate model and measurements. In addition surface mass balance measurements are also uncertain, Zemp et al. 2013 mention that the related uncertainty of the field measurements at point locations is estimated to be 0.14 m w.e. $a^{-1}$. What is the impact of this on the uncertainties? How does this change the confidence intervals?*

We will add a comprehensive sensitivity analysis which will also allow to quantify the influence of different parameters. Model runs for perturbed parameters were preformed following and expanding the analysis presented for Morteratsch glacier by Klok and Oerlemans (2002). In addition, we will estimate the uncertainties of point measurements based on Thibert et al. (2008) and show them together with an annual average misfit and annual modelled mass balance (current figure 4) for a better estimate of uncertainties. Please note that the parameters were not tuned for the alternative run (will be clarified).

————

**3   General suggestions and comments**

*P2 L27: Wrong Hock reference: I guess: Hock 2005: Progress in Physical Geography 29, 3 (2005) pp.362–391*

Will be corrected.

————

*P2 L29: acts→removes*

ok.

————

*P2 L34-43: Mention the other studies first, then relate to Pamir Alay*

ok.

————

*P2 L48: Remove relatively*

ok.

————

*P2 L49: Change to "…mass fluxes over the period from YYYY – YYYY "*

ok.

————

*P2:L50: Delete "to our knowledge"*

ok.

————

*P3 L58: The mean annual …. add "The"*

ok.

————

*P4 L64-70: Could this be moved to the introduction, feels a bit out of place*

Will be moved.

————

*P4 L87: Remove sentence starting with "Most recorded "*

ok.

————

*P4 L89: Could be misleading as you did correct for undercatch later?*

Agreed. Will be clarified.

————

*P5 L98: We assign observed daily minimum cloud cover to the first four time steps and daily average cloud cover for the rest of the day. What is the impact of this assumption, did you test it, could you verify it? It does not conform to the daily average if 4 steps are lower and the rest is the average?*

We will present mass balance sensitivity regarding the use of different cloud cover forcing for the morning and the afternoon. Please note that no 'average' cloud cover data is available as only a low and a high (not average) cloud cover values are reported per day reported. (The 'average' in L87 is actually wrong - will be corrected).

————

*P5 L105: What is done for precipitation in this period?*

As for the other climate variables, We use debiased TopoSCALE ERA5 data as model forcing. The use of data will be clarified.

————

*P5 L106-109: Remove the entire paragraph*

ok. Will move detailed description to supplement.

————

*P5 L110: What does this interquartile range filter do? Is this physically reasonable to remove your so to speak outliers, even more so for the outliers that were not detected? The SMB is non-linear with regard to the forcing, is the curve not smoothed this way and the SMB higher? Please clarify, investigate and add to the discussion.*

Upper and lower fences were calculated using the upper ($q_{75}$, median between 50th percentile and uppermost extreme) and lower quartiles ($q_{25}$, median between 50th percentile and lowermost extreme of the data set). (upper fence: $q_{75} + 3 \times q_{75}$-$q_{25}$). The filter was used to identify extreme outliers of the entire data data set (likely caused by malfunctions). More local outliers were than removed using a moving mean method. See also Wilks (2011).

As the thereby corrected AWS data was not used to force the model, there is no consequence on the SMB by this approach. As the AWS data was only marginally used, the whole pre-processing is not that relevant for the study and will be substantially shortened in the main manuscript.

————

*P5 L117: Why 1500W/m²?*

Values above appear as outliers when visually analyzing the instantaneous time series.

————

*P6: Section 2.4.2 If you are using ERA5, why is the incoming long-wave radiation not used directly rather than using a cloud cover, which is then adjusted and strangely distributed over the day, and an empiric parametrization using c1 and c2 which are likely different for HMA as your reference used the model on Svalbard*

See answer to general remark (second point) above.

————

*P7 L155: section 2.5.1: Shorten, or put to appendix*

ok.

————

*P8 L228: Is the temperature of the snowfall considered when fresh mass is added to the snowpack? If rain's is not.*

no.

————

*P8 L230: Remove 2nd mention of "subsurface" in this line.*

ok.

————

*P8 L234: What would be the impact of the penetrating short wave radiation with quite thin layers? In addition, as it is mentioned a fresh snow layer in summer often melts extremely fast this might be even more relevant?*

No data is available for this or comparable sites. The absorption can affect the snow temperature (Munneke et al., 2009). Results from Dalum et al. (2021) also highlight that this is a relevant process with impacts on the surface mass balance and refreezing of different areas on Greenland ice sheet.

————

P11 L277: Quantify what the certain threshold is.

ok.

————

P11 L289ff: Check general comments on refreezing.

ok. Will be clarified.

————

P12 L 315-331: Is this necessary as full text or is table 2 not enough?

Will be shortened.

————

P13 L344: See my general comment for how to compare measurements and model results, which objective did you use? Bias, MAD, RMSD, etc (Zolles et al. 2019)

We mainly aimed on reducing the bias (see also answer above).

————

*P13 Table 1: What is the impact of calibrating fresh snow albedo only in summer?*

The fresh snow albedo is mainly relevant during the melt season. It was calibrated with measurements from the early melt season. The sensitivity regarding perturbations of the fresh snow albedo will be discussed.

————

*P14 Table 2: The fresh snow density is huge if compared to what is measured. I have used the same value before, but did you try different values?*

No, we haven't. This would indeed be interesting. However, the current value produces reasonable subsurface densities as visible from figure 13.

————

*Page 17: Figure 5: The left subplot is not readable, with your choice of colors for the different time periods, you cannot see the values if there is a larger area at a later time for previous times, this is clear for 4000-4300m and maybe at the top (could also be non-changing area there). This has to be changed. The shared Y-axis may be a bit too far off from the other subplots, reduce the white space in between the panels or add the Y elevation axis to each.*

Colors in plot a will be improved and axes references will be added.

—————

*Figure 6: As mentioned above, the different stakes/stake locations or time could be highlighted here, this has the possibility to show more information, else remove.*

Will add more information and move the figure to the supplement.

—————

*L 360: If that is the only sentence about figure 9, remove figure 9.*

Will move the figure to the supplement.

—————

*P 18: Figure 7 same as for figure 6*

Will add more information and move the figure to the supplement.

—————

*Figure 8: Maybe go for multiple colors. The red frame overlaps with baseline at 0, you hardly can see Qlat. Maybe do not make it a full rectangle but just up from zero but no overlap with x-axis/baseline.*

Will be improved as suggested.

—————

*P19 Table 4: p-values 0.000→ <0.001*

ok.

—————

*P20 Figure 9: Shared Y-axis on the left panel not right, remove or supplement.*

ok.

—————

*P23: L421: Mention your value at "an overall mass loss" so the comparison to the other studies works, this might also give the word "somewhat" in L423 a meaning, else remove it.*

Will add values.

—————

*P24 L435: related→ correlated*

Will be changed.

—————

*P24 L44ff: Is the unit for this not simple m w.e. m-1*

Units will be corrected.

**4 Figures**

[Figure]

Figure I: Cumulative mass balance evolution for disturbed model parameters using ranges from literature. The cumulative mass balance is shown for three selected points (ablation area (a), lower accumulation area (b) and accumulation area (c)).

[Figure]

Figure II: Cumulative internal accumulation evolution for disturbed model parameters using ranges from literature. The cumulative internal accumulation is shown for three selected points (ablation area (a), lower accumulation area (b) and accumulation area (c)).

**References**

Dalum, C. T. V., Berg, W. J. V. D., and Broeke, M. R. V. D.: Impact of updated radiative transfer scheme in snow and ice in RACMO2.3p3 on the surface mass and energy budget of the Greenland ice sheet, The Cryosphere, 15, 1823–1844, https://doi.org/10.5194/tc-15-1823-2021, 2021.

Klok, E. J. and Oerlemans, J.: Model study of the spatial distribution of the energy and mass balance of Morteratschgletscher, Switzerland, Journal of Glaciology, 48, 505–518, https://doi.org/10.3189/172756502781831133, 2002.

Munneke, P. K., Broeke, M. R. V. D., Reijmer, C. H., Helsen, M. M., Boot, W., Schneebeli, M., and Steffen, K.: The role of radiation penetration in the energy budget of the snowpack at Summit, Greenland, The Cryosphere, 3, 155–165, 2009.

Thibert, E., Blanc, R., Vincent, C., and Eckert, N.: Glaciological and Volumetric Mass Balance Measurements: Error Analysis over 51 years for the Sarennes glacier, French Alps, Journal of Glaciology, 54, 522–532, 2008.

Wilks, D. S.: Statistical Methods in the Atmospheric Sciences, Academic Press, 3 edn., 2011.

---

## Author Comment (AC2)

**Response to reviewer 2 (Lindsey Nicholson and Niklas Richter)**

**Long-term firn and mass balance modelling for Abramov glacier, Pamir Alay**

Marlene Kronenberg, Ward van Pelt, Horst Machguth, Joel Fiddes, Martin Hoelzle, Felix Pertziger

Dear Reviewers,

We would like to thank you for your attention to our manuscript. We appreciate the interest in our study and thank for your constructive review and suggestions how to improve the quality of the paper. Below, we respond point by point to all comments, and state how we plan to account for them in a revised version of the paper. The responses (normal font style) to the reviewers' comments are written directly into the reviews (displayed in italic font style). Revised/additional figures can be found at the end of this response letter and are labeled with Roman numerals to them from figures in the manuscript.

Marlene Kronenberg, Fribourg, May 24, 2022

**1 General comments**

Originality: The study makes use of key datasets at Abramov glacier to perform a novel modelling study to reconstruct 52 years of mass balance with concurrent process understanding. While this in itself is novel and the glacier is unusually rich in data for the region, a more explicit statement of the scientific motivation and purpose of the paper as well as critical discussion of how it might contribute to wider glacier process understanding in this region would be welcome. The contribution of modelled subsurface firn conditions is certainly valuable for improving density assumptions for geodetic mass balance estimates and for understanding the evolving potential internal accumulation capacity of the glacier over time. The paper concludes that the effects of warming on glacier mass balance are partially mitigated by increased accumulation, which may be relevant at a regional scale, and highlights that the buffering effect of internal refreezing is diminishing over time.

We will state our scientific motivation more explicitly in the introduction and expand the discussion regarding its contribution to the regional understanding. More details are given below.

Scientific quality: The purpose of the paper could be more clearly stated with reference to what the key outputs are. For example saying that a process model allows you to build on the existing field firn study, and that the climate drivers of glacier mass change remain contested in this region, and can be investigated with a process-based model. We acknowledge that the parameter calibration is performed with rigorous manual calibration, yet by overlooking potential measurement uncertainties, as well as not using a multi-objective simulation/parameter evaluation this is not state of the art. Could you thus expand on the performance of the calibration? It would be good to justify your choice of sample number and location for the calibration datasets, and the treatment of uncertainties could benefit from recognition of a parameter equifinality issue (Rounce et al., 2020) in the optimisation. Rather than reporting only a single best-fit solution, incorporating a sensitivity analysis and its related uncertainties would be more robust. Indeed your comparison of the two sets of forcing data highlights the sensitivity of the modelled mass balance to the forcing data which demonstrates the value of a more comprehensive sensitivity study. This deserves even more emphasis given that the only difference between the two forcing data sets resides in the timespan 1980-1998, the period for which you calculated the biases and should hence be closest to the observations.

We will more clearly state the purpose (Quantification of (i) mass exchange processes in the firn and (ii) changes therin and (iii) of their contribution to the glacier wide mass balance over a long-time period) and the main outputs of our study. Our investigation on firn changes and their impacts on mass balance for Abramov glacier substantially contributes to the process-understanding for this poorly studied area located at the edge of regions with anomalous mass balance trends from a global perspective.

Regarding the calibration: A complete parameter exploration would be numerically extremely expensive and therefore not feasible. Hence, a "smart" solution as used here, selecting key parameters and choosing an appropriate order in which to calibrate them, has been adopted here. Parameters values were updated not only to reduce the bias between modelled and measured mass balances but also in order to simulate processes as realistic as possible. Doing so, the available data, which mainly consists of surface mass balance data, could optimally be used to tackle different processes relevant for a measured phenomena such as ablation (governed by snow melt, albedo degradation and ice melt). This conceptual approach has the advantage, that we found a unique and meaningful value for each parameter with respect to simulated processes. It allows also to circumvent the issue of equifinality which would likely occur if we optimised numerous parameters with the available calibration data. The approach is based on several assumptions and related uncertainties have indeed not been quantified in the current version of the manuscript. We agree that this should be improved. We have therefore conducted model runs with perturbed parameters for selected grid points and use them to evaluate the sensitivity of the modelled surface mass balance and internal accumulation to altered parameter values (cf. Fig. I and II). We will clarify the selection of the calibration data set and its visualisation on the maps and quantify uncertainties of in situ mass balance data following Thibert et al. (2008).

We plan to include a more comprehensive sensitivity discussion including sensitivities of parameters, forcing and initial conditions.

Given the number of pre-processing steps in the forcing data and selection of model parameters, the impact of uncertainty in these choices is poorly quantified, making the model output and performance difficult to interpret, as we cannot be clear about the quality of the forcing data compared to reality. It may be helpful to show plots of the forcing data in the supplementary material as a start. This would also help addressing the difference between the two forcing data sets. Relating to the question of the overarching goal of this study: if the goal is to draw out the connection between the climate forcing and glacier response, investigations on the annual time series of mass balance alongside climatological properties - for example by adding climatological information into Table 3 or attempting to understand the causes of positive mass balance years in the timeseries - could be a starting point. The timeframe of this study allows for an in-depth investigation on the drivers of glacier response also on the decadal scale, as continuously highlighted interdecadal differences in glacier response exist. Therefore, it could be worthwhile to investigate whether there exist any breakpoints in the climatological data that could explain the simulated decadal variations in the mass balance (Table 3).

These pre-processing steps served to prepare a continuous model forcing data set. We will add plots of the model forcing data to the supplementary material (cf. Figs. III and IV). Furthermore, the presentation of a sensitivity analysis will allow to contextualize the choices and enhance the interpretation of the model output and performance. Furthermore, we will visualize the mean annual bias between the model output and the surface mass balance measurements to Figure 4 what will also contribute to the interpretation of the model performance. The visualisation of the simulated energy fluxes and the the climate will be improved to allow for a better interpretation of the decadal variation of modelling results.

Presentation: (1) We suggest restructuring so that the model description comes before the description of the datasets - this will help guide the reader to know the constraints on the construction of the forcing data. If this move is made, we suggest headings as follows under a

Relevance: Key findings are that negative mass balance of -0.27m w.e./year persists for the period 1968/1969-2019/2020 alongside a loss of firn pore space causing a reduction of internal accumulation. Despite increasing air temperatures, no acceleration of glacier-wide mass loss was found over time as a result of increasing precipitation rates which appears to exert a strong control over annual mass balance at this glacier. This is of interest given that knowledge of precipitation in this region is quite poor and is thought to contribute to anomalous glacier behaviour in this region.

'methods' section: study site, model description, forcing data, calibration data, validation data. (2) Please consider also the best way to order Sections 3.5 and 3.4 as they describe parameterisations applied before the application/analysis of the modelled results as well as bias corrections based on whole model simulations. (3) Please revisit the naming of your sites: you have sites 1 and 2 in the accumulation zone, but it would be helpful to call your ablation zone site, site 3 to be consistent in the naming convention and also to show its location on the maps. Relatedly, please can you add a statement as to how representative these chosen sites are for mean ablation zone/accumulation zone conditions. (4) Please show which stake locations/data are used for calibration cf which for validation in one of the map figures as its currently unclear if the locations for calibration are identical to the locations of validation but just a different subset of the data. (5) As currently presented the figures and tables are often far from the associated text which could be improved for the final layout, some suggested changes to figures are included in the specific comments, and some English edits suggested in an annotated pdf.

(1) We agree that restructuring the manuscript will help to guide the reader through the manuscript and thank the reviewer for their suggestion. In the revised manuscript, we will first describe the model followed by the construction of the forcing data.

(2) Sections 3.4. and 3.5. will be re-organised so that the they chronologically describe the applied steps.

(3) We will change the naming of the site in the ablation zone to 'site 3' as suggested and indicate its location on all the maps. A statement of the representativeness of the three locations will be added alongside with results for the three points.

(4) We will more clearly indicate the location of calibration/validation data in Figure 1b, 3 and 10 and clarify in the text.

(5) We will improve the figures and text following the specific comments and the annotated pdf. And the position of the figures/tables will be optimised.

**2 Specific comments**

L13: Is this heterogenous behaviour really only in the last decade? This was a spelling error. We mean 'last decades'.

We will add more information here.

L25: It may be valuable to specify that the advantage of a physically based model is that it is expected to be more suitable for projecting processes into unknown climates, compared to a temperature index model calibrated for one period in time applied within a non-stationary climate projection. The additional caution of course is that many physically based models include stationary parameterizations.

We agree and will complete the statement as suggested.

L18: Is this increase progressive, or stepwise, and what is its magnitude (to help readers not so familiar with the region)?

L28: Please specify which processes and/or which models include/exclude them, or maybe better still just delete the sentence starting "Important ..." as it's really not needed here

We agree to delete the sentence starting "Important...".

L34: Suggest "In the data poor HMA, ..." or maybe better still just delete the sentence starting "In the ..." as it's really not needed given the following sentence

We agree to delete the sentence starting "In the ...".

L44: Suggest restructuring of this closing paragraph to more explicitly state some goals of the study. First state the problem/motivation (e.g. why do we care about this glacier/ timeseries/ region/... need for a continuous record in this poorly understood and anomalously behaving part of the mountain cryosphere?) and how doing this modelling study delivers a novel and useful solution (e.g. to evaluate decadal trends in surface energy balance fluxes, firn evolution ... and their connection to the forcing climate conditions, attempt more process understanding of previously suggested claims?)

This paragraph will be restructured as suggested.

L48: Delete "These in situ data, which are unique for the region, allow us to apply a rather complex model with relatively high data requirements."

Agreed.

L50: Delete the last sentence. Agreed.

L64-69: Give time period of Barundun long term study; and state the long term mass balance of both studies.

Will be completed.

L69: Where is this point site on the glacier?

It is located at s2. A reference to Figure 1b will be added here.

L70: You mention a regime shift in the 1970s - please return to this point in the discussion and in the evidence from your modelled study, does your model show this increase, what are its effects on the glacier scale mass balance and firn development?

The discussion will be completed regarding this point. The modelling results revealed an increase of precipitation, but also an increase of melt energy which especially affects the lower accumulation area during the most recent decades. Whereas refreezing rates stayed similar at high elevations (also at the site presented in Kronenberg et al. (2021), they strongly decreased at lower elevations, which contributed to an overall acceleration of mass loss during the most recent years.

L76: Careful how you use this term reference glacier surface - it has a specific meaning as explained in Huss 2012, but I am not sure you mean this.

Indeed, this was confusing. We will avoid the term in the revised manuscript.

L78: The creation of the 1968 DEM needs more clarification on how you did it and how reasonable it is. Why do you use only 2 DEMs and not intervening satellite DEMs (e.g. SRMT, ASTER), to examine the trend from timestep to timestep? Would it be possible to use coarser, but more frequent dDEMs to check the reasonableness of the linear backwards extrapolation of surface height change over time, e.g. the ones mentioned from Barandun et al. (2015)?

We used the annual height change grid to calculate the DEM1968. This will be clarified. The two DEMs were used as they were readily available at a high spatial resolution and thoroughly quality checked from Denzinger et al. (2021). Glacier dynamics are not included in the EBFM which is by default run for a constant glacier grid. The application of a linearly changing elevation is a straightforward and computationally effective way to include topographical changes. A preparation and quality check of additional DEMs was beyond the scope of this study. Furthermore, we consider DEMs from SAR data as not suitable as the SAR signals are known to penetrate in the subsurface, at least in the accumulation area.

L83: Was this weather station manually monitored? Is the non-digitized data a good candidate for https://www.zooniverse.org/projects/edh/weather-rescue/

Yes, the station was manually monitored. We do not have the original handwritten records. Most of the data are available in a digital format which we plan to deposit in an online repository (we plan to provide the link/reference the link in the final paper).

L85: Suggest rephrasing to "Here, we use data from January 1968 until December 1998 for the following parameters: daily average air pressure, windspeed, relative humidity, temperature and cloud cover, as well as daily precipitation sum, daily minimum temperature and cloud cover and daily maximum temperature."

Will be rephrased.

L89: Can you say something about expected undercatch based on any other measurements in the region or on the likely mountain undercatch for the device used? To my knowledge snow undercatch can be  $c_{50}$  percent in mountain regions depending on the sensor so it would be good to mention here.

Yes, the undercatch is assumed to be substantial especially during winter months when snow is falling at low temperatures (e.g. Sevruk, 1985; Førland and Hanssen-Bauer, 2000). The comparison with monthly snow height measurements indicated a substantial undercatch during winter months, which we account for by applying precipitation bias correction factors. This issue will be mentioned here.

L91: Why 3 hour timesteps, when the ERA5 data is available hourly? If this is because the model needs this time resolution add this to the model description which should preced this section. Did you consider better ways of reconstructing the daily cycle e.g. by using the daily cycle of the

ERA variables? For example it seems inconsistent that you introduce a daily cycle for cloud but then not for precipitation ... these choices need to be justified and their implications considered in the discussion section.

The model default resolution is 3 hours. This resolution allows to represent the day-night cycle but is computationally less demanding than e.g. an hourly resolution. We will first describe the model and then the forcing to improve clarity. The choices will be assessed and discussed in a comprehensive sensitivity discussion.

Section 2.3.2: It seems that you don't actually make much use of the recent weather station - is that correct? Just used in the pressure parameterisation and the radiation parameterization? a. Consider mentioning explicitly the use of each dataset in the analysis.

We will mention the use of each data set. We used the recent AWS data for the air pressure and radiation parametrisations.

L107: Need some more explanation of the occurrence of missing data, what percentage is typically missing and what happens to the NaN data periods?

Only data from the non-NaN periods was used. Will add percentages.

L110: What do you mean by an IQ range filter? Can you specify the scale of this here please? With this and the subsequent double application of the std filter it sounds like you substantially reduce your dataset variability range in a more or less arbitrary fashion before you begin with any physically based QC? Can you explain why?

Here, we followed Stainbank (2018) who performed extensive analysis of the data set which initially contained extreme outliers. We will provide more details to the manuscript.

L116: Other publications show the way for some more rigorous quality control e.g. could correct for snow over SW sensor, using the albedo readings, and for freezing of station using nonvariant wind direction, and for  $RH_{\delta}100$  it is commonly reset to 100 according to many standard instrument manuals (e.g. Campbell scientific). RH values above 100% are usually set to 100%. Also you should report how many/what percentage of the total readings are filtered out (by each of the filtering steps)? Potentially include the whole quality controlled data series in the supplement.

We will provide detailed information about these data in the supplementary material (including quality control).

L122: Could you please just mention why you did not use the extended ERA5 1950-1979 dataset. It might be still preliminary, but did you explore using it at all?

A only very preliminary version of the extended data set was available when we compiled the model forcing data. Our interest was extending the historical *in situ* data time series (1968-1998) to recent years (rather than extending it further into the past). Therefore, we worked with the 'original' ERA5 data set.

L124: Possibly expand on what TopoSCALE does - you input upper air field and a DEM, and get out radiation modifications based on the DEM, which all other fields remain the same for the

given pressure level? For readers unfamiliar with this product it would be worth explaining also why you want to do this to your data first. As a side note, how does TopoSCALE compare to TopoCLIM now available? https://gmd.copernicus.org/articles/15/1753/2022/. Can you also show the ERA5 gridcell location on your map please?

We will explain the TopoSCALE downscaling (3D interpolation of atmospheric fields available on pressure levels and a topographic correction of radiative fluxes) in more detail. The gridcell location will be shown in Figure 1.

TopoSCALE downscales current climate reanlysis data based on a high resolution DEM, whereas TopoCLIM downscales CORDEX RCM data based on high resolution TopoSCALE data, as a best guess of current reference climate, using quantile mapping. Together these methods can produce consistent timeseries of past, current and future climate.

L128: Its not clear why you need cloud cover fraction but presumably for the SEB model, so perhaps state this at the start of the section, or better still put the model description before the data description so that its clear to the reader what is needed to force the model. Does the clear sky value come from ERA5 + TopoSCALE? Why usage of cloud cover fraction instead of readily available longwave radiation from ERA5?

We will put the model description before. The model simulates LWin using cloud cover and Tcl is furthermore used for the simulation of SWin. We followed the approach by Klok and Oerlemans (2002) to get Tcl. The ERA5 long-wave radiation cannot be corrected as no radiation data was measured at the original station. Given the large biases in various other ERA 5 parameters, we prefer not to use any parameter from ERA5 that we cannot bias correct using data from a weather station We thus used the parametrisations in the model and opted against changing them due to a lack of suitable data.

L129: You may want to add units for completeness here.

Agreed.

L135: Are the parameter values developed for a particular site, or globally valid?

The parameter values were found for Nordenskiöldbreen (Svalbard), different values were found by Greuell et al. (1997) for Pasterze glacier (Alps), where  $\tau_{cl}$  depends more strongly on cloud cover. We tested both parameter sets and obtained more realistic radiation values compared to AWS data when using the values for the Arctic site by van Pelt et al. (2012). Due to a lack of data (no simultaneous measurements of cloud cover and SWin), the local values could not be calculated for Abramov glacier.

L140: "Even after downscaling (accounting for resolution diff.)..." here are you referring to what TopoSCALE does, without an explanation of what TopoSCALE is it is not clear to the uninitiated reader at this stage that it is a downscaling scheme, by which I understand it changes the resolution of the reanalysis data?.

Yes, TopoSCALE is a downscaling scheme. We will clarify.

L142: Explain why you aggregate to monthly for the bias correction when your data are available at different timescales, these steps seem confusing. Furthermore, you assume that biases 1980-1998 are the same as in recent decades, is that likely? Why didn't you for example check the other station data to correct biases. Is the height difference between the two weather station locations handled by TopoSCALE?

The two weather stations are located at different locations and elevations and different parameters are recorded with different instruments. A TopoSCALE downscaling can be performed for both locations. Unfortunately, the reproduction of the exact elevation/position of the AWS is hampered by the fact that the AWS is located on a small and exposed rock outcrop which cannot be perfectly reproduced by the DEM used. Furthermore, the AWS measurements are strongly affected by shading and warming effects by nearby peaks and rock walls. The shading effects cannot be reproduced when simulating or downscaling radiation based on DEMs, even if a relatively high resolution (e.g. 25 m) is applied. Therefore, the AWS data are of limited use for debiasing the TopoSCALE data and we opted for consistently use the weather station data available for more than three decades.

L147: After correction using monthly data, cloud cover fraction in summer showed bad fit, therefore set cloud cover to 0 for days without precipitation and to 1 if precipitation is above threshold, this seems flawed as cloudy days could occur without precipitation, would it not be better to correct this with longwave from ERA5 data? Or at least explain and justify these seemingly arbitrary choices - perhaps it is appropriate for this site, but please make a case for this here, using data or citations.

The cloud cover forcing indeed contains uncertainties which are poorly quantified. ERA5 longwave radiation can hardly be used here, as overall too humid conditions result from ERA5. We will present additional sensitivity runs using different cloud forcings and systematically analyze impacts on simulated radiation fluxes to allow for a quantification of related uncertainties.

Monthly averaged deltas were used to the correction of ERA5 TopoSCALE, but its resolution stays hourly. Yes, this model run was to analyze the sensitivity. Will be stated clearer and discussed in detail (new sensitivity discussion).

L150: Can you also show a comparison of the overlapping period of these two datasets, alongside the available weather station data so we can see how well they duplicate each other over periods of overlap - this is important to convince the reader of the homogeneity and robustness of the forcing data derived in this way. Probably fine in the supplement.

Will be added to the supplement (cf. Fig. IV).

L165: Density assumptions including a constant elevation threshold, could you please justify this, especially as your abstract implies the firm is changing density, and presumably changing in

L150: So what is the timescale of these two forcing datasets? Original ERA5 data is hourly, ERA5 TopoSCALE bias corrected data is monthly, and is this dataset 3 hourly. Here please state the purpose of this duplicate dataset ... what do you gain from running the model with two datasets. I think it tells you something about the sensitivity of your results to the forcing data, but please explain this clearly here, and what it tests and does not test.

spatial extent as well? You highlight in the text that this is a major source of error so it might be nice to consider simple uncertainty analyses based around output sensitivity to varying these assumptions. Also why is the firn line 4200m a.s.l., is this based on satellite data or the stake data, please justify this choice in the text.

This choice is done due to a lack of local density measurements at each stake. Values reported by Pertziger (1996) which were established based on average measurements from snow pit measurements. These data are only used for model validation. We will provide error bars to the validation data accounting for the uncertainty of these assumptions. The firn line was set to 4200 m a.s.l. based on field observations.

L174: Please add in the text that March is selected as the end of the winter seasons before the snowpack is modified by melt and that September is the end of the hydrological year at this site, or otherwise justify these with your measured data, or citations.

ok.

L175: What is this selection of 19 stakes for - later it becomes clear this is for calibration, but check this whole paragraph for what you really want to explain.

ok.

Section 3: Moving this section to before the data description would make it much clearer what forcing data is actually needed to drive the model and therefore make it easier to understand the data preparations steps.

Agreed.

Figure 2: It would be useful to refer to this figure more in the data description - e.g. when discussing the two forcing datasets, and also maybe include the borehole temperature data on it? Agreed.

L197: Does the Svalbard development location mean its particularly suitable for this glacier site? Better than alternative models for example? Certainly refreezing is important in Svalbard glaciers, right?

The model is considered particularly suitable as it combines an energy balance routine with a firn model with a suitable degree of complexity regarding the availability of data and objectives of our study. The surface energy balance model explicitly accounts for local topographic effects on the radiative fluxes (e.g. shading by surrounding topography, orientation of the grid cell) which is particularly relevant in rugged mountainous terrain. Refreezing also plays an important role on Svalbard glaciers (see also Marchenko et al. (2017)). Furthermore, the surface energy balance parametrisations were originally developped and applied for Morteratsch glacier (e.g. Klok and Oerlemans, 2002), where conditions are more similar to Abramov glacier compared to other sites for which such models were developed.

L201: Could you include a comment on the model performance from these studies - did it do well in comparison to others?

Will be completed. (In Svalbard, the model has been shown to perform well compared to other models as visible from the comparison with measured data (van Pelt et al., 2019)).

L206: State here which meteorological forcing data are required. Ok.

L212: Indication on connotation of fluxes (positive, negative) with respect to surface. Will be completed.

L214: How does the transition occur? Binary or gradual? It's a linear transition. Will be completed.

L216-208: This modelling of radiation seems odd as I thought this was the point of the TopoSCALE downscaling to give relevant radiative properties for a surface? Please start this section with the required forcing data for this model, and why you chose to not force it with the ERA5 incident radiative fluxes? Are cloud parameters for Svalbard (sea level and maritime) appropriate for Abramov? Can you quantify the better agreement and also state which weather station data was used here?

Our statement was indeed unclear. As stated above, the paper will be restructured and we will clearly highlight the necessary model inputs and the lacking availability of radiative fluxes for the first decades (ERA5 only from 1980 onwards, no measurements). We will also add a quantification.

**L223: Longwave radiation is computed in ERA5, why not use it to begin with?**

Please also refer to our answer above. This is the standard method applied by the EBFM (cf. van Pelt et al., 2012). No longwave radiation is available for the first decade as it was not measured at the station. We prefer to use a consistent method throughout the entire simulation period. Furthermore, the ERA5 longwave radiation would need to need to be pre-processed (spatial resolution, bias correction etc.).

L254: Liquid water instantly distributed following normal distribution, is this a widely used approach and do you know how it behaves in reality? Clearly this is a tricky parameter to know, but are there implications of this assumption?

The approach was extensively investigated by Marchenko et al. (2017). We also refer to Vandecrux et al. (2020) for a comprehensive review of available melt water percolation parameters.

L280: Does this initial subsurface condition differ greatly from just running the first year multiple times - would this not be a more precise way to evolve the starting conditions rather than a 20 year run? Either way its not known if this has a substantial impact on the results. Were there early borehole temperatures as well?

As e.g. visible from figure 4, the first year was particularly positive. It appears more robust, to use average conditions. The used approach represents *in situ* data well: The early borehole

temperatures indicate temperate conditions in the accumulation area as simulated by the model. The model furthermore represents the subsurface densities reasonably well.

L283: So 'adjust' here means optimisation of parameters in some way described later on .. are these point scale optimizations and how does that relate to the fact that the glacier outline is updated over time - are all optimization points within the final glacier boundary?

Yes, all optimization points are within the final boundary (big dots in figure 1b).

L303: In the comparisons does the SMB include refreezing (so the modelled climatic mass balance cf Cogley et al., 2011) or not? Looks like it from L347, but maybe clarify here?

As written, the simulated surface balance is used for comparison to stake data. We will add a reference to the description of the surface balance given earlier in the manuscript.

Table 2: Those parameters identified as optimized in this table are only those optimized through a whole model simulation. This seems a bit misleading as others are also optimized but to other data externally to a full model simulation, perhaps instead just separate literature/optimized values.

ok.

L317: Clarify pressure decay from historic average and modern average (October 2011 - 2020) as you mention the modern timeseries was only used from 2014 onwards due to data gaps (L102). This is a typo. Will be corrected.

L336: This precipitation correction factor is performed within a SEB model simulation like the albedo optimization or is it simply applied to the precipitation timeseries compared to the glaciological data? If the latter doesn't it belong in the forcing data preparation section? Here do you compare the March accumulation in the monthly snowpits to the March accumulation in the forcing data - if so this could be stated more clearly. Maybe also justify the choice of summer bias correction (1.15) from a single different site, is this representative of a range of sites for this instrument?

Yes, the precipitation correction factor for winter months is based on a similar simulation as e.g. the albedo optimization using March snow pit data for evaluation. The value for summer is not from a single site but based on a comprehensive analysis.

L345: "Overall, the mass balance of Abramov glacier is negative for the years from 1968/1969 to 2019/2020." - what is meant here? The mean over the whole period? Give a value or consider giving the cumulative mass balance numerically (which could also be included in Figure for as a line plot on a secondary axis).

Yes, there is an overall mass loss. Will be clarified and completed with numbers.

L347: Consider how to refer to the results - e.g. rather than listing what is shown in each figure you could focus on the description of what is revealed and cite the figure: e.g. The distributed

mean annual mass balance for 1968/1969-2019/2020 is shown in Fig. 3. The mass balance of Abramov glacier is predominantly negative for the years from 1968/1969 to 2019/2020, despite part of this being taken up by glacier retreat, and shows no significant trend in annual mass balances (+0.0002mw.e. a1, p-value=0.979). The most negative modelled mass balances occur at the start of the period, while the two decades between 1978 and 1998 are characterised by almost balanced mass budget, thereafter becoming more negative again (Figure 4). The modelled mass balance gradient shows that accumulation is lowest during the first decade (1968/1969-1977/1978) and highest during the last modelled decade (2008/2009-2017/2018), and ablation is largest during the first decade, followed by the second last decade (1998/1999-2007/2008).

Good point. Thanks for the suggestion. Will be improved accordingly.

Figure 3: Please highlight the different glacier extents with dates or emphasize the fact that the average over the whole timespan with the adjustment of the glacier area leads to the distinct pattern. It might also be nice to report a long term mean ELA in this or the gradient figure (5b). Will add the extents to Figure 3 and also visualize the ELA in the gradient figure.

Figure 4: It woulds be nice to show the cumulative modelled mass balance on a secondary axis. The cumulative mass balance will be shown here as well.

L357: Again here, avoid just listing what is in the figures, but rather tell us what the figures \*show\*. The title makes it seems like this is part of the model validation but in fact here you also show the model calibration results, right? Suggest moving the calibration part to the end of section 3.5. (e.g. The model calibration was performed on eight snow pits locations at the end of March and annual mass balance measurements from 19 stakes and up to eight snow pit locations for the period 1968/1969-1997/1998; the comparison of optimized model simulations to these in-situ observations used for model calibration shows a stronger RMSE for the annual mass balance, and despite some bias in the linear fit, the datasets generally approach the 1:1 linear fit line (Figure 6).) Then here in this section on validation say: e.g. For model validation we compare the simulated mass balance to a set of point measurements not used in the calibration, which can therefore provide an independent measure of model performance. This comparison is based on 146 stake readings not used in model calibration for 1968/1969-1997/1998 (Figure 7a) and all available point mass balance measurements for 2011/2012-2019/2020 (Figure 7b). In both cases the comparison scatter plots cluster around the unity fit line, but inherit the bias tendencies seen in the calibration (Figure 6)) Consider colouring the points in Figure 6 by decade or year as then you might also reveal periods of better/worse model calibration, which could be interesting.

Will be improved as suggested.

Figure 6: The different axis labelling in (b) might create a false impression, please homogenize them

Agreed.

Figure 7: So the performance isn't any better in the measurement period (a) compared to downscaled ERA5 period (b)? This could also be rephrased to the fact that both periods show good

agreement, but given the impact of the two different forcing data sets, I'm a bit hesitant to jump to that conclusion. Further: Could you maybe indicate the number of observations here just to be clear. The scatterplot also doesn't give us an indication of the point or location of the measurements. Generally, I would have preferred to clearly see the points on a map maybe with a highlighting of the calibration sites and also evaluate how you chose the calibration measurement. For the ablation stakes you use less than 10% to calibrate - where does this number come from?

The calibration/validation stakes are shown in the current Figure 1 using different symbols. We will use clearer symbols for distinction and refer to the figure here. We will also add the number of points used for validation in Figure 7. A very direct comparison of the model performance for both periods is hampered by the different amount of available measurements which are furthermore based to the ablation area (especially for the recent years). The bias is also larger for the recent years mainly due to an underestimation of ablation and possibly due to a lack of accumulation measurements in the scatter plot. Will plot the mean annual bias in figure 4 (along with a visualisation of uncertainties of measurements) to allow for a better interpretation.

Figure 8: Depending on the direction of the analysis, the decadal analysis could be further complemented with monthly or seasonal plots.

Agreed.

Table 4: Here annual trends, but earlier figure showed decadal MB, would be interesting to see if there are certain breakpoints in the timeseries corresponding to external forcing (disturbances in atmospheric forcing etc.)?

Will perform additional analysis to detect breakpoints and report in Table 4.

L371: Interesting that you have decreasing RH but increasing precipitation? Does need explanation?

This is likely related to changes in distinct seasons. Will report trends for seasons in the supplementary material.

L396: the correspondence in Fig 13 seems really good to me, perhaps comment in line here that these biases are accompanying a generally good agreement, as evidenced in the figure? Is this match in line with expectations from modelled densification?

Will add a comment in the text as suggested.

L403: Do you return to explain this cooling trend in the discussion?

A discussion of the cooling related to the reduced refreezing (less latent heat release) will be added to the discussion.

Figure 11 and 12 could be a bit larger. ok.

L421: Can you add the value you find in the text here to facilitate comparison with the other studies for which you do quote the values.

ok.

Figure 15: Its cool to see these data which have not been widely included in previous Abramov studies. What is the added value of this study compared to the others, how do the different studies compare to each other? (Please check the data of the Barandun paper - its labelled 2015 in legend and 2021 in cap- tion). This should be mentioned in the discussion. For example, due to the updating of the glacier extent and glacier elevation, the results of this study are more positive compared to the assumption of static glacier extents in the other studies? You could also include your model simulation results from the recent period and the more recent mass balances studies to complete the picture (e.g. given in Table 6 Barandun et al., 2018 and Barandun et al., 2021 Zenodo data set). This would also open the possibility to compare your modelled results to geodetic mass balances for the glacier available e.g. Hugonnet et al, 2021 and/or Miles et al., 2021 which includes an ASTER based elevation change rate based on results of Brun et al., 2017 (https://doi.org/10.5281/zenodo.3843292).

The figure will be extended for recent years as suggested. Our study includes a process-based simulation of internal accumulation which is not included in most of the previous estimates (Barandun et al. (2015) use a simple parametrisation) and also a more detailed (energy-balance) approach at the surface.

Agreed.

L485: This statement of performance is related to the validation against point measurements being better for the main run than the alternative run? Where do we see this in the results shown? Or is it just in the supplement. Suggest moving this statement to the part on uncertainty due to forcing data as it deserves more full discussion.

See answer to comment on line 513.

L490: Ideally this section would provide a formal uncertainty assessment. In your situation, as there are many unquantified errors it would be good to include a sensitivity analysis. If this is at all feasible, we strongly encourage you to add this, otherwise please justify why not. You can then describe your approach at end of this paragraph: "As the sources of uncertainty are diverse and sometimes not readily quantifiable, we here discuss the general sources and likely implications, rather than producing a formal uncertainty assessment for the modelled output."

We agree that this section should be improved. A comprehensive uncertainty assessment for a point location such as e.g. performed by Machguth et al. (2008) is beyond the scope of this study. Different types of errors could be accounted for by such a Monte Carly simulation. However, it gets extremely laborious and computationally heavy to reasonably well quantify the various errors and build the actual simulation, accounting for all identified sources of error. Therefore, we will rather present a relatively comprehensive sensitivity analysis. This analysis includes

L472: Move these first 2 sentences to the conclusions, as they seem a nice summary of what you have already said in this discussion section.

tests for different parameter values following Klok and Oerlemans (2002), model forcings and initial conditions.

L512: What is the implication of these large differences? Does that mean that the more recent timeframe is wrongly estimated? How can this happen if the bias correction is also based on this period? To get an overview over this, plots of the different forcing datasets (at least in Supplement) could help.

First and foremost, these differences emphasize the high model sensitivity to the meteorological forcing and highlight that the lack of continuous weather station data is problematic. On the other hand, the comparison to *in situ* data (cf. Fig. VII) show that the bias are not systematically different for both mass balance simulations. The overall simulation for the recent years is thus not systematically wrong. As the used cloud forcing is assumed to be prone to larger uncertainties than other forcings, the sensitivity to different cloud cover forcing will be separately evaluated and discussed.

L513: Where have you shown that the standard run is 'better' than the alternative run? Can you here add some quantification on the goodness of fit of the main and alternative run? For example in Figure S5 it would be good to see a comparisons with the actual measured MB for this period and report on which forcing data version matches most closely. This is of major importance as, if the alternative forcing produces a very poor match, then it calls into question the validity of the simulated mass balance from 1998 onwards, right? Also consider if this is not better in the main paper rather than the supplement?

Measurements of surface accumulation were overall better represented by the original model run (mean bias: -0.21 m w.e., RMS: 0.44 m w.e. for 1980-1998) than by the alternative model run (mean bias: -0.35 m w.e., RMS: 0.56 m w.e. for 1980-1998), which is relevant for our study with a strong focus on accumulation. The overall bias of measured surface ablation was lower for the alternative model run (+0.18 vs. +0.39 m w.e. for 1980-1998). Whereas the original run similarly underestimated the ablation throughout the entire ablation area (linear regression for 1980-1998 between measured and modelled ablation original run:  $y = 0.88 \times x + 0.16$  (R2 = (0.84), too negative mass balances were simulated around the ELA which compensated for the too low ablation rates on the glacier tongue (linear regression for 1980-1998 between measured and modelled ablation alternative run:  $y = 0.72 \times x - 0.37$  (R2 = 0.80)). The respective scatter plots will be provided in the supplementary material (cf. Figs. V and VI)). Furthermore, the annual mean surface accumulation and ablation bias of both model runs will be visualized in Figure S5 (cf. Fig. VII) which will be moved to the main manuscript. It is visible from Figure VIIb that (i) both model runs systematically underestimate surface accumulation and ablation and that (ii) there is no systematic difference between both runs (see also answer to previous comment).

**3 Figures**

---

## Author Response (AR1)

**Response to reviewer 1 (Tobias Zolles)**

**Long-term firn and mass balance modelling for Abramov glacier in the data-scarce Pamir Alay**

Marlene Kronenberg, Ward van Pelt, Horst Machguth, Joel Fiddes, Martin Hoelzle, Felix Pertziger

Dear Reviewer,

We would like to thank you again for your attention to our manuscript. We appreciate the interest in our study and thank for your constructive review and suggestions how to improve the quality of the paper. Below, we respond point by point to all comments, and state how we accounted for them in a revised version of the paper. The responses (normal font style) to the reviewers' comments and updated paragraphs (in quotation marks) are written directly into the reviews (displayed in italic font style).

Marlene Kronenberg,

*Fribourg, August 9, 2022*

**1 Summary**

*The authors apply an energy and mass balance model for firn and ice to a glacier in High-Mountain Asia (HMA) using the almost 50-year record of meteorological weather station data (AWS) together with down-scaled reanalysis data from ERA5. There is no significant trend in the annual mass balance found, though differences in space and time exist.*

*The manuscript is quite extensive and technical in the methods, for the main results as well as the abstract focusing on the modeling aspect. In particular, it describes the weather station data treatment quite extensively. These parts of the manuscript would benefit greatly from a shortening. The written part of the results is concise, but there are many figures which are not well included in the results or discussion.*

We agree that the manuscript is rather extensive and technical and have therefore shortened the data description in the main manuscript and provide additional information in the supplementary material. We have furthermore restructured the results in order to better embed the figures (some of them were moved to the supplement as suggested) and added references to figures in the discussion section.

*The manuscript describes the measurements, measurement data handling, and treatment in an extensive fashion. Furthermore, throughout the manuscript, the value of this data for the model performance is emphasized. This should be either reduced or also stressed in the abstract and title.*

We agree. We therefore updated the title:

"Long-term firn and mass balance modelling for Abramov glacier in the data-scarce Pamir Alay"

and stress this in the abstract:

"Exceptionally detailed glaciological and meteorological data exist for the Abramov glacier in the Pamir Alay range."

————

**2 General suggestions and comments**

*Shorten and homogenize sections 2.3.1 and 2.3.2 with 2.4.1, which are quite lengthy in comparison to the TopoSCALE and bias correction section which is quite brief. It is also not clear to me, what is done with the precipitation forcing for the AWS time period. For the monthly averages used for the bias correction of the TopoSCALE data mentioned in 2.4.3, is this a monthly average for every January or is it bias corrected for each individual month of the time series? In case of the first, this statement is wrong "Monthly averages of the final cloud fraction time series correspond to observed values for the years for which measurements are available", if the latter, it is a strange way to bias correct, please clarify this.*

We have shortened and homogenized these sections. More detailed descriptions can now be found in the supplement. The entire description of the AWS data has also be moved to the supplement, as these data was used to constrain parameters only (and not as a model forcing). For recent years, precipitation (and all the other forcing data) are based on TopoSCALE ERA5. This should be clearer now, as we reorganised the method/data section following the suggestion by Reviewer II. The statement about the cloud cover fraction was indeed wrong and was therefore corrected:

"Monthly averages of the final cloud fraction time series correspond to the monthly averages prior to correction".

———————

*Why is the cloud cover used and not incoming long-wave radiation?*

The main reason that it is best to use cloud cover instead of LWin is that it enables calculating more spatially detailed fields of LWin (and SWin). LWin does not only depend on cloud cover but also on relative humidity and temperature. Using (interpolated) coarse-scale LWin data instead of calculating LWin would not account for local variability of these weather variables.

All this should be clearer now, as we now present the model prior to the forcing data as suggested by Reviewer II. In addition we included the following statement:

"The EBFM is forced by meteorological data with a three-hourly resolution. The necessary input consists of the following variables: Air temperature, precipitation, air pressure, relative humidity, precipitation and cloud cover fraction."

———————

*Try to shorten 2.5.1 or maybe move the more detailed part to an appendix or supplement.*

We have shortened this section and moved information to the supplement.

———————

*Some statements are very vague or numbers are missing, try to avoid rather/certain thresholds/relatively/etc.*

We are using more precise language and added missing numbers.

———————

*Reconsidering your comment on the surface mass balance: The surface mass balance is the result of accumulation (+) and ablation (-) at the surface including precipitation (+), moisture exchange (+/-), mass loss through runoff (-) and refreezing above the previous summer surface (+). What is refreezing? If it is surface melt water, it is not accumulation for the SMB, as the melted snow was above the previous summer surface too before it melts. If it is refreezing of rain then it falls to the category of precipitation. Please clarify this. For example with "including solid precipitation" and "refreezing of rainwater above the previous summer surface".*

Indeed, the statement was wrong. We have corrected as follows:

"We calculate the climatic mass balance as the sum of the surface and the internal mass balance for hydrological years (1 October-30 September) (Cogley et al., 2011). The surface mass balance is the result of accumulation (+) and ablation (-) at the surface including precipitation (+), moisture exchange (+/-), mass loss through runoff (-). The internal mass balance accounts for re-freezing and storage of liquid water below the previous summer surface."

———————

*Section 3.5 is also quite extensive, try to shorten it, you already have most information in the table anyway*

We shortened the section as suggested.

———————

*For p-values where your statistical analysis tool gave you less than the significant digits change it to "<0.001" instead of = 0.000.*

Done.

———

*In the results section there are a lot of figures: Check for each figure if it is referred to in the main text. Do you really need it is as part of the main manuscript, can even more of them be shown in the supplement or removed altogether? The correlation plots and one of 11 and 12 could be potential starting points. Furthermore, check your figure axes, in the case of shared y-axes readability may be harder than expected.*

We have restructured the results section in order to better embed the figures (of which some were moved to the supplement). As the difference between Figs. 8 and 9 (previous 11 and 12) are important (what is pointed out more clearly now), we kept both figures in the main manuscript. Please also refer to answers to specific comments below.

———

*The uncertainty discussion and estimation did not quantify or investigate the influence of any assumptions like basing fresh snow albedo tuning only on the summer month, the bias correction approach on the input data, parameter choices, etc. What is the influence of the precipitation under catch correction? What is the influence of splitting the cloud cover differently over the day, not conforming o the daily average?*

We now provide a sensitivity analysis including the quantification of parameter and forcing (also based on the two different runs) sensitivities. We perform model runs for selected grid points using perturbed parameters following Klok and Oerlemans (2002) and different cloud cover forcings (different processing steps). Corresponding results are shown in the supplement (cf. Figs. S7-S10). Also, please note that there is no average cloud cover data available - average in line 87 was wrong, was corrected to max.). See also answer to specific comment regarding cloud cover below.

———

*How does the correlation between measured and modeled SMB depend on the point and time? Figure 7 a/b does not show us if the model fails for certain time periods or certain point measurements. This could be further investigated. Additionally, basing the quantification of agreement on $R^2$ is tricky, as this is just about the correlation and not absolute errors, so systematic over- and underestimation are not accounted for. There are multiple ways how to compare measurements with models (Zolles et al. 2019). The choice of comparison method has a direct influence on the evaluation.*

$R^2$ was not used for optimization. We aimed on reducing the bias while keeping an eye on the RMS and regression line to omit compensating effect (e.g. a compensation by an underestimation of melt by an underestimated accumulation). A temporally resolved quantification of the bias was added to Fig. 4b.

———

*There were two different simulations conducted, as mentioned in the summary, this could be used more to emphasize the improvement that additional measurement data could provide.*

A statement was added to section 4.2:

"The differences between both distributed runs further highlight the limitations of gridded products and emphasize the importance of *in situ* data."

———

*During the entire discussion, the uncertainties are all given as the relation to the model and model forcing, though not quantified apart from one alternative run based on a shorter tuning*

*period. The precipitation is here most likely the dominant factor due to uncertainty in climate model and measurements. In addition surface mass balance measurements are also uncertain, Zemp et al. 2013 mention that the related uncertainty of the field measurements at point locations is estimated to be 0.14 m w.e. a⁻¹. What is the impact of this on the uncertainties? How does this change the confidence intervals?*

We have added a more comprehensive sensitivity analysis which will also allow to quantify the influence of different parameters. Model runs for perturbed parameters were preformed following and expanding the analysis presented for Morteratsch glacier by Klok and Oerlemans (2002). In addition, we have estimated the uncertainties of point measurements based on Thibert et al. (2008) and show them together with an annual average misfit and annual modelled mass balance (Fig. 4) for a better estimate of uncertainties. Please note that the parameters were not tuned for the alternative run.

———

**3   General suggestions and comments**

*P2 L27: Wrong Hock reference: I guess: Hock 2005: Progress in Physical Geography 29, 3 (2005) pp.362–391*

The reference was corrected.

———

*P2 L29: acts→removes*

The statement was rephrased.

———

*P2 L34-43: Mention the other studies first, then relate to Pamir Alay*

Done.

———

*P2 L48: Remove relatively*

The sentence was removed as the paragraph was restructured following Reviewer II.

———

*P2 L49: Change to "...mass fluxes over the period from YYYY – YYYY "*

Done.

———

*P2:L50: Delete "to our knowledge"*

Done.

———

*P3 L58: The mean annual .... add "The"*

Done.

———

*P4 L64-70: Could this be moved to the introduction, feels a bit out of place*

This was moved to the introduction.

————

*P4 L87: Remove sentence starting with "Most recorded "*

Done.

————

*P4 L89: Could be misleading as you did correct for undercatch later?*

Agreed. We removed the statement.

————

*P5 L98: We assign observed daily minimum cloud cover to the first four time steps and daily average cloud cover for the rest of the day. What is the impact of this assumption, did you test it, could you verify it? It does not conform to the daily average if 4 steps are lower and the rest is the average?*

Please refer to the new sensitivity experiments regarding the use of different cloud cover forcing for the morning and the afternoon (Figs. S9, S10) and also note that no 'average' cloud cover data is available as only a low and a high (not average) cloud cover values are reported per day reported. (The 'average' in L87 was wrong and therefore corrected).

————

*P5 L105: What is done for precipitation in this period?*

As for the other climate variables, We use debiased TopoSCALE ERA5 data as model forcing. This should be clearer now, as we reorganized the methods and data section following reviewer II.

————

*P5 L106-109: Remove the entire paragraph*

Done. We moved this section to the supplement.

————

*P5 L110: What does this interquartile range filter do? Is this physically reasonable to remove your so to speak outliers, even more so for the outliers that were not detected? The SMB is non-linear with regard to the forcing, is the curve not smoothed this way and the SMB higher? Please clarify, investigate and add to the discussion.*

Upper and lower fences were calculated using the upper ($q_{75}$, median between 50th percentile and uppermost extreme) and lower quartiles ($q_{25}$, median between 50th percentile and lowermost extreme of the data set). (upper fence: $q_{75} + 3 \times q_{75}$-$q_{25}$). The filter was used to identify extreme outliers of the entire data data set (likely caused by malfunctions). More local outliers were than removed using a moving mean method. See also Wilks (2011).

As the corrected AWS data was not used to force the model, there is no consequence on the SMB by this approach. We have now clarified the use of each data set and moved the section about the AWS data pre-processing to the supplement as this data set was of minor importance for our study.

————

*P5 L117: Why $1500W/m^2$?*

Values above appear as outliers when visually analyzing the instantaneous time series.

————

*P6: Section 2.4.2 If you are using ERA5, why is the incoming long-wave radiation not used directly rather than using a cloud cover, which is then adjusted and strangely distributed over the day, and an empiric parametrization using c1 and c2 which are likely different for HMA as your reference used the model on Svalbard*

See answer to general remark (second point) above.

————

*P7 L155: section 2.5.1: Shorten, or put to appendix*

Done.

————

*P8 L228: Is the temperature of the snowfall considered when fresh mass is added to the snowpack? If rain's is not.*

No.

————

*P8 L230: Remove 2nd mention of "subsurface" in this line.*

Done.

————

*P8 L234: What would be the impact of the penetrating short wave radiation with quite thin layers? In addition, as it is mentioned a fresh snow layer in summer often melts extremely fast this might be even more relevant?*

No data is available for this or comparable sites. The absorption can affect the snow temperature (Munneke et al., 2009). Results from Dalum et al. (2021) also highlight that this is a relevant process with impacts on the surface mass balance and refreezing of different areas on Greenland ice sheet.

————

P11 L277: Quantify what the certain threshold is.

A quantification is given in the supplement.

————

P11 L289ff: Check general comments on refreezing.

See answer above.

————

P12 L 315-331: Is this necessary as full text or is table 2 not enough?

This was moved to the supplement.

————

P13 L344: See my general comment for how to compare measurements and model results, which objective did you use? Bias, MAD, RMSD, etc (Zolles et al. 2019)

We mainly aimed on reducing the bias (see also answer above).

————

*P13 Table 1: What is the impact of calibrating fresh snow albedo only in summer?*

The fresh snow albedo is mainly relevant during the melt season. It was calibrated with measurements from the early melt season. The sensitivity regarding perturbations of the fresh snow albedo show, that this parameter critically controls the simulated ablation (Fig. S7). Calibrating this value with summer ablation measurements is reasonable, as solid precipitation also occurs during summer and as these are the months when ablation takes place.

————

*P14 Table 2: The fresh snow density is huge if compared to what is measured. I have used the same value before, but did you try different values?*

No, we haven't. This would indeed be interesting. However, the current value produces reasonable subsurface densities as visible from Fig.10 (previously Fig. 13).

————

*Page 17: Figure 5: The left subplot is not readable, with your choice of colors for the different time periods, you cannot see the values if there is a larger area at a later time for previous times, this is clear for 4000-4300m and maybe at the top (could also be non-changing area there). This has to be changed. The shared Y-axis may be a bit too far off from the other subplots, reduce the white space in between the panels or add the Y elevation axis to each.*

The figure was improved as suggested.

————

*Figure 6: As mentioned above, the different stakes/stake locations or time could be highlighted here, this has the possibility to show more information, else remove.*

The figure was moved to the supplement.

————

*L 360: If that is the only sentence about figure 9, remove figure 9.*

The figure was moved to the supplement.

————

*P 18: Figure 7 same as for figure 6*

We decided to keep this figure in the main manuscript and moved the other scatter plots to the supplement.

————

*Figure 8: Maybe go for multiple colors. The red frame overlaps with baseline at 0, you hardly can see Qlat. Maybe do not make it a full rectangle but just up from zero but no overlap with x-axis/baseline.*

We now use a thinner line to display Qmelt so that Qlat is better visible.

————

*P19 Table 4: p-values 0.000→ <0.001*

Done.

————

*P20 Figure 9: Shared Y-axis on the left panel not right, remove or supplement.*

The axis was corrected and the figure moved to the supplement.

————

*P23: L421: Mention your value at "an overall mass loss" so the comparison to the other studies works, this might also give the word "somewhat" in L423 a meaning, else remove it.*

Done.

————

*P24 L435: related$\rightarrow$ correlated*

Done.

————

*P24 L44ff: Is the unit for this not simple m w.e. m-1*

The unit was corrected as suggested.

**References**

Cogley, J. G., Hock, R., Rasmussen, L. A., Arendt, A. A., Bauder, A., Braithwaite, R. J., Jansson, M., Kaser, G., Möller, M., Nicholson, L., and Zemp, M.: Glossary of Glacier Mass Balance, IHP-VII Technical Documents in Hydrology, IACS Contribution No.2, 86, 114, URL unesdoc.unesco.org/images/0019/001925/192525e.pdf, 2011.

Dalum, C. T. V., Berg, W. J. V. D., and Broeke, M. R. V. D.: Impact of updated radiative transfer scheme in snow and ice in RACMO2.3p3 on the surface mass and energy budget of the Greenland ice sheet, The Cryosphere, 15, 1823–1844, https://doi.org/10.5194/tc-15-1823-2021, 2021.

Klok, E. J. and Oerlemans, J.: Model study of the spatial distribution of the energy and mass balance of Morteratschgletscher, Switzerland, Journal of Glaciology, 48, 505–518, https://doi.org/10.3189/172756502781831133, 2002.

Munneke, P. K., Broeke, M. R. V. D., Reijmer, C. H., Helsen, M. M., Boot, W., Schneebeli, M., and Steffen, K.: The role of radiation penetration in the energy budget of the snowpack at Summit, Greenland, The Cryosphere, 3, 155–165, 2009.

Thibert, E., Blanc, R., Vincent, C., and Eckert, N.: Glaciological and Volumetric Mass Balance Measurements: Error Analysis over 51 years for the Sarennes glacier, French Alps, Journal of Glaciology, 54, 522–532, 2008.

Wilks, D. S.: Statistical Methods in the Atmospheric Sciences, Academic Press, 3 edn., 2011.

**Response to reviewer 2 (Lindsey Nicholson and Niklas Richter)**

**Long-term firn and mass balance modelling for Abramov glacier in the data-scarce Pamir Alay**

Marlene Kronenberg, Ward van Pelt, Horst Machguth, Joel Fiddes, Martin Hoelzle, Felix Pertziger

Dear Reviewers,

We would like to thank you again for your attention to our manuscript. We appreciate the interest in our study and thank for your constructive review and suggestions how to improve the quality of the paper. Below, we respond point by point to all comments, and state how we accounted for them in a revised version of the paper. The responses (normal font style) to the reviewers' comments and updated paragraphs (in quotation marks) are written directly into the reviews (displayed in italic font style). Additional figures can be found at the end of this response letter and are labeled with Roman numerals to distinguish them from figures in the main manuscript and the supplement.

Marlene Kronenberg,

*Fribourg, August 9, 2022*

**1 General comments**

*Originality: The study makes use of key datasets at Abramov glacier to perform a novel modelling study to reconstruct 52 years of mass balance with concurrent process understanding. While this in itself is novel and the glacier is unusually rich in data for the region,a more explicit statement of the scientific motivation and purpose of the paper as well as critical discussion of how it might contribute to wider glacier process understanding in this region would be welcome. The contribution of modelled subsurface firn conditions is certainly valuable for improving density assumptions for geodetic mass balance estimates and for understanding the evolving potential internal accumulation capacity of the glacier over time.The paper concludes that the effects of warming on glacier mass balance are partially mitigated by increased accumulation, which may be relevant at a regional scale, and highlights that the buffering effect of internal refreezing is diminishing over time.*

We have rewritten the last paragraph of the introduction to more clearly state our scientific motivation (cf. answer to next comment) and expanded the discussion regarding its contribution to the regional understanding. More details are given below.

————

*Scientific quality: The purpose of the paper could be more clearly stated with reference to what the key outputs are. For example saying that a process model allows you to build on the existing field firn study, and that the climate drivers of glacier mass change remain contested in this region, and can be investigated with a process-based model. We acknowledge that the parameter calibration is performed with rigorous manual calibration, yet by overlooking potential measurement uncertainties, as well as not using a multi-objective simulation/parameter evaluation this is not state of the art. Could you thus expand on the performance of the calibration? It would be good to justify your choice of sample number and location for the calibration datasets, and the treatment of uncertainties could benefit from recognition of a parameter equifinality issue (Rounce et al., 2020) in the optimisation. Rather than reporting only a single best-fit solution, incorporating a sensitivity analysis and its related uncertainties would be more robust. Indeed your comparison of the two sets of forcing data highlights the sensitivity of the modelled mass balance to the forcing data which demonstrates the value of a more comprehensive sensitivity study. This deserves even more emphasis given that the only difference between the two forcing data sets resides in the timespan 1980-1998, the period for which you calculated the biases and should hence be closest to the observations.*

We have rewritten the last paragraph of the introduction following the reviewers' suggestions to more clearly state our scientific motivation and the purposes of the study:

The paragraph was rewritten as follows: "Abramov glacier is located in the western part of HMA relatively nearby to the data scarce regions for which positive or balanced mass changes have been identified (Fig. 1). Recent firn investigations suggest that the glacier experienced a precipitation increase compared to the 1970s and that the firn the conditions remained similar since then (Kronenberg et al., 2021). The field data, however, only provide information about net accumulation rates and furthermore have a limited temporal and spatial resolution. Modelling the continuous firn and mass balance evolution of Abramov glacier with a process-based model will allow to put the observations into context and give insights into underlying processes. Such a model application of for western HMA unprecedented detail is possible thanks to detailed glaciological and meteorological measurements available for the temperate, valley-type Abramov glacier (Kislov, 1982; Pertziger, 1996; Schöne et al., 2013; Hoelzle et al., 2017). We will apply a coupled surface energy balance – multilayer subsurface model (van Pelt et al., 2012, 2019) to simulate the firn and mass balance evolution from 1968 to 2020. Our objective is to better understand

the underlying processes of observed mass balances for this glacier in response to climatic conditions which may likely be relevant also for other sites in the region. In our analysis, we put a main focus on the accumulation area aiming on a process-based temporally resolved quantification of accumulation processes within the firn and of their contribution to the glacier-wide mass balance."

Regarding the calibration: A complete parameter exploration would be numerically extremely expensive and therefore not feasible. Hence, a "smart" solution as used here, selecting key parameters and choosing an appropriate order in which to calibrate them, has been adopted here. Parameters values were updated not only to reduce the bias between modelled and measured mass balances but also in order to simulate processes as realistic as possible. Doing so, the available data, which mainly consists of surface mass balance data, could optimally be used to tackle different processes relevant for a measured phenomena such as ablation (governed by snow melt, albedo degradation and ice melt).

This conceptual approach has the advantage, that we found a unique and meaningful value for each parameter with respect to simulated processes. It allows also to circumvent the issue of equifinality which would likely occur if we optimised numerous parameters with the available calibration data. The approach is based on several assumptions and related uncertainties have indeed not been quantified in the previous version of the manuscript. We have therefore included additional sensitivity experiments which are introduced in section 2.7, presented in the supplement (Figs. S7-S15) and discussed in section 4.2.

"To assess the model sensitivity towards parameter choices, we performed several model runs for three selected grid points (sites 1, 2 and 3, Fig. 1b) testing single parameter perturbations following Klok and Oerlemans (2002). Additional sensitivity runs are performed using different cloud cover forcings. To assess the overall sensitivity to the model forcing, we carried out an alternative distributed run using the alternative data set with a shorter period of station measurements (1968-1979, Fig. 2). Detailed results of these sensitivity experiments are presented in the supplementary material (Figs. S7-S15)."

and in the discussion section:

"An important source of uncertainties is related to the model forcing. While station data is available for the first thirty years, data from a gridded climate product (ERA5) has to be used for the remaining modelling period. To homogenize the input, the downscaled ERA5 data is bias corrected to match with monthly averages of observations for the overlapping period. The creation of a cloud cover forcing needed for additional processing steps. The sensitivity of related choices is shown in Figs. S9 and S10. Despite the corrections of the TopoSCALE ERA5 forcing, differences in both data sets affect the model output as evident from the results of the alternative model run which yielded a more negative mass balance for the period for which both data sets are available (alternative forcing: -0.23 m w.e. a$^{-1}$ original forcing -0.05 m w.e. a$^{-1}$, Figures S11-S15). Uncertainties in the model forcing may thus be a further reason for the misfits between measured and modelled surface mass balances and our study demonstrates that results with higher confidence can be produced for the period for which in situ measurements serve as a model forcing. The differences between both distributed runs further highlight the limitations of gridded products and emphasize the importance of *in situ* data."

and

"Further model uncertainties are related to the choice of parameters. The sensitivity model runs for the modified parameters (Figs. S7 and S8) highlight that parameter perturbations have strong impacts on mass balance and internal accumulation. The deviations between different model runs is highest for the point location in the lower accumulation area (site 1, cf. Fig. 1b). This underlines the sensitivity and critical role of the firn cover in this elevation zone. Whereas, site 3 remains an ablation site for all parameter scenarios, the most extreme parameter scenarios result in almost balanced mass changes at site 2 ($\sim$4400 m a.s.l, Fig. S7a,c). At, site 1 ($\sim$4250 m a.s.l) internal accumulation reduces for almost all parameter scenarios (Fig. S8b). The largest impact on mass balance

and internal accumulations is caused by the perturbation of the winter precipitation correction factor ($Prec_{c-w}$), and the fresh snow albedo ($\alpha_{snow}$), which were both constrained by the exceptional *in situ* data available for Abramov glacier. "

Furthermore, we have more clearly described the selection of the calibration data set and quantify uncertainties of in situ mass balance measurements following Thibert et al. (2008) (shown in Fig. 4b).

*Given the number of pre-processing steps in the forcing data and selection of model parameters, the impact of uncertainty in these choices is poorly quantified, making the model output and performance difficult to interpret, as we cannot be clear about the quality of the forcing data compared to reality. It may be helpful to show plots of the forcing data in the supplementary material as a start. This would also help addressing the difference between the two forcing data sets. Relating to the question of the overarching goal of this study: if the goal is to draw out the connection between the climate forcing and glacier response, investigations on the annual time series of mass balance alongside climatological properties - for example by adding climatological information into Table 3 or attempting to understand the causes of positive mass balance years in the timeseries - could be a starting point.The timeframe of this study allows for an in-depth investigation on the drivers of glacier response also on the decadal scale, as continuously highlighted interdecadal differences in glacier response exist. Therefore, it could be worthwhile to investigate whether there exist any breakpoints in the climatological data that could explain the simulated decadal variations in the mass balance (Table 3).*

These pre-processing steps served to prepare a continuous model forcing data set. We added plots of the model forcing data to the supplementary material (Figs. S2 and S3). Furthermore, the sensitivity experiments allows to contextualize the choices and enhance the interpretation of the model output and performance (Figs. S7-S15). In addition, we now visualize the mean annual bias between the model output and the surface mass balance measurements in Figure 4b what also contributes to the interpretation of the model performance.

We expanded the uncertainty discussion regarding the implication on changes on accumulation processes following the main focus of the study (...the accumulation area aiming on a process-based temporally resolved quantification of accumulation processes within the firn and of their contribution to the glacier-wide mass balance..).

———————

*Relevance: Key findings are that negative mass balance of -0.27m w.e./year persists for the period 1968/1969-2019/2020 alongside a loss of firn pore space causing a reduction of internal accumulation. Despite increasing air temperatures, no acceleration of glacier-wide mass loss was found over time as a result of increasing precipitation rates which appears to exert a strong control over annual mass balance at this glacier. This is of interest given that knowledge of precipitation in this region is quite poor and is thought to contribute to anomalous glacier behaviour in this region.*

———————

*Presentation: (1) We suggest restructuring so that the model description comes before the description of the datasets - this will help guide the reader to know the constraints on the construction of the forcing data. If this move is made, we suggest headings as follows under a 'methods' section: study site, model description, forcing data, calibration data, validation data. (2) Please consider also the best way to order Sections 3.5 and 3.4 as they describe parameterisations applied before the application/analysis of the modelled results as well as bias corrections based on whole model simulations. (3) Please revisit the naming of your sites: you have sites*

*1 and 2 in the accumulation zone, but it would be helpful to call your ablation zone site, site 3 to be consistent in the naming convention and also to show its location on the maps. Relatedly, please can you add a statement as to how representative these chosen sites are for mean ablation zone/accumulation zone conditions. (4) Please show which stake locations/data are used for calibration cf which for validation in one of the map figures as its currently unclear if the locations for calibration are identical to the locations of validation but just a different subset of the data. (5) As currently presented the figures and tables are often far from the associated text which could be improved for the final layout, some suggested changes to figures are included in the specific comments, and some English edits suggested in an annotated pdf.*

(1) We have restructured the manuscript in order to better guide the reader. We now first describe the model followed by the construction of the forcing data.

(2) Sections 3.4. and 3.5. have been re-organised so that the they chronologically describe the applied steps (now sections 2.6. and 2.8).

(3) We have changed the naming of the site in the ablation zone to 'site 3' as suggested and indicate its location on the maps. As visible from Figs. 3 and 7, the points are representative for the surrounding area.

(4) We use different symbols for calibration/validation data in Fig. 1b and now specify in the legend which data are used for calibration/validation respectively.

(5) We followed specific comments and the annotated pdf in order to improve the figures and text.

————

**2   Specific comments**

*L13: Is this heterogenous behaviour really only in the last decade?*

This was a spelling error. We corrected to 'last decades'.

————

*L18: Is this increase progressive, or stepwise, and what is its magnitude (to help readers not so familiar with the region)?*

We have added more information here.

————

*L25: It may be valuable to specify that the advantage of a physically based model is that it is expected to be more suitable for projecting processes into unknown climates, compared to a temperature index model calibrated for one period in time applied within a non-stationary climate projection. The additional caution of course is that many physically based models include stationary parameterizations.*

We have completed following the suggestion.

————

*L28: Please specify which processes and/or which models include/exclude them, or maybe better still just delete the sentence starting "Important ..." as it's really not needed here*

The sentence was deleted as suggested.

————

*L34: Suggest "In the data poor HMA, …" or maybe better still just delete the sentence starting "In the …" as it's really not needed given the following sentence*

The sentence was deleted as suggested.

———

*L44: Suggest restructuring of this closing paragraph to more explicitly state some goals of the study. First state the problem/motivation (e.g. why do we care about this glacier/ timeseries/ region/… need for a continuous record in this poorly understood and anomalously behaving part of the mountain cryosphere?) and how doing this modelling study delivers a novel and useful solution (e.g. to evaluate decadal trends in surface energy balance fluxes, firn evolution … and their connection to the forcing climate conditions, attempt more process understanding of previously suggested claims?)*

See answer to general comment above.

———

*L48: Delete "These in situ data, which are unique for the region, allow us to apply a rather complex model with relatively high data requirements."*

Done.

———

*L50: Delete the last sentence.*

Done.

———

*L64-69: Give time period of Barandun long term study; and state the long term mass balance of both studies.*

Done.

———

*L69: Where is this point site on the glacier?*

We show the location in Figure 1b and refer to this figure.

———

*L70: You mention a regime shift in the 1970s - please return to this point in the discussion and in the evidence from your modelled study, does your model show this increase, what are its effects on the glacier scale mass balance and firn development?*

We now emphasize the importance of the precipitation increase when analysing and discussing the results (especially in Section 3.4).

———

*L76: Careful how you use this term reference glacier surface - it has a specific meaning as explained in Huss 2012, but I am not sure you mean this.*

Indeed, this was confusing. We avoid the term in the revised manuscript (the section is now in the supplement).

———

*L78: The creation of the 1968 DEM needs more clarification on how you did it and how reasonable it is. Why do you use only 2 DEMs and not intervening satellite DEMs (e.g. SRMT,*

*ASTER), to examine the trend from timestep to timestep? Would it be possible to use coarser, but more frequent dDEMs to check the reasonableness of the linear backwards extrapolation of surface height change over time, e.g. the ones mentioned from Barandun et al. (2015)?*

We have shortened the data section and moved this information to the supplementary material (section S1 Topographical data) where we added complementary information. A preparation and quality check of additional DEMs was beyond the scope of this study. Furthermore, we consider DEMs from SAR data as not suitable as the SAR signals are known to penetrate in the subsurface, at least in the accumulation area.

"Based on these data we calculate an annual height change grid, an elevation grid for 1968 (DEM1968) using the annual height change grid assuming the same linear elevation trend ...

and

As we do not account for glacier dynamics, the application of a linearly changing elevation is a straightforward and computationally effective way to include topographical changes."

————

*L83: Was this weather station manually monitored? Is the non-digitized data a good candidate for https://www.zooniverse.org/projects/edh/weather-rescue/*

Yes, the station was manually monitored. We do not have the original handwritten records. Most of the data are available in a digital format which we plan to deposit in an online repository (we plan to provide the link/reference the link in the final paper).

————

*L85: Suggest rephrasing to "Here, we use data from January 1968 until December 1998 for the following parameters: daily average air pressure, windspeed, relative humidity, temperature and cloud cover, as well as daily precipitation sum, daily minimum temperature and cloud cover and daily maximum temperature."*

We rephrased the statement following the suggestion.

————

*L89: Can you say something about expected undercatch based on any other measurements in the region or on the likely mountain undercatch for the device used? To my knowledge snow undercatch can be ¿50 percent in mountain regions depending on the sensor so it would be good to mention here.*

Yes, the undercatch is assumed to be substantial especially during winter months when snow is falling at low temperatures (e.g. Sevruk, 1985). The comparison with monthly snow height measurements indicated a substantial undercatch during winter months, which we account for by applying precipitation bias correction factors as explained in section 2.6. Following the suggestions by reviewer I, the section about the station data was shortened and we therefore do not further discuss undercatch here.

————

*L91: Why 3 hour timesteps, when the ERA5 data is available hourly? If this is because the model needs this time resolution add this to the model description which should preceed this section. Did you consider better ways of reconstructing the daily cycle e.g. by using the daily cycle of the ERA variables? For example it seems inconsistent that you introduce a daily cycle for cloud but then not for precipitation ... these choices need to be justified and their implications considered in the discussion section.*

We have clarified this by first presenting the model and then the forcing data which is introduced as follows: "The EBFM is forced by meteorological data with a three-hourly resolution. The necessary input consists of the following variables: Air temperature, precipitation, air pressure, relative humidity, precipitation, wind speed and cloud cover fraction."

Please refer to previous answers regarding the sensitivity experiments and discussion regarding the implication of our choices.

———

*Section 2.3.2: It seems that you don't actually make much use of the recent weather station - is that correct? Just used in the pressure parameterisation and the radiation parameterization? a. Consider mentioning explicitly the use of each dataset in the analysis.*

We have restructured and clarified the data/methods section as suggested. We only used the recent AWS data for the air pressure and radiation parametrisations and therefore moved to the description to the supplement. As the filtering procedure is described in detail by Stainbank (2018) and as data set was of minor importance of our study we do not provide more details.

———

*L107: Need some more explanation of the occurrence of missing data, what percentage is typically missing and what happens to the NaN data periods?*

Only data from the non-NaN periods was used. Please also refer the answer above.

———

*L110: What do you mean by an IQ range filter? Can you specify the scale of this here please? With this and the subsequent double application of the std filter it sounds like you substantially reduce your dataset variability range in a more or less arbitrary fashion before you begin with any physically based QC? Can you explain why?*

We followed Stainbank (2018) who performed extensive analysis of the data set which initially contained extreme outliers. Upper and lower fences were calculated using the upper ($q_{75}$, median between 50th percentile and uppermost extreme) and lower quartiles ($q_{25}$, median between 50th percentile and lowermost extreme of the data set). (upper fence: $q_{75} + 3 \times q_{75}$-$q_{25}$). The filter was used to identify extreme outliers of the entire data data set (likely caused by malfunctions). More local outliers were than removed using a moving mean method. See also Wilks (2011).

Please also see answers above regarding this data set.

———

*L116: Other publications show the way for some more rigorous quality control e.g. could correct for snow over SW sensor, using the albedo readings, and for freezing of station using non-variant wind direction, and for RH¿100 it is commonly reset to 100 according to many standard instrument manuals (e.g. Campbell scientific). RH values above 100% are usually set to 100%. Also you should report how many/what percentage of the total readings are filtered out (by each of the filtering steps)? Potentially include the whole quality controlled data series in the supplement.*

Please refer answers above regarding this data set.

———

*L122: Could you please just mention why you did not use the extended ERA5 1950-1979 dataset. It might be still preliminary, but did you explore using it at all?*

ERA5 data was mainly used to extend the forcing data set, therefore a continuous data set was necessary. We clarified this in the section about the ERA5 data set (also see answer to next comment).

––––––––

*L124: Possibly expand on what TopoSCALE does - you input upper air field and a DEM, and get out radiation modifications based on the DEM, which all other fields remain the same for the given pressure level? For readers unfamiliar with this product it would be worth explaining also why you want to do this to your data first. As a side note, how does TopoSCALE compare to TopoCLIM now available? https://gmd.copernicus.org/articles/15/1753/2022/. Can you also show the ERA5 gridcell location on your map please?*

We extended the description of the TopoSCALE downscaling as specified below and now show the location of the ERA5 grid points in Fig. S1.

TopoSCALE downscales current climate reanalysis data based on a high resolution DEM, whereas TopoCLIM downscales CORDEX RCM data based on high resolution TopoSCALE data, as a best guess of current reference climate, using quantile mapping. Together these methods can produce consistent time-series of past, current and future climate.

"Long-term weather station measurements are available for 1968 to 1998 (Fig. 1b, original station), we extend the time-series until 2020 using downscaled ERA5 reanalysis available from ECMWF (Hersbach et al., 2020). We use hourly output from ERA5 for 1980-2020. Data from the original Abramov weather station as well as data from the AWS are completely independent from the ERA5 data set, as the stations are not used during the assimilation procedure (Personal communication from H. Hersbach, ECMWF 2021). We interpolate ERA5 data from the nine grid points located nearest to Abramov glacier (Fig. S1). Air temperature, air pressure, relative humidity, precipitation, global and clear sky radiation are downscaled using TopoSCALE (Fiddes and Gruber, 2014) which performs a 3D interpolation of atmospheric fields available on pressure levels (to account for time-varying lapse rates) and a topographic correction of radiative fluxes. The latter includes a cosine correction of incident direct short-wave radiation on a slope, an adjustment of diffuse short-wave and long-wave radiation by the sky view factor, and an elevation correction of both long-wave and direct short-wave radiation. It has been extensively tested in various geographical regions and applications, e.g. permafrost in the European Alps (Fiddes et al., 2015), permafrost in the North Atlantic region (Westermann et al., 2015), Northern Hemisphere permafrost (Obu et al., 2019), Antarctic permafrost (Obu et al., 2020), Arctic snow cover (Aalstad et al., 2018), Arctic climate change (Schuler and rn Ims Østby, 2020), and Alpine snow cover (Fiddes et al., 2019). This approach enables us to provide a climate length pseudo-observation time series globally while accounting for the main topographic effects on atmospheric forcing."

––––––––

*L128: Its not clear why you need cloud cover fraction but presumably for the SEB model, so perhaps state this at the start of the section, or better still put the model description before the data description so that its clear to the reader what is needed to force the model. Does the clear sky value come from ERA5 + TopoSCALE? Why usage of cloud cover fraction instead of readily available longwave radiation from ERA5?*

We will put the model description before. The model simulates LWin using cloud cover and Tcl is furthermore used for the simulation of SWin. We followed the approach by Klok and Oerlemans (2002) to get Tcl. The ERA5 long-wave radiation cannot be corrected as no radiation data was measured at the original station. Given the large biases in various other ERA 5 parameters, we prefer not to use any parameter from ERA5 that we cannot bias correct using data from a weather station We thus used the parametrisations in the model and opted against changing them due to a lack of suitable data.

––––––––

*L129: You may want to add units for completeness here.*

Agreed.

————

*L135: Are the parameter values developed for a particular site, or globally valid?*

The parameter values were found for Nordenskiöldbreen (Svalbard), different values were found by Greuell et al. (1997) for Pasterze glacier (Alps), where $\tau_{cl}$ depends more strongly on cloud cover. We tested both parameter sets and obtained more realistic radiation values compared to AWS data when using the values for the Arctic site by van Pelt et al. (2012). Due to a lack of data (no simultaneous measurements of cloud cover and SWin), the local values could not be calculated for Abramov glacier.

————

*L140: "Even after downscaling (accounting for resolution diff.)..." here are you referring to what TopoSCALE does, without an explanation of what TopoSCALE is it is not clear to the uninitiated reader at this stage that it is a downscaling scheme, by which I understand it changes the resolution of the reanalysis data?.*

Yes, TopoSCALE is a downscaling scheme. Please also see answer to comment L124.

————

*L142: Explain why you aggregate to monthly for the bias correction when your data are available at different timescales, these steps seem confusing. Furthermore, you assume that biases 1980-1998 are the same as in recent decades, is that likely? Why didn't you for example check the other station data to correct biases. Is the height difference between the two weather station locations handled by TopoSCALE?*

The two weather stations are located at different locations and elevations and different parameters are recorded with different instruments. A TopoSCALE downscaling can be performed for both locations. Unfortunately, the reproduction of the exact elevation/position of the AWS is hampered by the fact that the AWS is located on a small and exposed rock outcrop which cannot be perfectly reproduced by the DEM used. Furthermore, the AWS measurements are strongly affected by shading and warming effects by nearby peaks and rock walls. The shading effects cannot be reproduced when simulating or downscaling radiation based on DEMs, even if a relatively high resolution (e.g. 25 m) is applied. Therefore, the AWS data are of limited use for debiasing the TopoSCALE data and we opted for consistently use the weather station data available for more than three decades.

————

*L147: After correction using monthly data, cloud cover fraction in summer showed bad fit, therefore set cloud cover to 0 for days without precipitation and to 1 if precipitation is above threshold, this seems flawed as cloudy days could occur without precipitation, would it not be better to correct this with longwave from ERA5 data? Or at least explain and justify these seemingly arbitrary choices - perhaps it is appropriate for this site, but please make a case for this here, using data or citations.*

The cloud cover forcing indeed contains uncertainties which were poorly quantified, we therfore performed respective sensitivity runs and show the results in Figs. S9 and S10. ERA5 longwave radiation can hardly be used here, as overall too humid conditions result from ERA5.

————

*L150: So what is the timescale of these two forcing datasets? Original ERA5 data is hourly, ERA5 TopoSCALE bias corrected data is monthly, and is this dataset 3 hourly. Here please*

*state the purpose of this duplicate dataset ... what do you gain from running the model with two datasets. I think it tells you something about the sensitivity of your results to the forcing data, but please explain this clearly here, and what it tests and does not test..*

Monthly averaged deltas were used to the correction of ERA5 TopoSCALE, but its resolution stays 3-hourly. We introduce the sensitivity experiments including the run with the alternative forcing in Section S2.7.

————

*L150: Can you also show a comparison of the overlapping period of these two datasets, alongside the available weather station data so we can see how well they duplicate each other over periods of overlap - this is important to convince the reader of the homogeneity and robustness of the forcing data derived in this way. Probably fine in the supplement.*

A figure was added to the supplement (Fig. S.3)

————

*L165: Density assumptions including a constant elevation threshold, could you please justify this, especially as your abstract implies the firn is changing density, and presumably changing in spatial extent as well? You highlight in the text that this is a major source of error so it might be nice to consider simple uncertainty analyses based around output sensitivity to varying these assumptions. Also why is the firn line 4200m a.s.l., is this based on satellite data or the stake data, please justify this choice in the text.*

This choice is done due to a lack of local density measurements at each stake. Values reported by Pertziger (1996) which were established based on average measurements from snow pit measurements. These data are only used for model validation. We now provide error bars to the validation data (Fig. 4b) accounting for the uncertainty of these assumptions following Thibert et al. (2008). The firn line was set to 4200 m a.s.l. based on field observations.

————

*L174: Please add in the text that March is selected as the end of the winter seasons before the snowpack is modified by melt and that September is the end of the hydrological year at this site, or otherwise justify these with your measured data, or citations.*

This section was shortened/rephrased following the suggestion by reviewer I. The information was added.

————

*L175: What is this selection of 19 stakes for - later it becomes clear this is for calibration, but check this whole paragraph for what you really want to explain.*

The paragraph was restructured.

————

*Section 3: Moving this section to before the data description would make it much clearer what forcing data is actually needed to drive the model and therefore make it easier to understand the data preparations steps.*

Done.

————

*Figure 2: It would be useful to refer to this figure more in the data description - e.g. when discussing the two forcing datasets, and also maybe include the borehole temperature data on it?*

We refer more often to this figure.

————

*L197: Does the Svalbard development location mean its particularly suitable for this glacier site? Better than alternative models for example? Certainly refreezing is important in Svalbard glaciers, right?*

The model is considered particularly suitable as it combines an energy balance routine with a firn model with a suitable degree of complexity regarding the availability of data and objectives of our study. The surface energy balance model explicitly accounts for local topographic effects on the radiative fluxes (e.g. shading by surrounding topography, orientation of the grid cell) which is particularly relevant in rugged mountainous terrain. Refreezing also plays an important role on Svalbard glaciers (see also Marchenko et al. (2017)). Furthermore, the surface energy balance parametrisations were originally developed and applied for Morteratsch glacier (e.g. Klok and Oerlemans, 2002), where conditions are more similar to Abramov glacier compared to other sites for which such models were developed.

————

*L201: Could you include a comment on the model performance from these studies - did it do well in comparison to others?*

In Svalbard, the model has been shown to perform well compared to other models as visible from the comparison with measured data (van Pelt et al., 2019)). We do not include this information here, as it can be found in the cited literature.

————

*L206: State here which meteorological forcing data are required.*

Done.

————

*L212: Indication on connotation of fluxes (positive, negative) with respect to surface.*

Done.

————

*L214: How does the transition occur? Binary or gradual?*

It's a linear transition. This was completed.

————

*L216-208: This modelling of radiation seems odd as I thought this was the point of the TopoSCALE downscaling to give relevant radiative properties for a surface? Please start this section with the required forcing data for this model, and why you chose to not force it with the ERA5 incident radiative fluxes? Are cloud parameters for Svalbard (sea level and maritime) appropriate for Abramov? Can you quantify the better agreement and also state which weather station data was used here?*

The manuscript was restructured and the data needs were clarified (The EBFM computes radiative fluxes using cloud cover as an input).

————

*L223: Longwave radiation is computed in ERA5, why not use it to begin with?*

Please also refer to our answer above. This is the standard method applied by the EBFM (cf. van Pelt et al., 2012). No longwave radiation is available for the first decade as it was not

measured at the station. We prefer to use a consistent method throughout the entire simulation period. Furthermore, the ERA5 longwave radiation would need to need to be pre-processed (spatial resolution, bias correction etc., but no suitable in situ data is available to do this).

—————

*L254: Liquid water instantly distributed following normal distribution, is this a widely used approach and do you know how it behaves in reality? Clearly this is a tricky parameter to know, but are there implications of this assumption?*

The approach was extensively investigated by Marchenko et al. (2017). We also refer to Vandecrux et al. (2020) for a comprehensive review of available melt water percolation parameters.

—————

*L280: Does this initial subsurface condition differ greatly from just running the first year multiple times - would this not be a more precise way to evolve the starting conditions rather than a 20 year run? Either way its not known if this has a substantial impact on the results. Were there early borehole temperatures as well?*

As e.g. visible from figure 4, the first year was particularly positive. It appears more robust, to use average conditions. The used approach represents *in situ* data well: The early borehole temperatures indicate temperate conditions in the accumulation area as simulated by the model. The model furthermore represents the subsurface densities reasonably well.

—————

*L283: So 'adjust' here means optimisation of parameters in some way described later on .. are these point scale optimizations and how does that relate to the fact that the glacier outline is updated over time - are all optimization points within the final glacier boundary?*

Yes, all optimization points are within the final boundary (big dots in figure 1b).

—————

*L303: In the comparisons does the SMB include refreezing (so the modelled climatic mass balance cf Cogley et al., 2011) or not? Looks like it from L347, but maybe clarify here?*

As written, the simulated surface balance is used for comparison to stake data. The computation of the surface balance is explained in the same section above.

—————

*Table 2: Those parameters identified as optimized in this table are only those optimized through a whole model simulation. This seems a bit misleading as others are also optimized but to other data externally to a full model simulation, perhaps instead just separate literature/optimized values.*

This was clarified in the table caption.

—————

*L317: Clarify pressure decay from historic average and modern average (October 2011 - 2020) as you mention the modern timeseries was only used from 2014 onwards due to data gaps (L102).*

This should be 2014 and was corrected.

—————

*L336: This precipitation correction factor is performed within a SEB model simulation like the albedo optimization or is it simply applied to the precipitation timeseries compared to the*

*glaciological data? If the latter doesn't it belong in the forcing data preparation section? Here do you compare the March accumulation in the monthly snowpits to the March accumulation in the forcing data - if so this could be stated more clearly. Maybe also justify the choice of summer bias correction (1.15) from a single different site, is this representative of a range of sites for this instrument?*

Yes, the precipitation correction factor for winter months is based on a similar simulation as e.g. the albedo optimization using March snow pit data for evaluation. The value for summer is not from a single site but based on a comprehensive analysis.

————

*L345: "Overall, the mass balance of Abramov glacier is negative for the years from 1968/1969 to 2019/2020." - what is meant here? The mean over the whole period? Give a value or consider giving the cumulative mass balance numerically (which could also be included in Figure for as a line plot on a secondary axis).*

We rephrased and added a value.

————

*L347: Consider how to refer to the results - e.g. rather than listing what is shown in each figure you could focus on the description of what is revealed and cite the figure: e.g. The distributed mean annual mass balance for 1968/1969-2019/2020 is shown in Fig. 3. The mass balance of Abramov glacier is predominantly negative for the years from 1968/1969 to 2019/2020, despite part of this being taken up by glacier retreat, and shows no significant trend in annual mass balances (+0.0002mw.e. a1, p-value=0.979). The most negative modelled mass balances occur at the start of the period, while the two decades between 1978 and 1998 are characterised by almost balanced mass budget, thereafter becoming more negative again (Figure 4). The modelled mass balance gradient shows that accumulation is lowest during the first decade (1968/1969-1977/1978) and highest during the last modelled decade (2008/2009-2017/2018), and ablation is largest during the first decade, followed by the second last decade (1998/1999-2007/2008).*

We have rephrased following the suggestion.

————

*Figure 3: Please highlight the different glacier extents with dates or emphasize the fact that the average over the whole timespan with the adjustment of the glacier area leads to the distinct pattern. It might also be nice to report a long term mean ELA in this or the gradient figure (5b).*

The glacier outlines were added to Fig. 3. The long-term mean ELA corresponds to the transition zone between red and blue colour tones.

————

*Figure 4: It woulds be nice to show the cumulative modelled mass balance on a secondary axis.*

We now show the cumulative mass balance in Fig. 4c.

————

*L357: Again here, avoid just listing what is in the figures, but rather tell us what the figures \*show\*. The title makes it seems like this is part of the model validation but in fact here you also show the model calibration results, right? Suggest moving the calibration part to the end of section 3.5. (e.g. The model calibration was performed on eight snow pits locations at the end of March and annual mass balance measurements from 19 stakes and up to eight snow pit locations for the period 1968/1969-1997/1998; the comparison of optimized model simulations to*

*these in-situ observations used for model calibration shows a stronger RMSE for the annual mass balance, and despite some bias in the linear fit, the datasets generally approach the 1:1 linear fit line (Figure 6).) Then here in this section on validation say: e.g. For model validation we compare the simulated mass balance to a set of point measurements not used in the calibration, which can therefore provide an independent measure of model performance. This comparison is based on 146 stake readings not used in model calibration for 1968/1969-1997/1998 (Figure 7a) and all available point mass balance measurements for 2011/2012-2019/2020 (Figure 7b). In both cases the comparison scatter plots cluster around the unity fit line, but inherit the bias tendencies seen in the calibration (Figure 6)) Consider colouring the points in Figure 6 by decade or year as then you might also reveal periods of better/worse model calibration, which could be interesting.*

The text has been rephrased following the suggestion. The calibration is now described in the methods section and the figure was moved to the supplement as suggested by reviewer I. The temporal variation of the bias between measured and modelled mass balances is now shown in Fig. 4b.

―――――

*Figure 6: The different axis labelling in (b) might create a false impression, please homogenize them*

Done.

―――――

*Figure 7: So the performance isn't any better in the measurement period (a) compared to downscaled ERA5 period (b)? This could also be rephrased to the fact that both periods show good agreement, but given the impact of the two different forcing data sets, I'm a bit hesitant to jump to that conclusion. Further: Could you maybe indicate the number of observations here just to be clear. The scatterplot also doesn't give us an indication of the point or location of the measurements. Generally, I would have preferred to clearly see the points on a map maybe with a highlighting of the calibration sites and also evaluate how you chose the calibration measurement. For the ablation stakes you use less than 10% to calibrate - where does this number come from?*

The number of point measurements used for comparison is given in the caption of Fig. 6 (prev. Fig.7). A very direct comparison of the model performance for both periods is hampered by the different amount of available measurements which are furthermore based to the ablation area (especially for the recent years). The bias is also larger for the recent years mainly due to an underestimation of ablation and possibly due to a lack of accumulation measurements in the scatter plot. We have now included a visualisation the mean annual bias in figure 4b (along with a visualisation of uncertainties of measurements) to allow for a better analysis in section 3.1:

"The RMSE between simulated surface mass balances and independent point measurements not used for calibration is similar for the recent and historical investigation periods ($\sim$0.7), whereas the mean bias lower for the period of historical measurements ($+0.05$) than for recent years ($+0.28$) (Fig. 6a and b). Differences in the mean bias are partly related to the spatial distribution of the validation data. As visible from 6a,b and 4b), both, accumulation and ablation were underestimated whereas more accumulation measurements are available for the historical period (1b). "

―――――

*Figure 8: Depending on the direction of the analysis, the decadal analysis could be further complemented with monthly or seasonal plots.*

We updated this section in order to focus on the main objectives of the study. Therefore, a decadal analysis provides enough detail.

"Temporal variation of climate variables, mass and energy fluxes: Significant increasing trends for model forcing air temperature (mean over hydrological years) and mean summer air temperatures are found for 1968/1969-2019/2020 (Table 4). Trends are also significant for air pressure (increase), relative humidity (decrease), cloud cover fraction (decrease) and precipitation sums (increase) (Table 4). Highest amounts of precipitation were recorded from 1978/1979 to 1997/1998 and from 2008/2009 to 2017/2018 (Table S2). Whereas the earlier two decades are characterised by almost balanced mass changes, more negative values were simulated for the most recent decade, when the internal accumulation was strongly reduced compared to earlier years (Table 3). Low internal accumulation was also simulated for the second and most balanced decade (1978/1979 to 1987/1988) (Table 3). Both decades of low internal accumulation follow years with exceptionally high amounts of available melt energy and comparably low precipitation rates (especially from (1968/1969 to 1977/1978) (Tables 3, S2; Figs. 12 and S5). Overall, the preceding conditions led to a reduction of the area where internal accumulation could take place (only higher elevation bands in Fig. 5c, see also video supplement). Following 1978/1979-1987/1988, when precipitation was high, the area where internal accumulation takes place in larger areas (Fig. 5c and video supplement). In most recent years which follow another decade characterised by negative mass balances the area and amount of internal accumulation are strongly reduced (Table 3, Fig. 5). In general, the simulated annual mass balances are more strongly correlated with annual precipitation ($R^2$=0.72) than to the summer air temperatures ($R^2$=0.29) Fig. S6). Examining the modelled energy fluxes for different sampled locations (lower accumulation area site1, accumulation area site 2 and ablation area site3, Fig. 1b) highlights, that most heat fluxes are characterised by an increasing trend (Table 4), whereas high incoming shortwave radiation occurred during the first decade characterised by the most pronounced mass loss (Figs. 12 and S5). In 2008/2009-2017/2018, when melt rates were highest for the points in the accumulation area, sensible heat flux and incoming long-wave radiation were high (Figs. 12, S5 and Table S2). "

––––––––

*Table 4: Here annual trends, but earlier figure showed decadal MB, would be interesting to see if there are certain breakpoints in the timeseries corresponding to external forcing (disturbances in atmospheric forcing etc.)?*

As the decadal variations together with the simulated firn changes allow to discuss the processes of interest we do not provide any additional timeseries analysis here. See also answer to previous comment.

––––––––

*L371: Interesting that you have decreasing RH but increasing precipitation? Does need explanation?*

This is likely related to changes in distinct seasons. This section was changed in order to follow the general direction of the analysis.

––––––––

*L396: the correspondence in Fig 13 seems really good to me, perhaps comment in line here that these biases are accompanying a generally good agreement, as evidenced in the figure? Is this match in line with expectations from modelled densification?*

The statement was completed:

"Modelled firn densities correspond well with measurements shown for four different dates in Fig. 10."

––––––––

*L403: Do you return to explain this cooling trend in the discussion?*

We added a paragraph about the cooling trend to the discussion.

"The changes are also reflected in a decrease of subsurface temperatures. The reduction and later absence of latent heat release through refreezing leads to cold firn properties at ∼4250 m a.s.l. for recent years (Fig. 8d-f)."

————

*Figure 11 and 12 could be a bit larger.*

Figures were enlarged.

————

*L421: Can you add the value you find in the text here to facilitate comparison with the other studies for which you do quote the values.*

Done.

————

*Figure 15: Its cool to see these data which have not been widely included in previous Abramov studies. What is the added value of this study compared to the others, how do the different studies compare to each other? (Please check the data of the Barandun paper - its labelled 2015 in legend and 2021 in caption). This should be mentioned in the discussion. For example, due to the updating of the glacier extent and glacier elevation, the results of this study are more positive compared to the assumption of static glacier extents in the other studies? You could also include your model simulation results from the recent period and the more recent mass balances studies to complete the picture (e.g. given in Table 6 Barandun et al., 2018 and Barandun et al., 2021 Zenodo data set). This would also open the possibility to compare your modelled results to geodetic mass balances for the glacier available e.g. Hugonnet et al, 2021 and/or Miles et al., 2021 which includes an ASTER based elevation change rate based on results of Brun et al., 2017 (https://doi.org/10.5281/zenodo.3843292).*

The Barandun reference was corrected. Our study includes a process-based simulation of internal accumulation which is not included in most of the previous estimates (Barandun et al. (2015) use a simple parametrisation) and also a more detailed (energy-balance) approach at the surface. The comparison shows, that our approach agrees relatively well with previous studies and furthermore accounts for firn processes as discussed in section 4.1. We do not include a visualisation of other mass balance estimates for recent years as those estimates are not based on in situ measurements and therefore don't serve to discuss processes.

————

*L472: Move these first 2 sentences to the conclusions, as they seem a nice summary of what you have already said in this discussion section.*

Agreed.

————

*L485: This statement of performance is related to the validation against point measurements being better for the main run than the alternative run? Where do we see this in the results shown? Or is it just in the supplement. Suggest moving this statement to the part on uncertainty due to forcing data as it deserves more full discussion.*

The introductory statement of this section was updated. See also answer to comment on line 513.

————

*L490: Ideally this section would provide a formal uncertainty assessment. In your situation, as there are many unquantified errors it would be good to include a sensitivity analysis. If this is at all feasible, we strongly encourage you to add this, otherwise please justify why not. You can then describe your approach at end of this paragraph: " As the sources of uncertainty are diverse and sometimes not readily quantifiable, we here discuss the general sources and likely implications, rather than producing a formal uncertainty assessment for the modelled output."*

A comprehensive uncertainty assessment for a point location such as e.g. performed by Machguth et al. (2008) is beyond the scope of this study. Different types of errors could be accounted for by such a Monte Carlo simulation. However, it gets extremely laborious and computationally heavy to reasonably well quantify the various errors and build the actual simulation, accounting for all identified sources of error.

We have therefore updated the introductory statement following the suggestion:

"The EBFM reproduces the observations satisfactorily, as shown by the the comparison of modelled and measured surface mass balances (Fig. 6) and the comparison of subsurface properties (Figs. 10, 11). These comparisons of modelled and measured data provide an overall estimate of the combined modelling and observational uncertainty. The sources of the modelling uncertainty are diverse and sometimes not readily quantifiable, we here discuss the general sources and likely implications based on the performed sensitivity experiments and general considerations, rather than producing a formal uncertainty assessment for the model output."

In addition, we refer to the results of the more extensive sensitivity experiments in the discussion as specified in our answer to the general remark above.

———

*L512: What is the implication of these large differences? Does that mean that the more recent timeframe is wrongly estimated? How can this happen if the bias correction is also based on this period? To get an overview over this, plots of the different forcing datasets (at least in Supplement) could help.*

First and foremost, these differences emphasize the high model sensitivity to the meteorological forcing and highlight that the lack of continuous weather station data is problematic. On the other hand, the comparison to *in situ* data (cf. Fig. S14b) show that the bias are not systematically different for both mass balance simulations. The overall simulation for the recent years is thus not systematically wrong. As the used cloud forcing is assumed to be prone to larger uncertainties than other forcing variables, the sensitivity to different cloud cover forcing was separately evaluated (Figs. S9, S10).

———

*L513: Where have you shown that the standard run is 'better' than the alternative run? Can you here add some quantification on the goodness of fit of the main and alternative run? For example in Figure S5 it would be good to see a comparisons with the actual measured MB for this period and report on which forcing data version matches most closely. This is of major importance as, if the alternative forcing produces a very poor match, then it calls into question the validity of the simulated mass balance from 1998 onwards, right? Also consider if this is not better in the main paper rather than the supplement?*

Measurements of surface accumulation were overall better represented by the original model run (mean bias: -0.21 m w.e., RMS: 0.44 m w.e. for 1980-1998) than by the alternative model run (mean bias: -0.35 m w.e., RMS: 0.56 m w.e. for 1980-1998), which is relevant for our study with a strong focus on accumulation. The overall bias of measured surface ablation was lower for the alternative model run (+0.18 vs. +0.39 m w.e. for 1980-1998). Whereas the original run

similarly underestimated the ablation throughout the entire ablation area (linear regression for 1980-1998 between measured and modelled ablation original run: y = 0.88 × x +0.16 ($R^2$ = 0.84), too negative mass balances were simulated around the ELA which compensated for the too low ablation rates on the glacier tongue (linear regression for 1980-1998 between measured and modelled ablation alternative run: y = 0.72 × x -0.37 ($R^2$ = 0.80)). See attached scatter plots Figs. I and II)). We now show the annual mean surface accumulation and ablation bias of both model runs in Figure S14 (previous Fig.S5). Note that both model runs systematically underestimate surface accumulation and ablation and that there is no systematic difference between both runs.

**3 Figures**

[Figure]

Figure I: Validation of original model run against an independent set of point surface mass balance measurements for 1979/1980-1997/1998. Measured versus modelled annual surface accumulation (a) and surface ablation (b).

[Figure]

Figure II: Validation of alternative model run against an independent set of point surface mass balance measurements for 1979/1980-1997/1998. Measured versus modelled annual surface accumulation (a) and surface ablation (b).

**References**

[revised manuscript text omitted]

---

## Referee Report (RR1)

**Review to Long-term firn and mass balance modelling for Abramov glacier, Pamir Alay**

Summary:

The authors apply an energy and mass balance model for firn and ice to a glacier in High-Mountain Asia (HMA) using the almost 50-year record of meteorological weather station data (AWS) together with down-scaled reanalysis data from ERA5. There is no significant trend in the annual mass balance found, though differences in space and time exist.

There is a significant improvement of the manuscript compared to the initial submission. The current manuscript is easier to follow and much more streamlined, some unnecessary figures have been moved to the supplement. Clear answers and adjustments have been made and the restructuring was to the benefit of the overall quality of the work. However, there are still some more minor changes to make which fall in line with the previous comments.

General remarks:

Ask yourself again what does the figure show and why do you show it. Does it show relevant info or just proves that the model works (its not a model development paper)? For example, Figure 11 shows subsurface temperatures being too high in the model for June 2018. The plot is shown, the results about it are briefly mentioned, but there is not really a discussion linked to it. Why is there disagreement or not between model and observations. One could speculate about the effect of fresh snow fall albedo increase in the model, you mention that during observations it fresh snow often melts early, etc. I am wondering if during 4.2 the discussion, fig. 11 could be linked more to the discussion or else why is it shown for this particular time period (2018) or at all.

All correlation/regression plots show regressions of data points with larger modeled than measured SMBs (away from the line of equality) (Fig 6,14, S4). This is seams to be a systematic effect from the modeling. Could you comment on this, and mention and discuss it in uncertainty/discussion. Why is this observed? It is only briefly mentioned that the model does underestimate ablation as well as accumulation, but does it, or only extreme values?

There are still some vague statements for quantities: "a certain threshold" etc. Try to avoid those and provide numbers.

Specific comments:

Figure 5a: The colors are updated now from gray shading, which makes it easier to see, but from my understanding of how the plot is composed it still plots the later years on top of the earlier so if the maximum glacier extent was observed at a later time scale one cannot see the extent of earlier times? So the problem remains, could transparency solve this?

Line numbering refers to the markup version:

Line 26: Should it be singular? climates-> climate?

line 285 : "Liquid water is instantly distributed along the depth axis following a normal distribution until a maximum depth zlim unless it reaches an impermeable ice layer before." how big is z_lim roughly?

Line 334 value→ valueS

Line 335ff: I am still wondering when you distribute precipitation over the entire day equally but the cloud cover not what effect this has. At least for days with precipitation shouldn't the cloud-cover be correlated with the precipitation. You reason for your choice of cloud cover distribution due to convection effects ("During the melt season, convection is a main driver of cloudiness and cloud formation mainly takes place along the mountain ridges (Suslov and Krenke, 1980)."). On non synoptic scale precipitation events, are you not expecting more in the second half of the day too? Would this play a role? As your albedo due to earlier precipitation on the day may be too high? Or are convective solid precipitation events (snowfall) rare? Your comment specifically also mentions the melt season, is this applied for the entire year – I assume yes, but its not 100% clear out of the manuscript.

Line 393 which monthly precipitation threshold value, provide a number

Line 480: It is not entirely clear what a stronger RMSE is, i.e. stronger wrong word, higher? But what is higher? What is the reference? Higher than bias? Higher than RMSE for winter/summer balance?

501 ", and mass loss through runoff " . I would add an "and "

504 Change "other results" to "other (output) variables"

505: The sentence does not make sense, though I get what you mean: ... is analysed on a gird/mask corresponding ... for decade-wise updated glacier extents. Until the end of the hydrological year 1978, the entire model output is analysed corresponding to the glacier area of 1975.

531-535 There are multiple formatting issues with the parenthesis ( and which units do the values provided here have? kg/m² (mm w.e.)

621: One of the parenthesis is too much or there is one too little

707: "EBFM reproduces the observations satisfactorily, as shown by the the comparison of modelled and measured surface mass balances (Fig. 6) and especially the comparison of subsurface properties (Figs. 10, 11). How does 4b fall in line with that, with annual biases around of up to 0.7m w.e.?

Figure 4b: Caption: I am not sure what is meant with bias of surface accumulation and surface ablation. Are those the values of measured accumulation and ablation over the entire glacier over the entire year or for ablation area? i.e. what is exactly compared here? You do not have a continuous height record at your measurement stations I guess?

Regarding the Supplement:

Supplement: Are the figures S2, S3 necessary? Can you really see something there? Does it make more sense or is it available as downloadable data?

S 14. Does it make sense to mention in the caption again what composes the alternative forcing? For better understanding

---

## Author Response (AR2)

**Response to reviewer comments**

**Long-term firn and mass balance modelling for Abramov glacier in the data-scarce Pamir Alay**

Marlene Kronenberg, Ward van Pelt, Horst Machguth, Joel Fiddes, Martin Hoelzle, Felix Pertziger

Dear Editor,

We appreciate the interest in our study and thank you for editing our study. We would like to thank the reviewer for his constructive comments and suggestions on the revised manuscript. Below, we respond point by point to all comments, and state how we accounted for them in a revised version of the paper. The responses (normal font style) to the reviewers' comments and updated paragraphs (in quotation marks) are written directly into the reviews (displayed in italic font style).

Marlene Kronenberg

*Fribourg, November 7, 2022*

**Summary**

*The authors apply an energy and mass balance model for firn and ice to a glacier in High-Mountain Asia (HMA) using the almost 50-year record of meteorological weather station data (AWS) together with down-scaled reanalysis data from ERA5. There is no significant trend in the annual mass balance found, though differences in space and time exist.*

*There is a significant improvement of the manuscript compared to the initial submission. The current manuscript is easier to follow and much more streamlined, some unnecessary figures have been moved to the supplement. Clear answers and adjustments have been made and the restructuring was to the benefit of the overall quality of the work. However, there are still some more minor changes to make which fall in line with the previous comments.*

**General remarks**

*Ask yourself again what does the figure show and why do you show it. Does it show relevant info or just proves that the model works (its not a model development paper)? For example, Figure 11 shows subsurface temperatures being too high in the model for June 2018. The plot is shown, the results about it are briefly mentioned, but there is not really a discussion linked to it. Why is there disagreement or not between model and observations. One could speculate about the effect of fresh snow fall albedo increase in the model, you mention that during observations it fresh snow often melts early, etc. I am wondering if during 4.2 the discussion, fig. 11 could be linked more to the discussion or else why is it shown for this particular time period (2018) or at all.*

Figure 11 nicely shows that the EBFM is able to reproduce temperate firn conditions and we therefore decided to show it. The misfit between modelled and measured subsurface temperatures in June 2018 indicates that firn warming the modelled firn warming at the beginning of the melt season is somewhat accelerated compared to the measurements. This might be due to a reduced modelled subsurface refreezing (cooling effect) caused by the lack of liquid water and/or the lack of cold content. Due to the relatively coarse spacing of thermistor measurements it is not possible to track melt water percolation and single refreezing events in the measurements. A discussion of the processes would thus remain rather speculative and we therefore decided not to discuss this further in Section 4.2.

*All correlation/regression plots show regressions of data points with larger modeled than measured SMBs (away from the line of equality) (Fig 6,14, S4). This is seams to be a systematic effect from the modeling. Could you comment on this, and mention and discuss it in uncertainty/discussion. Why is this observed? It is only briefly mentioned that the model does underestimate ablation as well as accumulation, but does it, or only extreme values?*

Several reasons for the misfit between modelled and measured mass balances are discussed in section 4. The reasons include grid resolution, model simplifications and the choice of constant parameter values (e.g. albedo) but also the distribution of point measurements. We completed section 4.2. with the following statements:

"The linearly updated glacier surface might thus be a reason for the underestimation of melt rates on the glacier tongue as visible from Figs. 6a,b and S4b."

and

"The lack of considering those processes also explains the misfit between modelled and measured accumulation as visible from Figs. 6a,b and S4a,b to some extent. Other reasons for the misfit are likely related to the limited spatial representativeness of the rather low number of accumulation observations."

*There are still some vague statements for quantities: "a certain threshold" etc. Try to avoid those and provide numbers.*

The threshold values are given in the supplementary material (see also answer to specific comments below).

**Specific comments**

*Figure 5a: The colors are updated now from gray shading, which makes it easier to see, but from my understanding of how the plot is composed it still plots the later years on top of the earlier so if the maximum glacier extent was observed at a later time scale one cannot see the extent of earlier times? So the problem remains, could transparency solve this?*

The glacier area reduced over the decades shown in Figure 5a. This is now specified in the caption:

"The largest glacier area was observed for the first decade (shown in dark blue) and reduced over time."

————

*Line numbering refers to the markup version:*

*Line 26: Should it be singular? climates-¿ climate?*

Agreed.

————

*Line 285 : "Liquid water is instantly distributed along the depth axis following a normal distribution until a maximum depth zlim unless it reaches an impermeable ice layer before." how big is z_lim roughly?*

$z_{lim}$ is 6 m as specified in Table 2.

————

*Line 334 value→ valueS*

Agreed.

————

*Line 335ff: I am still wondering when you distribute precipitation over the entire day equally but the cloud cover not what effect this has. At least for days with precipitation shouldn't the cloud-cover be correlated with the precipitation. You reason for your choice of cloud cover distribution due to convection effects ("During the melt season, convection is a main driver of cloudiness and cloud formation mainly takes place along the mountain ridges (Suslov and Krenke, 1980)."). On non synoptic scale precipitation events, are you not expecting more in the second half of the day too? Would this play a role? As your albedo due to earlier precipitation on the day may be too high? Or are convective solid precipitation events (snowfall) rare? Your comment specifically also mentions the melt season, is this applied for the entire year – I assume yes, but its not 100% clear out of the manuscript.*

Yes, we apply this the entire year. (We refer to the melt season because this is when convective clouds were observed). We agree that non-synoptic scale precipitation events also rather occur later in the day. However, only data are reported. We consider albedo related Uncertainties due to too high precipitation in the morning to be relatively low as precipitation is not frequent during the melt season.

―――――

*Line 393 which monthly precipitation threshold value, provide a number*

The values are given in S1.4 of the supplementary material as specified here.

―――――

*Line 480: It is not entirely clear what a stronger RMSE is, i.e. stronger wrong word, higher? But what is higher? What is the reference? Higher than bias? Higher than RMSE for winter/summer balance?*

We rephrased as follows:

"The comparison of optimized model simulations to surface mass balance observations used for model calibration shows a higher Root Mean Square Error (RMSE) for the annual mass balance than for the accumulation in March..."

―――――

*line 501 ", and mass loss through runoff " . I would add an "and "*

Agreed.

―――――

*line 504 Change "other results" to "other (output) variables"*

Agreed.

―――――

*line 505: The sentence does not make sense, though I get what you mean: ... is analysed on a gird/mask corresponding ... for decade-wise updated glacier extents. Until the end of the hydrological year 1978, the entire model output is analysed corresponding to the glacier area of 1975.*

We rephrased as follows:

"After modelling, the glacier wide mass balance and other variables are calculated for decade-wise updated glacier extents. Until the end of the hydrological year 1978, the entire model output is analysed. For the next ten hydrological years, ..."

―――――

*lines 531-535 There are multiple formatting issues with the parenthesis (and which units do the values provided here have? kg/m² (mm w.e.)*

The formatting issues were corrected and units were added.

―――――

*line 621: One of the parenthesis is too much or there is one too little*

Agreed.

―――――

*line707: "EBFM reproduces the observations satisfactorily, as shown by the the comparison of modelled and measured surface mass balances (Fig. 6) and especially the comparison of subsurface properties (Figs. 10, 11). How does 4b fall in line with that, with annual biases around of up to 0.7m w.e.?*

In Figure 4b, we show accumulation and ablation biases separately for single years, whereas the overall bias is given in Fig. 6. Indeed, biases are relatively large for some years. We therefore added the following sentence here:

"Whereas the overall biases between modelled and measured surface mass balances are low (0.05 m w.e. for historical and 0.28 m w.e. for recent years), relatively large biases exist for single years (Fig. 4b)."

————

*Figure 4b: Caption: I am not sure what is meant with bias of surface accumulation and surface ablation. Are those the values of measured accumulation and ablation over the entire glacier over the entire year or for ablation area? i.e. what is exactly compared here? You do not have a continuous height record at your measurement stations I guess?*

We clarified the caption as follows:

"Mean annual surface accumulation bias (blue) and mean annual ablation bias (red) between point measurements and model output for corresponding grid points".

---

## Author Response (AR3)

**Response letter**

**Long-term firn and mass balance modelling for Abramov glacier in the data-scarce Pamir Alay**

Marlene Kronenberg, Ward van Pelt, Horst Machguth, Joel Fiddes, Martin Hoelzle, Felix Pertziger

Dear Editor,

thank you very much for your feedback. We have finalised the manuscript following your suggestions and made the following technical corrections:

1. We corrected the spelling mistake.

2. We removed "where".

3. Table 2 is referenced in line 270, we added a reference to table 2 when describing z_lim on line 135.

4. We completed the caption of Figure 11: "Modelled and measured subsurface temperature for a station located nearby site 2 ($\sim$ 4400 m a.sl.) for four selected dates visualising the constantly temperate conditions at depths below $\sim$ 10 m in spring 2018. The data are plotted for days around the onset of modelled surface melt occurring in March 2018 (a,b) and subsequent subsurface warming (b,c,d). The 6th of June visualized in d corresponds to the date with the coldest measured temperatures at a depth of about 7 m in 2018. The location of the site is indicated in Fig. 1b)."

5. We completed the discussion with a statement regarding the sub-daily variation of cloud cover and precipitation (L484): "Furthermore, the cloud cover forcing from the period with measurements is another source uncertainties which are related to the observational nature of the data and our choice to implement sub-daily cloud cover variations throughout the year while keeping precipitation constant throughout a day."

6. We changed L500 as suggested and added a statement regarding equifinality (L504): "The sensitivity model runs for the modified parameters (Figs. S7 and S8) highlight that parameter perturbations have strong impacts on mass balance and internal accumulation and that equifinality might be an issue, which has not been addressed in this study."

Marlene Kronenberg

*Fribourg, November 15, 2022*